# Partial Channel Dependence with Channel Masks for Time Series Foundation Models

## Abstract

Recent advancements in foundation models have been successfully extended to the time series (TS) domain, facilitated by the emergence of large-scale TS datasets. However, previous efforts have primarily focused on designing model architectures to address explicit heterogeneity among datasets such as various numbers of channels, while often overlooking implicit heterogeneity such as varying dependencies between channels. In this work, we introduce the concept of *partial channel dependence* (PCD) for models capturing channel dependencies (CDs) via attention, which enables a more sophisticated adjustment of CDs based on dataset-specific information. To achieve PCD, we propose a *channel mask* that captures the relationships between channels within a dataset using two key components: 1) a **correlation matrix** that encodes relative dependencies between channels, and 2) **domain parameters** that learn the absolute dependencies specific to each dataset, refining the correlation matrix. We validate the effectiveness of PCD across four tasks in TS including forecasting, classification, imputation, and anomaly detection, under diverse settings, including few-shot and zero-shot scenarios with both TS foundation models and single-task models.

## 1 Introduction

Multivariate time series (MTS) forecasting has been explored with two different strategies: the channel-dependent (CD) strategy and the channel-independent (CI) strategy, with the former emphasizing inter-channel dependencies, while the latter ignoring these dependencies and dealing with channels individually. However, most previous works have focused on the model architecture to either capture or disregard CD, often overlooking the potential differences in CD across datasets.

Foundation models (FMs) have emerged in various domains (Touvron et al., 2023; Rombach et al., 2022; Kirillov et al., 2023), including the time series (TS) domain (Goswami et al., 2024; Liu et al., 2024b). These models are pretrained on diverse datasets and are designed to solve multiple tasks using a single model. However, directly applying FMs to TS is challenging due to the heterogeneity across TS datasets, which can be categorized into two types: explicit and implicit heterogeneity.

*Explicit heterogeneity* arises from observable differences across datasets, such as varying sequence lengths and the number of channels. This poses challenges for a time series foundation model (TSFM), as it must accommodate these varying input shapes within a single model. In contrast, *implicit heterogeneity* stems from unobservable differences, such as varying CD, with some datasets exhibiting strong CD and others showing weak CD. This variability poses a challenge for a TSFM, as it assumes a uniform CI (Goswami et al., 2024) or CD (Woo et al., 2024) model across all datasets, even though each dataset may benefit from a distinct approach (CI or CD), as shown in Figure 1.

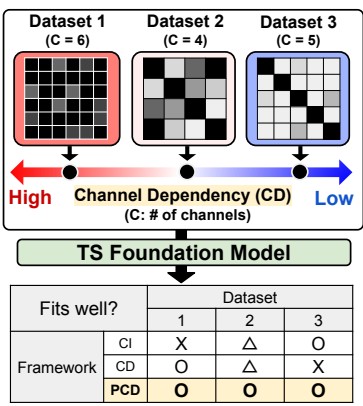

Figure 1: PCD aims to capture the varying CD across datasets.

Despite the importance of addressing both types of heterogeneity, previous TSFMs have primarily concentrated on *explicit heterogeneity* by focusing on the model architecture to accommodate TS inputs with varying shapes (Liu et al., 2024b; Woo et al., 2024), often overlooking implicit heterogeneity. In this paper, we focus on *implicit heterogeneity*, particularly the varying CD across datasets, when building TSFMs. To address this, we consider TSFMs not only in terms of the model architecture for explicit heterogeneity, but also in terms of the *dataset itself*.

To this end, we introduce the concept of *partial channel dependence* (PCD) which adjusts the CD estimated by the Transformer-based model by leveraging the characteristics of the dataset, as capturing the varying CD across datasets with a single model can be challenging. Specifically, we propose a *channel mask* (CM) that adjusts the dependencies between channels to achieve PCD. A CM consists of 1) a **correlation matrix** to encode relative dependencies between channels and 2) **domain parameters** that learn the absolute dependencies specific to each dataset to refine the correlation matrix. The proposed CM, constructed using dataset-specific information, is multiplied to the (channel-wise) attention matrix (i.e., CD estimated by the model). We conduct extensive experiments to validate the effectiveness of CMs with task-specific models and TSFMs on various tasks including forecasting, classification, imputation, and anomaly detection, under various settings such as few-shot and zero-shot. The main contributions are summarized as follows:

- We introduce the concept of partial channel dependence (PCD), where the channel dependence (CD) captured by the model is adjusted based on the characteristics of the TS dataset.
- We propose a channel mask (CM) to achieve PCD, which is a matrix that captures 1) relative dependencies between channels with a correlation matrix, and 2) absolute dependencies specific to each dataset with domain parameters that refine the correlation matrix. The proposed CM is a plug-and-play method applicable to any model that captures CD using an attention mechanism.
- We present extensive experiments with both TSFMs and single-task models across four different tasks under various settings, demonstrating consistent performance gains. For example, applying CMs to TSFMs, e.g., UniTS (Gao et al., 2024), and to single-task models, e.g., iTransformer (Liu et al., 2024a), yields performance gains across all 20 and 13 forecasting tasks, respectively.

## 2 RELATED WORKS

**MTS forecasting models** can be categorized into CI and CD models, where CI models process channels independently without accounting for dependencies between them, whereas CD models account for these dependencies. For CI models, DLinear (Zeng et al., 2023) and RLinear (Li et al., 2023) employ linear models along the time dimension, PatchTST (Nie et al., 2023) divides TS into patches and feeds them into a Transformer (Vaswani et al., 2017) in a CI manner, and PITS (Lee et al., 2024) combines CI and patch independent architectures with multi-layer perceptrons (MLPs). For CD models, Crossformer (Zhang & Yan, 2023) employs a two-stage attention mechanism to capture both temporal dependencies (TD) and CD, TSMixer (Chen et al., 2023) utilizes MLPs with patching to capture both dependencies, and CrossGNN (Huang et al., 2023) employs a linear complexity graph neural network to refine the CD. Recently, iTransformer (Liu et al., 2024a) inverts the traditional Transformer framework in TS domain by treating each channel as a token instead of each patch, thereby shifting the focus from capturing TD to CD, LIFT (Zhao & Shen, 2024) captures the lead-lag relationship between channels, and CDAM (Qi et al., 2024) minimizes redundant information while enhancing relevant mutual information between channels to effectively capture CD. Among these two frameworks, we highlight the importance of CD, as CI can be achieved with CD by disregarding dependencies between irrelevant channels when well captured (Nie et al., 2023). Nonetheless, most CD models primarily focus on architectural solutions for handling CD and often overlook the characteristics of TS datasets, motivating us to consider CD varying across datasets.

**TS foundation models** often borrow knowledge from other fields, such as natural language processing, primarily due to the lack of large-scale datasets in the TS domain. In response to this challenge, there have been efforts to adapt large language models (LLMs) for TS tasks: GPT4TS (Zhou et al., 2023) fine-tunes the embedding layers of LLMs and Time-LLM (Jin et al., 2024) aligns TS data with LLM-based text prototypes to address TS tasks. Recent works have focused on pretraining TSFMs exclusively on TS datasets from various sources. MOMENT (Goswami et al., 2024) and Timer (Liu et al., 2024b) collect extensive and heterogeneous sets of TS datasets to pretrain Transformer-based TSFMs, while MOIRAI (Woo et al., 2024) enhances the Transformer architecture to address domain-specific challenges in constructing TSFMs. UniTS (Gao et al., 2024) proposes a TSFM that handles various tasks with a single architecture through prompt-tuning. These TSFMs either adopt a CI or CD architecture, and we argue that the CD architecture, which is mostly based on the attention mechanism of Transformers, is crucial for TSFMs. This is due to the capacity-robustness trade-off of architectures (Han et al., 2023), with the higher capacity of the CD architecture benefiting larger datasets used for training TSFMs. However, these TSFMs based on CD architectures do not account for the heterogeneity among datasets in terms of CD, while different TS datasets exhibit varying degrees of CD. This motivates us to adjust CD in TSFMs based on the characteristics of each dataset.

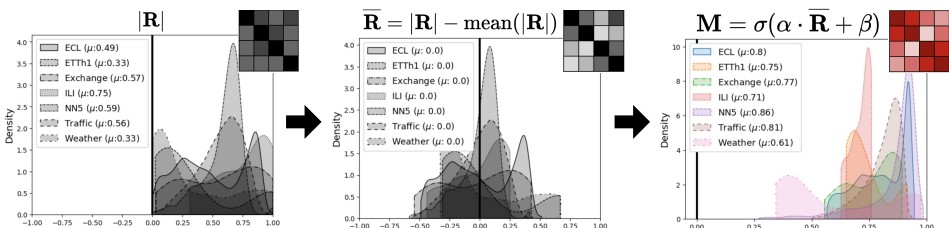

Figure 2: **CM for PCD.** To achieve PCD, we propose a CM, which consists of a correlation matrix between channels and domain parameters that refine the matrix based on the dataset.

Figure 3: **Domain parameters to adjust correlation matrix.** As correlation is a relative measure depending on the dataset, we refine the correlation matrix using the domain parameters. First, we normalize $|\mathbf{R}|$ by subtracting its mean, resulting in $\bar{\mathbf{R}}$. We then scale and shift $\bar{\mathbf{R}}$ using domain parameters $\alpha$ and $\beta$, respectively, and apply a sigmoid function, resulting in $\mathbf{M} = \sigma(\alpha \cdot \bar{\mathbf{R}} + \beta)$.

## 3 METHODOLOGY

In this section, we introduce a channel mask (CM), a simple yet effective method for achieving PCD. A CM employs a correlation matrix to capture relative dependencies between channels and adjusts it with domain parameters to learn absolute dependencies specific to each dataset. We also introduce a new metric, the channel dependence ratio (CD ratio), which uses a CM to quantify the degree of CD for each dataset. The overall framework of a CM is illustrated in Figure 2.

### 3.1 COMPONENTS OF CHANNEL MASK

As shown in Figure 2, a CM consists of two components: 1) correlation matrix ($\mathbf{R}$) between channels, and 2) domain parameters ($\alpha$ and $\beta$), which scale and shift the matrix according to the dataset's characteristics, along with a sigmoid function to normalize the values between 0 and 1.

**Correlation matrix.** Correlation measures the relationships between channels and has been used in previous works to analyze CD (Yang et al., 2024; Zhao & Shen, 2024). Building on these approaches, we employ a correlation matrix ($\mathbf{R}$) between channels to create a CM. However, high correlation does not always indicate a strong positive relationship, as the values range from $-1$ to $1$, with strong negative relationships near $-1$. To address this issue, we use the absolute value of the matrix $|\mathbf{R}|$.

**Domain parameters.** We argue that $|\mathbf{R}|$ alone might be insufficient for modeling a CM for the following reasons: First, correlation is a relative measure that depends on the dataset. As shown in the first panel of Figure 3, different datasets exhibit different distributions of the elements of $|\mathbf{R}|$. To align these differences, we normalize $|\mathbf{R}|$ by subtracting the mean value, resulting in $\bar{\mathbf{R}}$, as shown in the second panel of Figure 3. Second, the relationship between correlation and CD may vary across datasets (i.e., the same correlation can correspond to different levels of CD depending on the dataset). To deal with this discrepancy among datasets, we introduce two learnable domain parameters, $\alpha$ and $\beta$, which scale and shift $|\mathbf{R}|$, respectively, as shown in the third panel of Figure 3. Using these parameters along with a sigmoid function, we model a CM for achieving PCD as $\mathbf{M} = \sigma(\alpha \cdot \bar{\mathbf{R}} + \beta)$.

### 3.2 CHANNEL MASK WITH ATTENTION MATRIX

The proposed CM adjusts the CD estimated by the model by performing element-wise multiplication with the attention matrix of Transformers, with the general adjustment modeled by $\mathbf{A}$:

$$\text{Attn}(\mathbf{Q}, \mathbf{K}, \mathbf{V}) = \text{Softmax}\left(\mathbf{A} \odot \frac{\mathbf{Q}\mathbf{K}^{\top}}{\sqrt{d_k}}\right) \cdot \mathbf{V}, \text{ where } \mathbf{A} = \begin{cases} \mathbf{I}_{C \times C} & \text{if CI,} \\ \mathbf{1}_{C \times C} & \text{if CD,} \\ \mathbf{M} = \sigma(\alpha \cdot \bar{\mathbf{R}} + \beta) & \text{if PCD,} \end{cases} \quad (1)$$

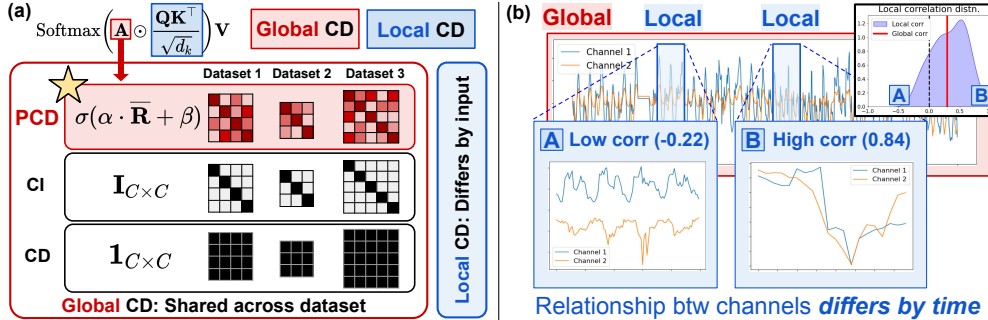

Figure 4: **Global and local dependencies.** (a) shows a CM and an attention matrix, which capture the global and local dependencies between channels, respectively. (b) illustrates the global and local correlations between two channels of ETTh1 (Zhou et al., 2021).

and $C$ is the number of channels. Note that Equation 1 incorporates both CI and CD frameworks within a single expression: As shown in Figure 2, $\mathbf{A}$ is the identity matrix ($\mathbf{I}_{C \times C}$) in the CI framework, while $\mathbf{A}$ is a matrix of ones ($\mathbf{1}_{C \times C}$) in the CD framework. In contrast, our PCD framework represents $\mathbf{A}$ as $\mathbf{M} = \sigma(\alpha \cdot \bar{\mathbf{R}} + \beta)$, enabling a more refined adjustment of CD tailored to the dataset.

**Global and local CD.** As a correlation matrix is calculated based on the entire TS dataset, a CM captures the global CD, which represents the CD shared across all time steps. This complements the local CD captured by the conventional attention matrix, which represents the CD that varies by input time step. As shown in Figure 4(a), our PCD framework captures both global and local CDs through the element-wise multiplication of a CM and an attention matrix ($\mathbf{QK}^\top/\sqrt{d_k}$). Furthermore, Figure 4(b), which illustrates two channels of ETTh1 (Zhou et al., 2021), shows that the dependency can differ across time steps even within the same dataset, underscoring the need to capture both global and local CDs. Further analysis on the necessity of capturing both CDs is discussed in Table 12.

### 3.3 CHANNEL DEPENDENCE RATIO

To quantify the degree of CD for each dataset, we propose to measure the *channel dependence ratio* (CD ratio), a metric based on a CM. The CD ratio of $\mathbf{M}$, denoted as $r(\mathbf{M})$, is the average of the off-diagonal elements of $\mathbf{M}$, excluding the autocorrelations of their respective channels. This metric yields a value of 0 for CI cases and 1 for CD cases, with higher values indicating a greater preference

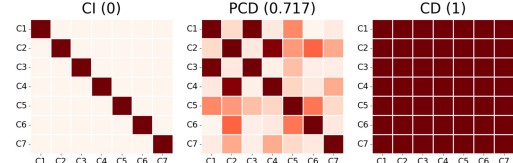

Figure 5: **CD ratio.** CD ratio of $\mathbf{I}_{C \times C}$ for CI, $\sigma(\alpha \cdot \bar{\mathbf{R}} + \beta)$ for PCD, and $\mathbf{1}_{C \times C}$ for CD.

for interaction between channels. Figure 5 shows the visualization of $\mathbf{M}$ and its corresponding CD ratio for ETTh1 (Zhou et al., 2021), with a ratio of 0.717 for PCD. We find that $\mathbf{M}$ effectively captures the degree of CD for each dataset, as datasets with higher $r(\mathbf{M})$ tend to have greater performance gains with CD architecture compared to CI architecture, as illustrated in Figure 7.

## 4 EXPERIMENTS

We demonstrate the effectiveness of our method in both single-task and multi-task scenarios under supervised (SL) or self-supervised (SSL) settings, where we employ iTransformer (iTrans.) (Liu et al., 2024a) for single-task SL, TimeSiam (Dong et al., 2024) for single-task SSL, and UniTS (Gao et al., 2024) for multi-task SSL. As shown in Table 1, we consider four different tasks: forecasting (FCST), classification (CLS), imputation (IMP), and anomaly detection (AD), across various dataset sizes including few-shot and zero-shot settings. As evaluation metrics, we use the mean squared error (MSE) and mean absolute error (MAE) for FCST and IMP, accuracy (Acc.) for CLS, and $F_1$ score for AD. Dataset statistics and implementation details can be found in Appendix A and B, respectively.

| Model | | | TS downstream tasks | | | | Data % | Section | |
|---|---|---|---|---|---|---|---|---|---|
| | | | FCST | CLS | IMP | AD | | Summary | Full |
| Single-task | SL | iTransformer | ✓ | - | - | - | Full | Section 4.1 | Appendix C |
| | SSL | TimeSiam | ✓ | - | - | - | | - | Appendix E |
| Multi-task (FM) | SSL | UniTS | ✓ | ✓ | - | - | Full | Section 4.2.1 | Appendix D.1 |
| | | | ✓ | ✓ | ✓ | ✓ | Few-shot | Section 4.2.2 | Appendix D.2 |
| | | | ✓ | - | - | - | Zero-shot | - | Section 4.2.3 |

Table 1: Summary of experiments.

| 20 Tasks | | Shared (1 model) | | | | | | | | Task-specific (20 models) | | | | | | | |
|---|---|---|---|---|---|---|---|---|---|---|---|---|---|---|---|---|---|
| | | UniTS + CM | | | | UniTS | | | | iTransformer | | TimesNet | | PatchTST | | GPT4TS | |
| | | Sup. | | PT | | Sup. | | PT | | Sup. | | Sup. | | Sup. | | FT | |
| Dataset | $H$ | MSE | MAE | MSE | MAE | MSE | MAE | MSE | MAE | MSE | MAE | MSE | MAE | MSE | MAE | MSE | MAE |
| NN5 | 112 | 0.641 | 0.568 | **0.586** | **0.536** | 0.635 | 0.556 | 0.611 | 0.552 | 0.623 | 0.554 | 0.629 | 0.541 | 0.634 | 0.568 | 0.623 | 0.545 |
| ECL | 96 | 0.176 | 0.278 | **0.168** | **0.272** | 0.172 | 0.273 | 0.174 | 0.277 | 0.204 | 0.288 | 0.184 | 0.289 | 0.212 | 0.299 | 0.198 | 0.285 |
| | 192 | 0.188 | 0.287 | **0.184** | 0.286 | 0.185 | **0.284** | 0.189 | 0.289 | 0.208 | 0.294 | 0.204 | 0.307 | 0.213 | 0.303 | 0.200 | 0.288 |
| | 336 | 0.199 | 0.295 | 0.199 | 0.301 | **0.196** | 0.297 | 0.205 | 0.304 | 0.224 | 0.310 | 0.217 | 0.320 | 0.228 | 0.317 | 0.214 | 0.302 |
| | 720 | **0.230** | **0.321** | 0.231 | 0.326 | 0.238 | 0.321 | 0.251 | 0.340 | 0.265 | 0.341 | 0.284 | 0.363 | 0.270 | 0.348 | 0.254 | 0.333 |
| ETTh1 | 96 | 0.388 | 0.405 | 0.389 | 0.408 | 0.390 | 0.408 | 0.390 | 0.411 | **0.382** | **0.399** | 0.478 | 0.448 | 0.389 | 0.400 | 0.396 | 0.413 |
| | 192 | 0.438 | 0.436 | 0.432 | 0.432 | **0.428** | 0.432 | 0.432 | 0.439 | 0.431 | **0.426** | 0.561 | 0.504 | 0.440 | 0.43 | 0.458 | 0.448 |
| | 336 | 0.478 | 0.455 | 0.475 | 0.451 | **0.462** | 0.451 | 0.480 | 0.460 | 0.476 | **0.449** | 0.612 | 0.537 | 0.482 | 0.453 | 0.508 | 0.472 |
| | 720 | **0.483** | **0.472** | 0.515 | 0.492 | 0.489 | 0.476 | 0.532 | 0.500 | 0.495 | 0.487 | 0.601 | 0.541 | 0.486 | 0.479 | 0.546 | 0.503 |
| Exchange | 192 | 0.231 | 0.340 | 0.210 | 0.330 | 0.239 | 0.342 | 0.221 | 0.337 | **0.175** | **0.297** | 0.259 | 0.370 | 0.178 | 0.301 | 0.177 | 0.300 |
| | 336 | 0.431 | 0.472 | 0.387 | 0.451 | 0.479 | 0.486 | 0.387 | 0.453 | **0.322** | **0.409** | 0.478 | 0.501 | 0.328 | 0.415 | 0.326 | 0.414 |
| ILI | 60 | 2.02 | **0.885** | 2.15 | 0.923 | 2.48 | 0.944 | 2.45 | 0.994 | **1.99** | 0.905 | 2.37 | 0.966 | 2.31 | 0.970 | 1.90 | 0.868 |
| Traffic | 96 | 0.486 | 0.322 | **0.483** | 0.324 | 0.496 | 0.325 | 0.502 | 0.330 | 0.606 | 0.389 | 0.611 | 0.336 | 0.643 | 0.405 | 0.524 | 0.351 |
| | 192 | **0.492** | **0.325** | 0.500 | 0.330 | 0.497 | 0.327 | 0.523 | 0.331 | 0.592 | 0.382 | 0.643 | 0.352 | 0.603 | 0.387 | 0.519 | 0.346 |
| | 336 | **0.506** | 0.331 | 0.520 | 0.337 | 0.509 | **0.328** | 0.552 | 0.338 | 0.600 | 0.384 | 0.662 | 0.363 | 0.612 | 0.389 | 0.530 | 0.350 |
| | 720 | **0.523** | **0.340** | 0.575 | 0.362 | 0.525 | 0.350 | 0.626 | 0.369 | 0.633 | 0.401 | 0.678 | 0.365 | 0.652 | 0.406 | 0.562 | 0.366 |
| Weather | 96 | 0.165 | 0.211 | 0.166 | 0.219 | **0.161** | **0.211** | 0.175 | 0.214 | 0.193 | 0.232 | 0.169 | 0.220 | 0.194 | 0.233 | 0.182 | 0.222 |
| | 192 | **0.210** | **0.254** | 0.216 | 0.261 | 0.212 | 0.255 | 0.226 | 0.266 | 0.238 | 0.269 | 0.223 | 0.264 | 0.238 | 0.268 | 0.228 | 0.261 |
| | 336 | **0.266** | **0.294** | 0.273 | 0.300 | 0.266 | 0.295 | 0.280 | 0.303 | 0.291 | 0.306 | 0.279 | 0.302 | 0.290 | 0.304 | 0.282 | 0.299 |
| | 720 | **0.342** | **0.343** | 0.350 | 0.349 | 0.343 | 0.344 | 0.352 | 0.350 | 0.365 | 0.354 | 0.359 | 0.355 | 0.363 | 0.35 | 0.359 | 0.349 |
| Best Count (/20) | | 8 | 11 | 4 | 2 | 5 | 4 | 0 | 0 | 4 | 5 | 0 | 0 | 0 | 0 | - | - |
| Average | | **0.445** | **0.382** | 0.452 | 0.384 | 0.469 | 0.386 | 0.478 | 0.393 | 0.466 | 0.394 | 0.525 | 0.412 | 0.488 | 0.401 | 0.449 | 0.386 |

Table 3: **Results of multi-task forecasting.** Applying a CM to UniTS results in SOTA performance, outperforming standard UniTS and other task-specific models. In particular, it brings improvements across all 20 forecasting tasks under prompt-tuning settings.

## 4.1 SINGLE-TASK MODEL: APPLICATION TO ITRANSFORMER

To demonstrate the effectiveness of our method, we apply our method to iTransformer (Liu et al., 2024a) to solve TS forecasting tasks on 13 datasets. Table 2 presents the average MSE and MAE across four different horizons ($H$), showing consistent improvement across all datasets. Specifically, the performance gains in MSE on the PEMS datasets (Liu et al., 2022) (03, 04, 07, 08) are significantly large (12.7%, 19.0%, 19.6%, 40.2%), whereas the gains on the ETT datasets (Zhou et al., 2021) (h1, h2, m1, m2) are relatively small (2.8%, 0.3%, 2.5%, 1.4%), suggesting a potential variation in the need for a CM across different datasets. Full results are described in Appendix C.1.

| Dataset | iTransformer | | + CM | | Impr. | |
|---|---|---|---|---|---|---|
| | MSE | MAE | MSE | MAE | MSE | MAE |
| ETTh1 | 0.457 | 0.449 | **0.444** | **0.441** | **2.8%** | **1.8%** |
| ETTh2 | 0.384 | 0.407 | **0.383** | **0.406** | **0.3%** | **0.2%** |
| ETTm1 | 0.408 | 0.412 | **0.398** | **0.406** | **2.5%** | **1.5%** |
| ETTm2 | 0.293 | 0.337 | **0.289** | **0.335** | **1.4%** | **0.6%** |
| PEMS03 | 0.142 | 0.248 | **0.124** | **0.231** | **12.7%** | **6.9%** |
| PEMS04 | 0.121 | 0.232 | **0.098** | **0.210** | **19.0%** | **9.5%** |
| PEMS07 | 0.102 | 0.205 | **0.082** | **0.183** | **19.6%** | **10.7%** |
| PEMS08 | 0.254 | 0.306 | **0.152** | **0.231** | **40.2%** | **24.5%** |
| Exchange | 0.368 | 0.409 | **0.363** | **0.406** | **1.4%** | **0.7%** |
| Weather | 0.260 | 0.281 | **0.250** | **0.275** | **3.8%** | **2.1%** |
| Solar | 0.234 | 0.261 | **0.228** | **0.258** | **2.6%** | **1.1%** |
| ECL | 0.179 | 0.270 | **0.168** | **0.262** | **6.1%** | **3.0%** |
| Traffic | 0.428 | 0.282 | **0.422** | **0.281** | **1.4%** | **0.4%** |
| Avg. | 0.279 | 0.315 | **0.261** | **0.302** | **6.3%** | **4.3%** |

Table 2: FCST on single-task model.

## 4.2 MULTI-TASK MODEL: APPLICATION TO UNITS

To validate the effectiveness of our method on a TS foundation model, we apply it to UniTS (Gao et al., 2024) which solves diverse tasks without the need for fine-tuning, relying solely on prompt-tuning.

### 4.2.1 FORECASTING AND CLASSIFICATION TASKS

Table 4 presents a summary of the results from 20 forecasting tasks and 18 classification tasks under both supervised (Sup.) and prompt-tuning (PT) settings, with the full results for both tasks provided in Table 3 and Appendix D.1, respectively. The results indicate that applying our method improves performance in all 20 FCST and 13 CLS tasks. Notably, our method outperforms task-specific models that are individually trained for each task, while our model remains a single shared

| | | UniTS | + CM | Impr. |
|---|---|---|---|---|
| FCST (MSE) | Sup. | 0.469 | **0.445** | **5.1%** |
| | PT | 0.478 | **0.452** | **5.4%** |
| CLS (Acc.) | Sup. | 80.6 | **82.0** | **1.7%** |
| | PT | 75.1 | **78.3** | **4.3%** |

Table 4: 20 FCST and 18 CLS tasks.

model capable of solving various tasks without fine-tuning. Additionally, compared to GPT4TS (Zhou et al., 2023), which is a TSFM that reprograms the pretrained GPT-2 model (Radford et al., 2019), our method achieves superior performance with less than 1% of the parameters (164.5M vs. 1.57M).

| Ratio | Model | | MSE | Acc. |
|---|---|---|---|---|
| | iTransformer | FT | 0.598 | 51.4 |
| 5% | UniTS | PT | 0.549 | 49.4 |
| | | FT | 0.505 | 53.8 |
| | UniTS + CM | PT | 0.546 | **54.9** |
| | | FT | **0.489** | 54.8 |
| | iTransformer | FT | 0.524 | 56.5 |
| 15% | UniTS | PT | 0.525 | 53.2 |
| | | FT | 0.487 | 59.7 |
| | UniTS + CM | PT | 0.522 | 55.4 |
| | | FT | **0.481** | **60.4** |
| | iTransformer | FT | 0.510 | 59.9 |
| 20% | UniTS | PT | 0.525 | 58.9 |
| | | FT | 0.486 | 63.6 |
| | UniTS + CM | PT | 0.453 | 60.0 |
| | | FT | **0.425** | **64.8** |

(a) 9 FCST and 6 CLS tasks.

| Ratio | Model | | MSE |
|---|---|---|---|
| | TimesNet | FT | 0.246 |
| | PatchTST | FT | 0.191 |
| | iTransformer | | 0.186 |
| 25% | UniTS | PT | 0.179 |
| | | FT | 0.167 |
| | UniTS + CM | PT | 0.179 |
| | | FT | **0.158** |
| | TimesNet | FT | 0.292 |
| | PatchTST | FT | 0.236 |
| | iTransformer | | 0.226 |
| 50% | UniTS | PT | 0.232 |
| | | FT | 0.213 |
| | UniTS + CM | PT | 0.225 |
| | | FT | **0.201** |

(b) 6 IMP tasks.

| Model | | $F_1$ |
|---|---|---|
| Anomaly Trans. | - | 79.2 |
| TimesNet | FT | 74.2 |
| PatchTST | FT | 84.3 |
| iTransformer | FT | 83.1 |
| UniTS | PT | 81.7 |
| | FT | 85.6 |
| UniTS + CM | PT | 82.0 |
| | FT | **86.6** |

(c) 5 AD tasks.

Table 5: Four tasks under few-shot settings.

| Dataset | UniTS | | + CM | | Impr. | |
|---|---|---|---|---|---|---|
| | MSE | MAE | MSE | MAE | MSE | MAE |
| Solar | 0.597 | 0.607 | **0.586** | **0.585** | **1.9%** | **3.6%** |
| River | **1.374** | 0.698 | **1.374** | **0.686** | 0.0% | **1.7%** |
| Hospital | 1.067 | 0.797 | **1.020** | **0.777** | **4.4%** | **2.5%** |
| Avg. | 1.013 | 0.701 | **0.993** | **0.683** | **2.0%** | **2.6%** |

(a) Zero-shot dataset.

| Dataset | UniTS | | + CM | | Impr. | |
|---|---|---|---|---|---|---|
| | MSE | MAE | MSE | MAE | MSE | MAE |
| ECL | 0.237 | 0.329 | **0.231** | **0.323** | **2.5%** | **1.8%** |
| ETTh1 | 0.495 | 0.463 | **0.492** | 0.463 | **0.6%** | 0.0% |
| Traffic | 0.632 | 0.372 | **0.592** | **0.369** | **6.3%** | **0.8%** |
| Weather | 0.335 | 0.336 | 0.335 | 0.336 | 0.0% | 0.0% |

(b) Zero-shot horizon.

Table 6: Zero-shot FCST tasks.

### 4.2.2 FEW-SHOT LEARNING

For the tasks under the few-shot settings, we conduct four different tasks (FCST, CLS, IMP, AD), following the experimental settings of UniTS. Full results are described in Appendix D.2.

**Few-shot FCST and CLS.** We experiment nine forecasting tasks and six classification tasks under the few-shot settings with data ratios of 5%, 15%, and 20%. Table 5a presents the results, which indicates that our method outperforms both iTransformer and UniTS in both PT and fine-tuning (FT) settings.

**Few-shot IMP.** We experiment six imputation tasks under the few-shot setting with a data ratio of 10%, where the goal is to impute 25% and 50% of missing data points. Table 5b presents the results, indicating that our method outperforms UniTS and other state-of-the-art (SOTA) single-task models (Wu et al., 2023; Nie et al., 2023; Liu et al., 2024a) in both PT and FT settings.

**Few-shot AD.** We experiment five anomaly detection tasks under the few-shot setting with a data ratio of 5%, where the results in Table 5c indicate that our method outperforms UniTS and other SOTA methods in both PT and FT settings.

### 4.2.3 ZERO-SHOT LEARNING

We perform TS forecasting tasks under two types of zero-shot settings: 1) *Zero-shot dataset*: We evaluate our model on an unseen dataset that was not included during training. 2) *Zero-shot task*: We assess the model's ability to predict a new forecasting horizon that was not encountered during training, by adding the mask tokens at the end of the TS to predict the desired future time steps.

**Zero-shot dataset.** For the TS forecasting task on unseen datasets, we evaluate our method using three datasets (NREL, 2006; McLeod & Gweon, 2013; Hyndman et al., 2008). Table 6a presents the results, demonstrating consistent improvements by incorporating CMs.

**Zero-shot horizon.** For the TS forecasting task with new forecasting horizons, we predict additional 384 time steps (by adding 24 masked tokens of length 16 at the end of the TS) on top of the base forecasting horizon of 96. Table 6b presents the results with four different datasets (Zhou et al., 2021; Wu et al., 2021), showing performance gains on three out of four datasets.

| Domain params. | | ✗ | ✓ |
|---|---|---|---|
| Dataset | $C$ | $r(|\mathbf{R}|)$ | $r(\mathbf{M})$ |
| Weather | 21 | 0.296 (2) | 0.587 (1) |
| ILI | 7 | 0.708 (7) | 0.706 (2) |
| ETTh1 | 7 | 0.222 (1) | 0.717 (3) |
| Exchange | 8 | 0.513 (4) | 0.749 (4) |
| ECL | 321 | 0.489 (3) | 0.800 (5) |
| Traffic | 862 | 0.564 (5) | 0.808 (6) |
| NN5 | 111 | 0.584 (6) | 0.857 (7) |

Table 8: CD ratio comparison with rank.

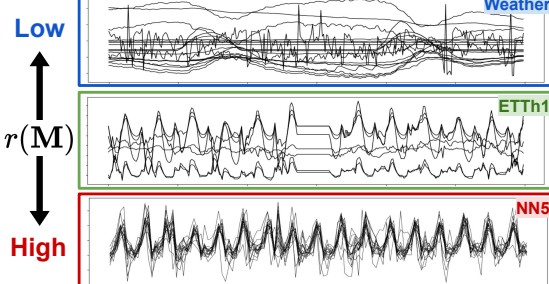

Figure 6: TS visualization by $r(\mathbf{M})$.

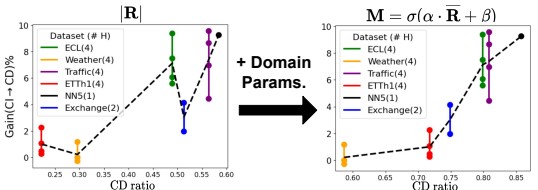

Figure 7: Performance gain by CD vs. CD ratio.

| Method | Dataset | MSE | MAE |
|---|---|---|---|
| UniTS | | 1.006 | 0.701 |
| + CM | FCST + CLS | 0.995 | 0.684 |
| | FCST | **0.993** | **0.683** |
| | Closest | **0.993** | **0.683** |

Table 9: Domain params for unseen datasets.

## 5 ANALYSIS

**Effectiveness of CM.** To demonstrate the effectiveness of a CM, we conduct an ablation study using the correlation matrix (Corr.) and the domain parameters (Dom.). Table 7 presents the result with 20 forecasting tasks and 18 classification tasks with UniTS under the prompt-tuning setting, indicating that incorporating both components yields the best performance. Note that, to isolate the effect of the domain parameters, we replace $\bar{\mathbf{R}}$ with the identity matrix ($\mathbf{I}$) in the forth row of Table 7.

| Components | | A | FCST (20) | | CLS (18) |
|---|---|---|---|---|---|
| Corr. | Dom. | | MSE | MAE | Acc. |
| | | $\mathbf{I}$ | 0.502 | 0.408 | 75.4% |
| | | $\mathbf{1}$ | 0.478 | 0.393 | 75.1% |
| ✓ | | $|\mathbf{R}|$ | 0.474 | 0.390 | 78.8% |
| ✓ | | $\bar{\mathbf{R}}$ | 0.471 | 0.388 | 78.1% |
| | ✓ | $\sigma\left(\alpha \cdot \mathbf{I} + \beta\right)$ | 0.497 | 0.406 | 76.2% |
| ✓ | ✓ | $\sigma\left(\alpha \cdot \bar{\mathbf{R}} + \beta\right)$ | **0.452** | **0.384** | **80.6%** |

Table 7: Ablation study of CM.

**CD ratio comparison.** Table 8 presents the CD ratios of CMs with and without[1] domain parameters ($r(\mathbf{M})$ and $r(|\mathbf{R}|)$), when using UniTS. The results show that while datasets with higher $r(|\mathbf{R}|)$ generally have higher $r(\mathbf{M})$, this relationship is not consistent; for instance, Weather (Wu et al., 2021) exhibits lower CD despite having a stronger correlation compared to ETTh1 (Zhou et al., 2021). Figure 6 supports these findings by visualizing the channels of the datasets, revealing that the channels of ETTh1 tend to be more dependent on each other than those of Weather. These results underscore the importance of using domain parameters to adjust $|\mathbf{R}|$ for learning absolute dependencies specific to each dataset. Furthermore, datasets with a larger number of channels ($C$) tend to have higher $r(\mathbf{M})$, which aligns with the prior work (Ahamed & Cheng, 2024) emphasizing CD over CI for datasets with more channels.

**Effectiveness of domain parameters.** To demonstrate the importance of domain parameters in reflecting the degree of CD, we compare the CD ratio and the performance gain achieved with the CD framework against the CI framework with UniTS. Figure 7 shows that the gain is highly correlated with the CD ratio of a CM with the domain parameters ($r(\mathbf{M})$), but less so without them ($r(|\mathbf{R}|)$).

**Domain parameters for unseen dataset.** For an unseen dataset, selecting the appropriate domain parameters is challenging, as these parameters are not learned during training. To address this issue, we propose three strategies: 1) averaging the parameters across all datasets, 2) averaging the parameters from the forecasting datasets, and 3) selecting parameters from the dataset with the closest $r(\bar{\mathbf{R}})$. Table 9 demonstrates the robustness of these strategies, consistently outperforming UniTS.

**Visualization of CM.** Figure 8 shows the CMs of ECL (Wu et al., 2021) and ETTh1 (Zhou et al., 2021), illustrating the dependencies between the channels of each dataset. The CM of ETTh1 reveals a hidden relationship between the first and fifth channels when using domain parameters, which is not identified by the correlation matrix alone.

---

[1]For a CM without domain parameters, we use the absolute correlation matrix ($|\mathbf{R}|$) instead of its zero-centered scaled version ($\bar{\mathbf{R}}$) to ensure a fair comparison with $\mathbf{M}$, which is also scaled between 0 and 1.

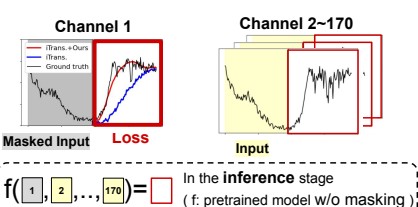

Figure 8: **Visualization of CMs w/ and w/o domain parameters**. The figure shows the correlation matrices and the CMs of two datasets, with each color scaled from 0 (light) to 1 (dark).

Figure 9: Masked channel prediction.

| $H$ | PEMS04 ($C = 307$) | | | PEMS08 ($C = 170$) | | |
|---|---|---|---|---|---|---|
| | iTrans. | + CM | Impr. | iTrans. | + CM | Impr. |
| 12 | 0.549 | **0.300** | **45.4%** | 0.628 | **0.200** | **68.1%** |
| 24 | 0.718 | **0.351** | **51.1%** | 0.678 | **0.241** | **64.5%** |
| 48 | 0.750 | **0.409** | **45.5%** | 1.197 | **1.059** | **11.5%** |
| 96 | 0.758 | **0.513** | **32.3%** | 1.375 | **1.217** | **11.5%** |
| Avg. | 0.694 | **0.393** | **43.3%** | 0.970 | **0.679** | **29.9%** |

Table 11: Results of masked channel prediction.

| Components | | Average MSE across four horizons | | | | | | | | | | | | | Avg. |
|---|---|---|---|---|---|---|---|---|---|---|---|---|---|---|---|
| Global | Local | ETTh1 | ETTh2 | ETTm1 | ETTm2 | PEMS03 | PEMS04 | PEMS07 | PEMS08 | Exchange | Weather | Solar | ECL | Traffic | |
| ✓ | | 0.466 | **0.383** | **0.398** | **0.289** | 0.206 | 0.116 | 0.101 | 0.162 | **0.363** | 0.259 | 0.233 | 0.176 | 0.429 | 0.275 |
| | ✓ | 0.457 | 0.384 | 0.408 | 0.293 | 0.142 | 0.121 | 0.102 | 0.254 | 0.368 | 0.260 | 0.234 | 0.179 | 0.428 | 0.279 |
| ✓ | ✓ | **0.444** | **0.383** | **0.398** | **0.289** | **0.124** | **0.098** | **0.082** | **0.152** | **0.363** | **0.250** | **0.228** | **0.168** | **0.422** | **0.261** |

Table 12: Effect of capturing global and local CD.

**Various TS metrics.** To demonstrate the effectiveness of CMs using metrics beyond (Pearson) correlation, we apply CMs to iTransformer with three different metrics: 1) Euclidean distance (Euc.), which we min-max normalize to the range (0,1) and subtract from 1 to convert it into a similarity metric; 2) cosine similarity (Cos.), for which we take the absolute value, following the same intuition as correlation; and 3) dynamic time warping (DTW), where we apply the same process as with the Euclidean distance. Table 10 presents the TS forecasting result in terms of average MSE for four different horizons, indicating that CMs yield a performance gain regardless of the metric used, with the best performance achieved with correlation. Note that we use DTW only for datasets with fewer than 100 channels due to its computational complexity.

| Dataset | w/o CM | w/ CM | | | |
|---|---|---|---|---|---|
| | | Euc. | Cos. | DTW | Corr. |
| ETTh1 | 0.457 | 0.445 | 0.446 | **0.444** | **0.444** |
| ETTh2 | 0.384 | 0.384 | 0.384 | 0.385 | **0.383** |
| ETTm1 | 0.408 | 0.402 | 0.403 | 0.401 | **0.398** |
| ETTm2 | 0.293 | 0.292 | 0.290 | 0.292 | **0.289** |
| PEMS03 | 0.142 | 0.146 | 0.134 | - | **0.124** |
| PEMS04 | 0.121 | 0.111 | 0.105 | - | **0.098** |
| PEMS07 | 0.102 | 0.092 | 0.087 | - | **0.082** |
| PEMS08 | 0.254 | 0.163 | 0.179 | - | **0.152** |
| Exchange | 0.368 | 0.364 | **0.363** | 0.364 | **0.363** |
| Weather | 0.260 | 0.256 | 0.255 | 0.254 | **0.250** |
| Solar | 0.234 | 0.232 | 0.229 | - | **0.228** |
| ECL | 0.179 | 0.173 | 0.171 | - | **0.168** |
| Traffic | 0.428 | 0.432 | 0.443 | - | **0.422** |
| Avg. | 0.279 | 0.269 | 0.268 | - | **0.261** |

Table 10: Various metrics for CMs.

**Masked channel prediction.** To evaluate the model's ability to capture CD, we introduce a novel evaluation method, *masked channel prediction*, which involves predicting the future values of the masked channel using the historical values of the unmasked channels. Specifically, we calculate the average loss for each channel when masked once, with the loss for the $c$-th channel expressed as:

$$L_{(c)}(y, \hat{y}) = \text{MSE}(y[:, c], \hat{y}_{(c)}[:, c]), \quad \text{where } \hat{y}_{(c)} = f(x_{(c)}), \tag{2}$$

where $x_{(c)}$ is $x$ with the $c$-th channel masked, and $\hat{y}_{(c)}$ is the predicted output using $x_{(c)}$ as the input. Note that masked channel prediction is an *evaluation method* that does not require additional training, and instead uses a model pretrained without any masking.

To assess the effectiveness of CMs in capturing CD, we experiment masked channel prediction with iTransformer with and without CMs, imputing the historical values of the masked channels with there average values, which are essentially zero with normalization. The results in Table 11 demonstrate significant improvements by incorporating CMs. Furthermore, Figure 9 visualizes the predicted results for PEMS08 (Liu et al., 2022), where models with CMs predict masked channels better than models without CMs. We provide more results in Appendix H.

**Global & local CD.** To demonstrate the effect of attention matrices capturing the local CD of the input TS and CMs capturing the global CD of the entire TS, we conduct an ablation study, as shown in Table 12. Specifically, to observe the local, global, and combined effects, we use the attention weights $\mathbf{W}$ in $\text{Attn}(\mathbf{Q}, \mathbf{K}, \mathbf{V}) = \text{Softmax}(\mathbf{W}) \cdot \mathbf{V}$ in the following manner: $\mathbf{Q}\mathbf{K}^\top / \sqrt{d_k}$ for local

| | ETTh1 | ETTh2 | ETTm1 | ETTm2 | PEMS03 | PEMS04 | PEMS07 | PEMS08 | Exchange | Weather | Solar | ECL | Traffic | Avg. |
|---|---|---|---|---|---|---|---|---|---|---|---|---|---|---|
| | | | | | | Average MSE across four horizons | | | | | | | | |
| $\alpha, \beta$ | **0.444** | **0.383** | **0.398** | **0.289** | **0.124** | **0.098** | **0.082** | **0.152** | **0.363** | **0.250** | 0.228 | **0.168** | 0.422 | **0.261** |
| E | 0.452 | 0.391 | 0.402 | 0.291 | 0.150 | 0.106 | 0.096 | 0.202 | 0.364 | 0.255 | 0.234 | 0.177 | 0.416 | 0.272 |
| $\mathbf{E_1}, \mathbf{E_2}$ | 0.452 | 0.391 | 0.402 | 0.291 | 0.152 | 0.105 | **0.095** | 0.205 | 0.364 | 0.255 | 0.233 | 0.177 | **0.415** | 0.272 |
| **A** | 0.454 | 0.391 | 0.402 | 0.291 | 0.138 | 0.099 | 0.102 | 0.182 | 0.364 | 0.259 | **0.226** | 0.177 | 0.418 | 0.269 |
| - | 0.457 | 0.384 | 0.408 | 0.293 | 0.142 | 0.121 | 0.102 | 0.254 | 0.368 | 0.260 | 0.234 | 0.179 | 0.428 | 0.279 |

Table 13: **Results of various domain parameters.** Using scalar domain parameters $(\alpha, \beta)$ which scale and shift the correlation matrix yields the best results.

| $L, H = 96$ | Weather ($C = 21$) | | | ECL ($C = 321$) | | |
|---|---|---|---|---|---|---|
| | iTrans. | - | + CM | iTrans. | - | + CM |
| Attention matrix | ✓ | | ✓ | ✓ | | ✓ |
| Channel mask | | ✓ | ✓ | | ✓ | ✓ |
| Train (sec/epoch) | 26.2 | 24.1 | 26.7 | 33.2 | 26.0 | 36.4 |
| Inference (ms) | 11.1 | 11.1 | 11.2 | 12.4 | 11.0 | 13.2 |
| Avg. MSE | 0.260 | 0.259 | **0.250** | 0.179 | 0.176 | **0.168** |

Table 14: Efficiency analysis.

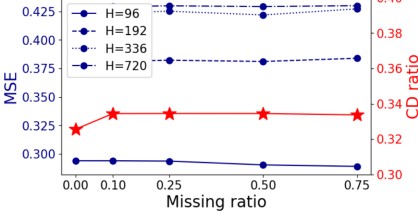

Figure 10: Robustness to missingness.

CD, $\mathbf{M}$ for global CD, and $\mathbf{M} \odot \mathbf{QK}^{\top}/\sqrt{d_k}$ for both. The results show the average MSE for four different horizons, indicating that using both components yields the best results. Additionally, using only CMs provides better performance than attention matrices in some datasets.

**Extending domain parameters.** The proposed domain parameters $\alpha$ and $\beta$ are scalars that adjust $\bar{\mathbf{R}}$ by changing its elements monotonically. For further flexibility, we design alternative options for the parameters: 1) a vector $\mathbf{E}$ for each channel and 2) a matrix $\mathbf{A}$ for each dataset. Both options are used to construct an adjustment matrix that is element-wise multiplied to $\bar{\mathbf{R}}$, as shown in Table 15. The first

| Domain parameters | | Channel mask ($\mathbf{M}$) | Asym. |
|---|---|---|---|
| Scalar | $\alpha, \beta \in \mathbb{R}^1$ | $\sigma\left(\alpha \cdot \bar{\mathbf{R}} + \beta\right)$ | ✗ |
| Vector | $\mathbf{E} \in \mathbb{R}^d$ | $\text{Norm}(\mathbf{E}\mathbf{E}^T) \odot \bar{\mathbf{R}}$ | ✗ |
| | $\mathbf{E_1}, \mathbf{E_2} \in \mathbb{R}^d$ | $\text{Norm}(\mathbf{E_1}\mathbf{E_2}^T) \odot \bar{\mathbf{R}}$ | ✓ |
| Matrix | $\mathbf{A} \in \mathbb{R}^{C \times C}$ | $\mathbf{A} \odot \bar{\mathbf{R}}$ | ✓ |

Table 15: Extension of domain parameters.

option serves as identifiable vectors for each channel, with the adjustment matrix constructed based on the inner product between these vectors and normalized with $\text{Norm}(\cdot) = \text{Softmax}(\text{ReLU}(\cdot))$, while the second option acts as the adjustment matrix itself. For the vector parameters, we also implement an asymmetric matrix version that requires two different vectors for each channel: one for the inner vector ($\mathbf{E_1}$) and the other for the outer vector ($\mathbf{E_2}$), as described in the previous work (Wu et al., 2019). Table 13 shows that using scalar parameters achieves the best performance, demonstrating the efficiency of CMs by requiring only two additional parameters per dataset.

**Efficiency analysis.** To demonstrate the efficiency of CMs, we compare the training and inference times of iTransformer on two datasets (Wu et al., 2021) with varying numbers of channels, using only attention matrices, only CMs, and both. Table 14 indicates that incorporating CMs does not significantly impact computational time, even with datasets containing a large number of channels, with training time measured per epoch and inference time measured per data instance. It is important to note that correlation matrices can be precomputed offline, making CMs practical for use.

**Robustness to missing values.** To demonstrate the robustness of our method to missing values, we analyze scenarios where some TS values are randomly missing at ratios of 10%, 25%, 50%, and 75%, with the missing values linearly interpolated using adjacent values. Figure 10 shows the result on ETTh2 (Zhou et al., 2021) using iTransformer, indicating that both $r(|\bar{\mathbf{R}}|)$ and the performance remain robust despite the missing values, making our method applicable in real-world scenarios.

## 6 CONCLUSION

In this work, we introduce the concept of PCD to adjust the CD estimated by the model using a CM, a plug-and-play method that captures both relative and absolute dependencies between channels using dataset-specific information. Our results demonstrate that incorporating prior knowledge of datasets is crucial when building TSFMs, leading to superior performance across various models and settings. However, since our method can only be applied to Transformer-based methods, which are the most widely used architecture for FMs, we aim to develop a novel approach to achieve PCD without relying on Transformer-based methods in the future. We hope our work highlights the importance of utilizing dataset-specific information when building FMs across different domains.

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

# A    DATASET DESCRIPTION

## A.1    DATASET FOR SINGLE-TASK MODEL: ITRANSFORMER

For TS forecasting in a single-task setting, we evaluate the effectiveness of our proposed method using 13 datasets, with their statistics described in Table A.1. We adhere to the same data processing and train-validation-test split protocol as iTransformer (Liu et al., 2024a), ensuring that the training, validation, and test sets are separated in chronological order. The input length is consistently set to 96 across all datasets. Note that $N$ and $C$ denote the size of the dataset and number of channels in a dataset, respectively.

| Dataset | $C$ | Prediction Length | $(N_{\text{train}}, N_{\text{val}}, N_{\text{test}})$ |
|---|---|---|---|
| ETTh1 (Zhou et al., 2021) | 7 | {96, 192, 336, 720} | (8545, 2881, 2881) |
| ETTh2 (Zhou et al., 2021) | 7 | {96, 192, 336, 720} | (8545, 2881, 2881) |
| ETTm1 (Zhou et al., 2021) | 7 | {96, 192, 336, 720} | (34465, 11521, 11521) |
| ETTm2 (Zhou et al., 2021) | 7 | {96, 192, 336, 720} | (34465, 11521, 11521) |
| Exchange (Wu et al., 2021) | 8 | {96, 192, 336, 720} | (5120, 665, 1422) |
| Weather (Wu et al., 2021) | 21 | {96, 192, 336, 720} | (36792, 5271, 10540) |
| ECL (Wu et al., 2021) | 321 | {96, 192, 336, 720} | (18317, 2633, 5261) |
| Traffic (Wu et al., 2021) | 862 | {96, 192, 336, 720} | (12185, 1757, 3509) |
| Solar-Energy (Lai et al., 2018) | 137 | {96, 192, 336, 720} | (36601, 5161, 10417) |
| PEMS03 (Liu et al., 2022) | 358 | {12, 24, 48, 96} | (15617, 5135, 5135) |
| PEMS04 (Liu et al., 2022) | 307 | {12, 24, 48, 96} | (10172, 3375, 3375) |
| PEMS07 (Liu et al., 2022) | 883 | {12, 24, 48, 96} | (16911, 5622, 5622) |
| PEMS08 (Liu et al., 2022) | 170 | {12, 24, 48, 96} | (10690, 3548, 3548) |

Table A.1: Single-task forecasting datasets.

## A.2 DATASET FOR MULTI-TASK MODEL: UNITS

The datasets used in the experiment are aggregated from the Monash Forecasting Repository (Goda-hewa et al., 2021), the Time Series Classification Website (Middlehurst et al., 2024), and the Time Series Library (Wu et al., 2023). The combined training set includes more than 35 million time steps and over 6,000 variables (channels). Note that $N$, $L$, $C$ denote the training size, input length, and number of channels in a dataset, respectively.

### A.2.1 MULTI-TASK LEARNING

For TS forecasting and classification in a multi-task setting, we evaluate the effectiveness of our proposed method using 20 datasets for forecasting and 18 datasets for classification. The statistics of these datasets are summarized in Table A.2 and A.3, respectively.

| Category | Dataset | Prediction Length | $N$ | $L$ | $C$ |
|---|---|---|---|---|---|
| Finance | NN5 (Taieb et al., 2012) | 112 | 409 | 112 | 111 |
| | Exchange (Wu et al., 2021) | 192
336 | 5024
4880 | 96 | 8 |
| Electricity | ECL (Wu et al., 2021) | 96
192
336
720 | 18221
18125
17981
17597 | 96 | 321 |
| | ETTh1 (Zhou et al., 2021) | 96
192
336
720 | 8449
8353
8209
7825 | 96 | 7 |
| Illness | ILI (Wu et al., 2021) | 60 | 581 | 36 | 7 |
| Traffic | Traffic (Wu et al., 2021) | 96
192
336
720 | 12089
11993
11849
11465 | 96 | 862 |
| Weather | Weather (Wu et al., 2021) | 96
192
336
720 | 36696
36600
36456
36072 | 96 | 21 |

Table A.2: Multi-task forecasting datasets.

| Category | Dataset | # classes | $N$ | $L$ | $C$ |
|---|---|---|---|---|---|
| Finance | SharePriceIncrease (Dau et al., 2019) | 2 | 965 | 60 | 1 |
| Audio | JapaneseVowels (Bagnall et al., 2018) | 9 | 270 | 29 | 12 |
| | SpokenArabicDigits (Bagnall et al., 2018) | 10 | 6599 | 93 | 13 |
| | Heartbeat (Bagnall et al., 2018) | 2 | 204 | 405 | 61 |
| ECG | ECG5000 (Dau et al., 2019) | 5 | 500 | 140 | 1 |
| | NonInvasiveFetalECGThorax1 (Dau et al., 2019) | 52 | 1800 | 750 | 1 |
| EEG | Blink (Bagnall et al., 2018) | 2 | 500 | 510 | 4 |
| | FaceDetection (Bagnall et al., 2018) | 2 | 5890 | 62 | 144 |
| | SelfRegulationSCP2 (Bagnall et al., 2018) | 2 | 200 | 1152 | 7 |
| Sensors | ElectricDevices (Dau et al., 2019) | 7 | 8926 | 96 | 1 |
| | Trace (Dau et al., 2019) | 4 | 100 | 275 | 1 |
| | FordB (Dau et al., 2019) | 2 | 3636 | 500 | 1 |
| Human Activity | MotionSenseHAR (Bagnall et al., 2018) | 6 | 966 | 200 | 12 |
| | EMOPain (Bagnall et al., 2018) | 3 | 968 | 180 | 30 |
| | UWaveGestureLibrary (Bagnall et al., 2018) | 8 | 120 | 315 | 3 |
| Traffic | Chinatown (Dau et al., 2019) | 2 | 20 | 24 | 1 |
| | MelbournePedestrian (Dau et al., 2019) | 10 | 1194 | 24 | 1 |
| | PEMS-SF (Bagnall et al., 2018) | 7 | 267 | 144 | 963 |

Table A.3: Multi-task classification datasets.

### A.2.2 FEW-SHOT LEARNING

For TS forecasting, classification, imputation, and anomaly detection in a few-shot setting, we evaluate the effectiveness of our proposed method using nine datasets for forecasting, six datasets for classification, four datasets for imputation, and five datasets for anomaly detection. The statistics of these datasets related to forecasting and classification are summarized in Table A.4, Table A.5, A.6, and A.7, respectively.

| Category | Dataset | Prediction Length | $N$ | $L$ | $C$ |
|---|---|---|---|---|---|
| Electricity | ETTh2 (Zhou et al., 2021) | 96
192
336
720 | 8449
8353
8209
7825 | 96 | 7 |
| | ETTm1 (Zhou et al., 2021) | 96
192
336
720 | 34369
34273
34129
33745 | 96 | 7 |
| Weather | SaugeenRiverFlow (McLeod & Gweon, 2013) | 24 | 18921 | 48 | 1 |

Table A.4: Few-shot forecasting datasets.

| Category | Dataset | # classes | $N$ | $L$ | $C$ |
|---|---|---|---|---|---|
| ECG | ECG200 (Dau et al., 2019) | 2 | 100 | 96 | 1 |
| EEG | SelfRegulationSCP1 (Bagnall et al., 2018) | 2 | 268 | 896 | 6 |
| Human Activity | RacketSports (Bagnall et al., 2018)
Handwriting (Bagnall et al., 2018)
Epilepsy (Bagnall et al., 2018) | 4
26
4 | 151
150
137 | 30
152
207 | 6
3
3 |
| Sensor | StarLightCurves (Dau et al., 2019) | 3 | 1000 | 1024 | 1 |

Table A.5: Few-shot classification datasets.

| Category | Dataset | $L$ | $C$ |
|---|---|---|---|
| Electricity | ETTm1 (Zhou et al., 2021)
ETTh1 (Zhou et al., 2021)
ECL (Wu et al., 2021) | 96
96
96 | 7
7
321 |
| Weather | Weather (Wu et al., 2021) | 96 | 21 |

Table A.6: Few-shot imputation datasets.

| Category | Dataset | $L$ | $C$ |
|---|---|---|---|
| Machine | SMD (Su et al., 2019)
PSM (Abdulaal et al., 2021) | 96
96 | 38
25 |
| Spacecraft | MSL (Hundman et al., 2018)
SMAP (Hundman et al., 2018) | 96
96 | 55
25 |
| Infrastructure | SWaT (Mathur & Tippenhauer, 2016) | 96 | 51 |

Table A.7: Few-shot anomaly detection datasets.

### A.2.3 ZERO-SHOT LEARNING

For TS forecasting in a zero-shot setting, we evaluate the effectiveness of our proposed method using six datasets. Three of these datasets are used for the zero-shot setting with unseen datasets, while the remaining four datasets are used for the zero-shot setting with new prediction lengths. The statistics for the three unseen datasets are summarized in Table A.8.

| Category | Dataset | Prediction Length | $L$ | $C$ |
|---|---|---|---|---|
| Electricity | Solar (NREL, 2006) | 64 | 128 | 137 |
| Weather | SaugeenRiverFlow (McLeod & Gweon, 2013) | 128 | 256 | 1 |
| Healthcare | Hospital (Hyndman et al., 2008) | 16 | 32 | 767 |

Table A.8: Zero-shot forecasting datasets.

## B IMPLEMENTATION DETAILS

It is important to note that we follow the experimental settings of iTransformer for single-task and UniTS for multi-task settings, respectively. For the implementation, we use the official code repositories of both methods, running the provided scripts without modifications. However, for UniTS in the prompt tuning setting, we encountered an issue where the model failed to converge using the provided script. This was resolved by setting the hidden dimension to $D = 32$, which we applied uniformly across both UniTS and its integration with our method. The following sections outline the specific settings we adhered to.

### B.1 IMPLEMENTATION FOR SINGLE-TASK MODEL: ITRANSFORMER

Following iTransformer (Liu et al., 2024a), we use the Adam optimizer (Kinga et al., 2015) and L2 loss for model optimization. The batch size is consistently set to 32, and the number of training epochs is fixed at 10. Since our approach is plug-and-play, we do not adjust any hyperparameters for our method; instead, we use the same hyperparameters employed by iTransformer.

### B.2 IMPLEMENTATION FOR MULTI-TASK MODEL: UNITS

**Model architecture.** In a multi-task setting, the UniTS network consists of three UniTS blocks, along with one `GEN` tower and one `CLS` tower. For each data source, specific prompt and task tokens are assigned, with forecasting tasks on the same source but with varying forecast lengths using the same prompt and `GEN` token. To enable zero-shot learning on new datasets, a shared prompt and `GEN` token are applied across all data sources. The embedding dimensions are set to 64 for the supervised version, and 32 for the prompt-tuning version, and all blocks in UniTS retain the same feature shape.

**Model training.** In multi-task settings, models are trained jointly on multiple tasks following a unified protocol. To match the largest dataset, samples from each dataset are repeated within each epoch. Supervised training is conducted over 5 epochs with gradient accumulation, yielding an effective batch size of 1024. The initial learning rate is set at 3.2e-2 and is adjusted using a multi-step decay schedule. For self-supervised pretraining, the models training with an are trained for 10 epochs with effective batch size of 4096, starting with a learning rate of 6.4e-3, which is adjusted using a cosine decay schedule.

### B.3 CONSTRUCTION OF CORRELATION MATRIX

For constructing the correlation matrix for CM, we used the datasets corresponding to the training period for forecasting datasets and the training instances for classification datasets. Specifically, for a forecasting dataset with shape $(C, L_{\text{train}} + L_{\text{val}} + L_{\text{test}})$, we compute the correlation matrix with shape $(C, C)$ using only the training period with shape $(C, L_{\text{train}})$. For a classification dataset with shape $(N_{\text{train}} + N_{\text{val}} + N_{\text{test}}, C, L)$, we compute the correlation matrix with shape $(C, C)$ using only the training instances with shape $(N_{\text{train}}, C, L)$ by averaging across the instances.

## C  APPLICATION TO ITRANSFORMER

To demonstrate the effectiveness of our method on a model with a single-task setting, we apply it to the TS forecasting task using iTransformer (Liu et al., 2024a) on 13 datasets, with the results shown in Table C.1.

| Metric | | iTransformer MSE | iTransformer MAE | + CM MSE | + CM MAE |
|---|---|---|---|---|---|
| ETTh1 | 96 | 0.387 | 0.405 | **0.385** | **0.404** |
| | 192 | 0.441 | 0.436 | **0.438** | **0.434** |
| | 336 | 0.491 | 0.462 | **0.475** | **0.454** |
| | 720 | 0.509 | 0.494 | **0.477** | **0.474** |
| | Avg. | 0.457 | 0.449 | **0.444** | **0.441** |
| ETTh2 | 96 | 0.301 | 0.350 | **0.295** | **0.347** |
| | 192 | 0.381 | 0.399 | **0.380** | **0.397** |
| | 336 | **0.423** | **0.432** | 0.427 | 0.434 |
| | 720 | **0.430** | 0.446 | 0.432 | **0.445** |
| | Avg. | 0.384 | 0.407 | **0.383** | **0.406** |
| ETTm1 | 96 | 0.342 | 0.377 | **0.331** | **0.369** |
| | 192 | 0.383 | 0.396 | **0.372** | **0.390** |
| | 336 | 0.418 | 0.418 | **0.412** | **0.414** |
| | 720 | 0.487 | 0.456 | **0.479** | **0.453** |
| | Avg. | 0.408 | 0.412 | **0.398** | **0.406** |
| ETTm2 | 96 | 0.186 | **0.272** | **0.184** | **0.272** |
| | 192 | 0.254 | 0.314 | **0.251** | **0.311** |
| | 336 | 0.317 | 0.353 | **0.312** | **0.350** |
| | 720 | 0.416 | 0.409 | **0.412** | **0.408** |
| | Avg. | 0.293 | 0.337 | **0.289** | **0.335** |
| Exchange | 96 | 0.086 | 0.206 | **0.085** | **0.205** |
| | 192 | 0.181 | 0.303 | **0.180** | **0.302** |
| | 336 | 0.338 | 0.422 | **0.337** | **0.421** |
| | 720 | 0.869 | 0.704 | **0.850** | **0.696** |
| | Avg. | 0.368 | 0.409 | **0.363** | **0.406** |
| Weather | 96 | 0.174 | 0.215 | **0.165** | **0.209** |
| | 192 | 0.224 | 0.258 | **0.213** | **0.251** |
| | 336 | 0.281 | 0.298 | **0.274** | **0.296** |
| | 720 | 0.359 | 0.351 | **0.350** | **0.346** |
| | Avg. | 0.260 | 0.281 | **0.250** | **0.275** |
| Solar | 96 | 0.201 | 0.234 | **0.197** | **0.231** |
| | 192 | 0.238 | 0.263 | **0.232** | **0.260** |
| | 336 | 0.248 | 0.273 | **0.241** | **0.270** |
| | 720 | 0.249 | 0.275 | **0.241** | **0.273** |
| | Avg. | 0.234 | 0.261 | **0.228** | **0.258** |

| Metric | | iTransformer MSE | iTransformer MAE | + CM MSE | + CM MAE |
|---|---|---|---|---|---|
| PEMS03 | 12 | 0.071 | 0.174 | **0.063** | **0.168** |
| | 24 | 0.097 | 0.208 | **0.087** | **0.197** |
| | 48 | 0.161 | 0.272 | **0.133** | **0.250** |
| | 96 | 0.240 | 0.338 | **0.212** | **0.316** |
| | Avg. | 0.142 | 0.248 | **0.124** | **0.231** |
| PEMS04 | 12 | 0.081 | 0.188 | **0.075** | **0.181** |
| | 24 | 0.099 | 0.211 | **0.086** | **0.196** |
| | 48 | 0.133 | 0.246 | **0.108** | **0.222** |
| | 96 | 0.172 | 0.283 | **0.125** | **0.242** |
| | Avg. | 0.121 | 0.232 | **0.098** | **0.210** |
| PEMS07 | 12 | 0.067 | 0.165 | **0.061** | **0.157** |
| | 24 | 0.088 | 0.190 | **0.076** | **0.179** |
| | 48 | 0.113 | 0.218 | **0.086** | **0.188** |
| | 96 | 0.140 | 0.246 | **0.104** | **0.208** |
| | Avg. | 0.102 | 0.205 | **0.082** | **0.183** |
| PEMS08 | 12 | 0.088 | 0.193 | **0.085** | **0.190** |
| | 24 | 0.138 | 0.243 | **0.126** | **0.234** |
| | 48 | 0.334 | 0.353 | **0.178** | **0.241** |
| | 96 | 0.458 | 0.436 | **0.221** | **0.260** |
| | Avg. | 0.254 | 0.306 | **0.152** | **0.231** |
| ECL | 96 | 0.148 | 0.240 | **0.140** | **0.235** |
| | 192 | 0.167 | 0.258 | **0.158** | **0.252** |
| | 336 | 0.179 | 0.272 | **0.172** | **0.267** |
| | 720 | 0.220 | 0.310 | **0.202** | **0.295** |
| | Avg. | 0.179 | 0.270 | **0.168** | **0.262** |
| Traffic | 96 | 0.395 | 0.268 | **0.391** | **0.266** |
| | 192 | 0.417 | 0.277 | **0.409** | **0.275** |
| | 336 | 0.433 | 0.283 | **0.426** | **0.282** |
| | 720 | 0.467 | 0.300 | **0.460** | **0.300** |
| | Avg. | 0.428 | 0.282 | **0.422** | **0.281** |

Table C.1: TS forecasting results with 13 datasets.

## D  APPLICATION TO UNITS

To demonstrate the effectiveness of our method on a TS foundation model, we apply it to four different TS tasks using UniTS (Gao et al., 2024) on datasets from various domains, under multiple settings, including multi-task, few-shot, and zero-shot settings. All experimental settings follow those outlined in UniTS (Gao et al., 2024). The sections and tables outlining the full experiment results are listed in Table D.1.

| Settings | Section | TS downstream tasks | | | |
|---|---|---|---|---|---|
| | | FCST | CLS | IMP | AD |
| Multi-task | D.1 | Table 3 | Table D.2 | - | - |
| Few-shot | D.2 | Table D.3,D.4,D.5 | Table D.6,D.7,D.8 | Table D.9 | Table D.10 |
| Zero-shot | 4.2.3 | Table 3 | - | - | - |

Table D.1: Summary of experiments.

### D.1  MULTI-TASK LEARNING

For experiments under multi-task settings, we perform 20 TS forecasting and 18 classification tasks, where the full results are shown in Table 3 and Table D.2, respectively.

| 18 Tasks | Shared (1 model) | | | | Task-specific (18 models) | | | | | |
|---|---|---|---|---|---|---|---|---|---|---|
| | UniTS + CM | | UniTS | | iTransformer | TimesNet | PatchTST | Pyraformer | Autoformer | GPT4TS |
| | Sup. | PT | Sup. | PT | | | Sup. | | | FT |
| Heartbeat | 67.3 | 70.2 | 59.0 | 69.3 | 66.8 | **72.7** | 65.9 | **72.7** | 71.7 | 69.8 |
| JapaneseVowels | 94.1 | 93.2 | 93.5 | 90.8 | 95.9 | **97.6** | 94.1 | 85.4 | 94.1 | 94.6 |
| PEMS-SF | 83.2 | 82.1 | 83.2 | 85.0 | 83.2 | 77.5 | **83.8** | 83.2 | 79.2 | 79.2 |
| SelfRegulationSCP2 | **58.3** | 51.7 | 47.8 | 53.3 | 48.9 | 52.8 | 48.9 | 56.7 | 45.0 | 45.6 |
| SpokenArabicDigits | 97.1 | 93.5 | 97.5 | 92.0 | 97.8 | **98.7** | 97.5 | 92.1 | 97.3 | 97.5 |
| UWaveGestureLibrary | **84.4** | 83.8 | 79.1 | 75.6 | 82.2 | **84.4** | 81.9 | 72.2 | 42.2 | 81.9 |
| ECG5000 | 93.4 | 93.4 | 92.6 | 93.4 | 93.3 | 92.6 | **94.3** | 91.4 | 91.9 | 93.0 |
| NonInvasiveFetalECGThorax1 | 89.5 | 55.2 | 90.5 | 27.1 | 88.2 | 88.9 | 86.5 | 21.4 | 21.7 | 89.7 |
| Blink | **99.1** | 95.6 | **99.1** | 91.1 | 93.3 | 87.6 | 89.6 | 88.2 | 63.1 | 92.4 |
| FaceDetection | 64.7 | 54.6 | 64.1 | 57.6 | 66.0 | 66.2 | 63.9 | **67.3** | 59.2 | 66.1 |
| ElectricDevices | 62.4 | 60.5 | 60.3 | 55.4 | 57.3 | 49.5 | 59.5 | **65.4** | 56.1 | 62.9 |
| Trace | **99.0** | 93.0 | 91.0 | 82.0 | 79.0 | 91.0 | 77.0 | 74.0 | 60.0 | 96.0 |
| FordB | **76.2** | 64.2 | 76.0 | 62.8 | 72.7 | 68.9 | 61.4 | 55.3 | 66.4 | 77.7 |
| MotionSenseHAR | 92.8 | **94.3** | 92.8 | 93.2 | 93.6 | 90.6 | 75.8 | 88.7 | 30.2 | 96.2 |
| EMOPain | 75.5 | 80.8 | 78.0 | 80.3 | 79.4 | 78.0 | 79.2 | **81.4** | 69.9 | 79.4 |
| Chinatown | 97.7 | 98.0 | 97.7 | 98.0 | 97.4 | 97.7 | 97.7 | 27.4 | 96.8 | 96.5 |
| MelbournePedestrian | 89.3 | 78.3 | 87.3 | 77.0 | 89.3 | **95.7** | 80.4 | 52.3 | 75.0 | 94.0 |
| SharePriceIncrease | 62.9 | 66.6 | 61.9 | **68.4** | 61.9 | 65.0 | 68.0 | 63.1 | 61.5 | 63.7 |
| 1st Count (/18) | 5 | 2 | 2 | 2 | 0 | 5 | 2 | 4 | 0 | - |
| 2nd Count (/18) | 6 | 5 | 3 | 1 | 5 | 2 | 2 | 2 | 1 | - |
| Average Score | **82.0** | 78.3 | 80.6 | 75.1 | 80.3 | 80.9 | 78.1 | 68.8 | 65.6 | 82.0 |

Table D.2: Results of multi-task classification.

## D.2 FEW-SHOT LEARNING

For the few-shot tasks, we conduct four distinct tasks: forecasting (FCST), classification (CLS), imputation (IMP), and anomaly detection (AD), which are discussed in Sections D.2.1, D.2.2, D.2.3, and D.2.4, respectively.

### D.2.1 FEW-SHOT FORECASTING

The results of few-shot forecasting with data ratios of 5%, 15%, and 20% are shown in Tables D.3, D.4, and D.5, respectively.

| 5% | | iTransformer | | UniTS | | | | UniTS + CM | | | |
|---|---|---|---|---|---|---|---|---|---|---|---|
| | | FT | | PT | | FT | | PT | | FT | |
| Data | $H$ | MSE | MAE | MSE | MAE | MSE | MAE | MSE | MAE | MSE | MAE |
| ETTh2 | 96 | 0.554 | 0.500 | **0.405** | **0.417** | 0.418 | 0.424 | 0.421 | 0.427 | 0.421 | 0.425 |
| | 192 | 0.440 | 0.438 | 0.400 | 0.406 | 0.377 | 0.397 | 0.386 | 0.402 | **0.370** | **0.389** |
| | 336 | 0.478 | 0.467 | 0.425 | 0.433 | 0.420 | 0.433 | 0.423 | 0.431 | **0.416** | **0.425** |
| | 720 | 0.483 | 0.480 | 0.446 | 0.457 | 0.439 | 0.452 | **0.424** | 0.444 | 0.428 | **0.443** |
| RiverFlow | 24 | 1.141 | 0.514 | 1.115 | 0.504 | 1.112 | 0.504 | **1.097** | 0.503 | **1.097** | **0.500** |
| ETTm1 | 96 | 0.504 | 0.462 | 0.436 | 0.434 | 0.384 | 0.404 | 0.428 | 0.436 | **0.354** | **0.384** |
| | 192 | 0.555 | 0.485 | 0.462 | 0.448 | 0.414 | 0.418 | 0.475 | 0.458 | **0.393** | **0.405** |
| | 336 | 0.567 | 0.496 | 0.560 | 0.494 | 0.453 | 0.442 | 0.550 | 0.493 | **0.420** | **0.423** |
| | 720 | 0.659 | 0.539 | 0.703 | 0.558 | 0.526 | 0.483 | 0.689 | 0.554 | **0.483** | **0.455** |
| Average | | 0.598 | 0.487 | 0.549 | 0.461 | 0.505 | 0.440 | 0.546 | 0.462 | **0.489** | **0.429** |

Table D.3: Results of few-shot forecasting (5%).

| 15% | | iTransformer | | UniTS | | | | UniTS + CM | | | |
|---|---|---|---|---|---|---|---|---|---|---|---|
| | | FT | | PT | | FT | | PT | | FT | |
| Data | $H$ | MSE | MAE | MSE | MAE | MSE | MAE | MSE | MAE | MSE | MAE |
| ETTh2 | 96 | 0.441 | 0.440 | 0.403 | 0.412 | **0.399** | **0.409** | 0.416 | 0.423 | 0.403 | 0.411 |
| | 192 | 0.398 | 0.410 | 0.396 | 0.404 | 0.394 | 0.399 | 0.388 | 0.403 | **0.387** | **0.399** |
| | 336 | 0.436 | 0.441 | 0.432 | 0.435 | 0.441 | 0.435 | **0.419** | 0.435 | 0.430 | **0.431** |
| | 720 | 0.438 | 0.453 | 0.448 | 0.457 | 0.449 | 0.453 | **0.415** | **0.442** | 0.433 | 0.446 |
| RiverFlow | 24 | **1.067** | **0.467** | 1.077 | 0.492 | 1.069 | 0.489 | 1.073 | 0.492 | 1.072 | 0.487 |
| ETTm1 | 96 | 0.423 | 0.419 | 0.407 | 0.420 | 0.353 | 0.386 | 0.408 | 0.426 | **0.342** | **0.380** |
| | 192 | 0.464 | 0.439 | 0.434 | 0.432 | 0.384 | 0.400 | 0.449 | 0.447 | **0.377** | **0.399** |
| | 336 | 0.492 | 0.457 | 0.490 | 0.464 | 0.416 | 0.420 | 0.502 | 0.475 | **0.406** | **0.148** |
| | 720 | 0.558 | 0.493 | 0.641 | 0.537 | 0.480 | 0.455 | 0.621 | 0.530 | **0.470** | **0.451** |
| Average | | 0.524 | 0.450 | 0.525 | 0.450 | 0.487 | 0.428 | 0.522 | 0.452 | **0.481** | **0.425** |

Table D.4: Results of few-shot forecasting (15%).

| 20% | | iTransformer | | UniTS | | | | UniTS + CM | | | |
|---|---|---|---|---|---|---|---|---|---|---|---|
| | | FT | | PT | | FT | | PT | | FT | |
| Data | $H$ | MSE | MAE | MSE | MAE | MSE | MAE | MSE | MAE | MSE | MAE |
| ETTh2 | 96 | 0.418 | 0.426 | 0.411 | 0.414 | **0.391** | **0.405** | 0.411 | 0.422 | 0.395 | 0.409 |
| | 192 | 0.395 | 0.407 | 0.383 | **0.398** | 0.395 | 0.403 | **0.381** | 0.400 | 0.390 | 0.400 |
| | 336 | 0.431 | 0.438 | **0.419** | 0.431 | 0.430 | **0.430** | 0.423 | 0.430 | 0.438 | 0.433 |
| | 720 | 0.431 | 0.449 | 0.440 | 0.453 | 0.444 | 0.449 | **0.418** | **0.422** | 0.456 | 0.456 |
| RiverFlow | 24 | **1.056** | **0.462** | 1.069 | 0.487 | 1.069 | 0.489 | 1.071 | 0.487 | 1.067 | 0.489 |
| ETTm1 | 96 | 0.408 | 0.410 | 0.409 | 0.421 | 0.344 | 0.379 | 0.403 | 0.425 | **0.339** | **0.376** |
| | 192 | 0.444 | 0.428 | 0.443 | 0.439 | 0.377 | 0.397 | 0.450 | 0.450 | **0.375** | **0.396** |
| | 336 | 0.471 | 0.445 | 0.505 | 0.472 | 0.408 | 0.418 | 0.507 | 0.481 | **0.403** | **0.415** |
| | 720 | 0.536 | 0.482 | 0.648 | 0.536 | 0.472 | 0.453 | 0.621 | 0.531 | **0.466** | **0.448** |
| Average | | 0.510 | 0.438 | 0.525 | 0.450 | 0.486 | 0.425 | 0.521 | 0.453 | **0.482** | **0.425** |

Table D.5: Results of few-shot forecasting (20%).

### D.2.2 FEW-SHOT CLASSIFICATION

The results of few-shot classification with data ratios of 5%, 15%, and 20% are shown in Tables D.6, D.7, and D.8, respectively.

| 5% | iTransformer | UniTS | | UniTS + CM | |
|---|---|---|---|---|---|
| | FT | PT | FT | PT | FT |
| ECG200 | 78.0 | 67.0 | 77.0 | **80.0** | 77.0 |
| Handwriting | 5.4 | 4.6 | 4.7 | 4.8 | **5.5** |
| SelfRegulationSCP1 | 62.8 | 66.2 | 74.7 | **77.8** | 73.7 |
| RacketSports | 37.5 | 31.6 | 35.5 | 39.5 | **47.4** |
| Epilepsy | 39.9 | 44.9 | 47.1 | 44.9 | **57.2** |
| StarLightCurves | 85.1 | 82.3 | 83.8 | **86.3** | 85.4 |
| Average | 51.4 | 49.4 | 53.8 | **54.9** | 54.8 |

Table D.6: Results of few-shot classification (5%).

| 15% | iTransformer | UniTS | | UniTS + CM | |
|---|---|---|---|---|---|
| | FT | PT | FT | PT | FT |
| ECG200 | 81.0 | 74.0 | 78.0 | 73.2 | **82.0** |
| Handwriting | **9.8** | 7.3 | 8.1 | 9.2 | 8.5 |
| SelfRegulationSCP1 | 67.9 | 59.0 | **76.5** | 69.3 | 68.6 |
| RacketSports | **54.6** | 40.1 | 50.7 | 44.7 | 51.3 |
| Epilepsy | 41.3 | 52.9 | 58.0 | 61.6 | **68.1** |
| StarLightCurves | 84.2 | 85.8 | **87.1** | 85.9 | 85.5 |
| Average | 56.5 | 53.2 | 59.7 | 55.4 | **60.4** |

Table D.7: Results of few-shot classification (15%).

| 20% | iTransformer | UniTS | | UniTS + CM | |
|---|---|---|---|---|---|
| | FT | PT | FT | PT | FT |
| ECG200 | 81.0 | 76.0 | 77.0 | **85.0** | 82.0 |
| Handwriting | **11.8** | 8.0 | 8.5 | 7.6 | 9.8 |
| SelfRegulationSCP1 | 77.1 | 68.6 | 70.6 | **77.8** | 74.4 |
| RacketSports | 54.6 | 51.3 | **57.9** | 38.8 | 50.7 |
| Epilepsy | 62.3 | 81.9 | 72.5 | **84.1** | 61.6 |
| StarLightCurves | 84.8 | 87.3 | 86.0 | **90.0** | 87.8 |
| Average | 59.9 | 58.9 | 63.6 | 60.0 | **64.8** |

Table D.8: Results of few-shot classification (20%).

### D.2.3  FEW-SHOT IMPUTATION

The results of few-shot imputation with data ratios of 25% and 50% are shown in Table D.9

| Ratio | | | ECL | ETTh1 | ETTh2 | ETTm1 | ETTm2 | Weather | Avg. |
|---|---|---|---|---|---|---|---|---|---|
| | TimesNet | | 0.245 | 0.369 | 0.193 | 0.442 | 0.119 | 0.106 | 0.246 |
| | PatchTST | FT | 0.195 | 0.315 | 0.147 | 0.309 | 0.092 | 0.089 | 0.191 |
| | iTransformer | | 0.174 | 0.301 | 0.185 | 0.254 | 0.113 | 0.087 | 0.186 |
| 25% | UniTS | PT | 0.139 | 0.311 | 0.178 | 0.268 | 0.102 | 0.078 | 0.179 |
| | | FT | 0.160 | 0.284 | 0.150 | 0.241 | **0.090** | 0.077 | 0.167 |
| | UniTS + CM | PT | 0.139 | 0.310 | 0.176 | 0.262 | 0.100 | 0.078 | 0.179 |
| | | FT | **0.129** | **0.275** | **0.149** | **0.231** | **0.090** | **0.073** | **0.158** |
| | TimesNet | | 0.258 | 0.412 | 0.211 | 0.607 | 0.140 | 0.125 | 0.292 |
| | PatchTST | FT | 0.230 | 0.353 | 0.175 | 0.442 | **0.111** | 0.105 | 0.236 |
| | iTransformer | | 0.203 | 0.332 | 0.205 | 0.372 | 0.136 | 0.106 | 0.226 |
| 50% | UniTS | PT | 0.172 | 0.352 | 0.251 | 0.380 | 0.134 | 0.103 | 0.232 |
| | | FT | 0.191 | 0.322 | 0.198 | 0.352 | 0.118 | 0.095 | 0.213 |
| | UniTS + CM | PT | 0.162 | 0.353 | 0.240 | 0.370 | 0.128 | 0.097 | 0.225 |
| | | FT | **0.151** | **0.307** | **0.197** | **0.345** | 0.116 | **0.093** | **0.201** |

Table D.9: Results of few-shot imputation.

### D.2.4  FEW-SHOT ANOMALY DETECTION

The results of few-shot anomaly detection with data ratio of 5% are shown in Table D.10.

| | | MSL | PSM | SMAP | SMD | SWAT | Avg. |
|---|---|---|---|---|---|---|---|
| Anomaly Trans. | - | 78.0 | 90.2 | 68.3 | 77.8 | 81.5 | 79.2 |
| TimesNet | FT | 33.9 | 91.0 | 68.5 | 84.0 | **93.4** | 74.2 |
| iTransfomer | FT | 80.4 | 96.5 | 67.2 | 82.4 | 89.0 | 83.1 |
| PatchTST | FT | 79.9 | 96.6 | 68.7 | 83.8 | 92.6 | 84.3 |
| UniTS | PT | 73.2 | 95.5 | 65.9 | 81.2 | 92.9 | 81.7 |
| | FT | **81.3** | **97.3** | 71.6 | 85.5 | 92.5 | 85.6 |
| UniTS + CM | PT | 73.7 | 95.5 | 66.0 | 82.0 | 92.9 | 82.0 |
| | FT | **81.3** | **97.3** | **75.9** | **86.2** | 92.6 | **86.6** |

Table D.10: Results of few-shot anomaly detection.

## E  APPLICATION TO TIMESIAM

To demonstrate the effectiveness of our proposed model on TimeSiam (Dong et al., 2024), which uses a self-supervised pretraining framework for TS with Siamese networks, we conduct experiments using iTransformer (Liu et al., 2024a) as the backbone, with two datasets that vary in channel size: Exchange, with a small number of channels (8), and ECL, with a large number of channels (321). Specifically, we apply variants of our method by using the domain parameter only during the fine-tuning stage and during both pretraining and fine-tuning stages. The results, shown in Table E.1, validate both components of our method, with the best performance achieved when using domain parameters at both pretraining and fine-tuning stages.

| | | | TimeSiam | | + CM | | | | | |
|---|---|---|---|---|---|---|---|---|---|---|
| Correlation matrix | | | - | | ✓ | | ✓ | | ✓ | |
| Domain parameters | Pretrain | | - | | - | | - | | ✓ | |
| | Fine-tune | | - | | - | | ✓ | | ✓ | |
| Dataset | $H$ | | MSE | MAE | MSE | MAE | MSE | MAE | MSE | MAE |
| Exchange $(C = 8)$ | 96 | | 0.092 | 0.215 | 0.089 | **0.207** | **0.088** | **0.207** | **0.088** | 0.209 |
| | 192 | | **0.182** | 0.306 | **0.182** | 0.304 | **0.182** | 0.303 | **0.182** | 0.305 |
| | 336 | | 0.341 | 0.426 | 0.336 | 0.422 | 0.332 | 0.417 | 0.329 | 0.417 |
| | 720 | | 0.806 | 0.679 | 0.792 | 0.670 | 0.788 | 0.668 | 0.783 | 0.666 |
| | Avg. | | 0.356 | 0.407 | 0.350 | 0.401 | 0.349 | 0.399 | 0.346 | 0.398 |
| ECL $(C = 321)$ | 96 | | 0.147 | 0.239 | **0.140** | **0.236** | **0.140** | **0.236** | 0.141 | 0.237 |
| | 192 | | 0.162 | 0.253 | **0.157** | 0.251 | **0.157** | 0.251 | 0.157 | 0.250 |
| | 336 | | 0.175 | 0.269 | 0.173 | 0.268 | 0.173 | 0.268 | 0.172 | 0.267 |
| | 720 | | 0.215 | 0.304 | **0.203** | 0.297 | **0.203** | 0.297 | 0.203 | 0.296 |
| | Avg. | | 0.175 | 0.266 | **0.168** | 0.263 | **0.168** | 0.263 | 0.168 | 0.262 |

Table E.1: Results of TS forecasting with TimeSiam.

## F  LOOKBACK WINDOW SIZE VS. PERFORMANCE

Following the previous work (Liu et al., 2024a), we conduct an experiment to evaluate the effect of varying the lookback window size ($L$) on performance, using three datasets: ECL (Wu et al., 2021), Traffic (Wu et al., 2021), and PEMS03 (Liu et al., 2022) with iTransformer (Liu et al., 2024a) as the backbone. The results, shown in Figure F.1, indicate that the effectivness of CM remains robust to the choice of $L$ for all three datasets.

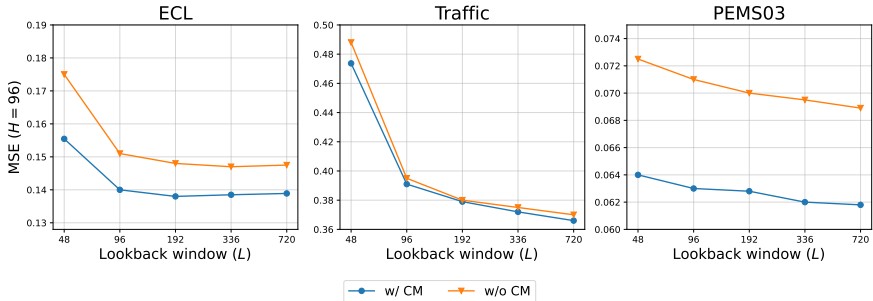

Figure F.1: **Effect of CM under various lookback window sizes.** Forecasting performance with the lookback length $L \in \{48, 96, 192, 336, 720\}$, with forecast horizon $H = 12$ for PEMS03 and $H = 96$ for other datasets.

## G CM UNDER EXTREME CASES

To evaluate the effectiveness of CM under extreme cases, we design a scenario where the channels in TS exhibit no correlation. Specifically, we generate a synthetic TS dataset with two channels using sine waves oscillating at frequencies of 0.5 and 2.0 over a length of 18,000 (similar to ETTh (Zhou et al., 2021)), as shown in Figure G.1. We conduct TS forecasting using this dataset with iTransformer (Liu et al., 2024a) as the backbone, with an input window size and forecasting horizon of 96, following the experimental protocol used in ETTh1. The result yields a CD ratio of CM approximately 0.018 and a forecasting MSE of around 0.0014, confirming strong channel independence and demonstrating the effectiveness of our method even under extreme CI conditions.

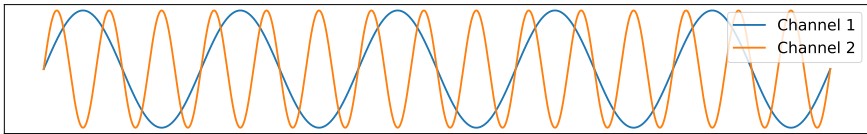

Figure G.1: Synthetic dataset with two uncorrelated channels

## H MASKED CHANNEL PREDICTION

Tables H.1 and H.2 show the results of masked channel prediction for five datasets (Wu et al., 2021; Liu et al., 2022), indicating significant improvement when a CM is applied to iTransformer compared to when it is not used.

| Horizon | Exchange Avg. MSE(C1~C8) | | | ECL Avg. MSE(C1~C321) | | |
|---|---|---|---|---|---|---|
| | iTrans. | + CM | Impr. | iTrans. | + CM | Impr. |
| 96 | 0.139 | **0.138** | **1.2%** | 0.846 | **0.526** | **37.8%** |
| 192 | 0.236 | **0.232** | **1.5%** | 0.849 | **0.563** | **33.7%** |
| 336 | 0.383 | **0.374** | **2.4%** | 0.861 | **0.594** | **31.0%** |
| 720 | 0.934 | **0.917** | **1.8%** | 0.891 | **0.741** | **16.8%** |
| Avg. | 0.423 | **0.415** | **1.8%** | 0.862 | **0.606** | **29.7%** |

Table H.1: Results of masked channel prediction (Exchange, ECL).

| Horizon | PEMS04 Avg. MSE(C1~C307) | | | PEMS07 Avg. MSE(C1~C883) | | | PEMS08 Avg. MSE(C1~C170) | | |
|---|---|---|---|---|---|---|---|---|---|
| | iTrans. | + CM | Impr. | iTrans. | + CM | Impr. | iTrans. | + CM | Impr. |
| 12 | 0.549 | **0.300** | **45.4%** | 0.835 | **0.343** | **58.9%** | 0.628 | **0.200** | **68.1%** |
| 24 | 0.718 | **0.351** | **51.1%** | 0.865 | **0.448** | **48.1%** | 0.678 | **0.241** | **64.5%** |
| 48 | 0.750 | **0.409** | **45.5%** | 1.038 | **0.511** | **50.8%** | 1.197 | **1.059** | **11.5%** |
| 96 | 0.758 | **0.513** | **32.3%** | 1.040 | **0.640** | **38.5%** | 1.375 | **1.217** | **11.5%** |
| Avg. | 0.694 | **0.393** | **43.3%** | 0.945 | **0.486** | **48.6%** | 0.970 | **0.679** | **29.9%** |

Table H.2: Results of masked channel prediction (PEMS datasets).

