# OpenReview forum: "Partial Channel Dependence with Channel Masks for Time Series Foundation Models"
_ICLR.cc/2025/Conference — Submitted to ICLR 2025_

### Official Review · Reviewer_JKMr · 2024-11-04

**Soundness:** 2
**Presentation:** 3
**Contribution:** 2
**Rating:** 5
**Confidence:** 4

**Summary:**

This work proposes a time-series foundation model designed to capture varying channel dependencies across different datasets. By using a correlation matrix as a prior, the model incorporates a learnable domain parameter to construct a domain-specific channel mask, effectively capturing global channel dependencies. In conjunction with local channel dependency masks (as introduced in prior works, such as iTransformer), the proposed model accommodates diverse channel dependencies across datasets, leading to performance improvements in tasks such as forecasting, imputation, and anomaly detection (in full-shot, few-shot, and zero-shot settings).

**Strengths:**

- The paper presents a straightforward algorithm with clear explanations.
- The manuscript covers a broad range of experiments across various time-series tasks (anomaly detection, forecasting, imputation, etc.) in different settings (supervised, few-shot/zero-shot domain/task) with extensive analyses (e.g., robustness to missing values) that support its claimed advantages.
- The newly introduced experiment, "masked channel prediction," is particularly novel and promising, though it is only briefly discussed in the manuscript.

**Weaknesses:**

Despite the demonstrated performance gains across multiple tasks and settings, some of the motivation and design choices remain insufficiently addressed in the manuscript. Additionally, certain reported experimental results differ from what appears in the paper, and key experimental details are not fully explained.

First, I would appreciate clarification from the authors regarding the principles underlying channel dependency in the proposed time-series foundation model:

- Why is it crucial to account for varying channel dependencies? To what extent is this heterogeneity in channel dependency necessary for constructing a robust foundation model for time-series data? Prior research has highlighted its importance for forecasting tasks both empirically [1][2] and theoretically [3], yet the rationale for its role in the foundation model is not fully studied. What is the anticipated impact of the time-series foundation model, and how does it address channel dependencies? (Simply stating "inherent heterogeneity" may be insufficient in this context.)
- The global mask encapsulates correlation across various domains during training, which is then reused during testing without further adjustment. Is it sufficient to rely on the global correlation matrix alone? Since local correlations may vary over time, the correlation matrix at test time might differ significantly from that during training. How do you ensure its stability under these conditions?
- Is correlation an appropriate metric for capturing channel dependencies? Some studies in forecasting suggest a "causal relationship" between channels, while others tackle spurious correlations (where channels appear correlated but are not causally related).

Furthermore, several design choices would benefit from clarification:

- What motivated the choice to use domain-specific global attention while sharing local attention across multiple domains? What specific roles do global and local attention play in the model's functioning?
- Is the "pair-wise dot product" between global and local attention sufficient to achieve the intended effect? Under extreme test-time scenarios where variables v1 and v2 exhibit low global correlation (approaching zero) but high local correlation, this design might yield low attention scores, potentially failing to capture abrupt correlation increases under certain test-time conditions.

In addition, some minor experimental inconsistencies and essential experimental details require clarification:

- Some scores of baselines appear lower than what has been reported in the original paper. In Table 3, the MSE/MAE score on iTransformer is way higher than it has been reported (Appendix F, Table 8 in [4]) Have these results been reproduced, and if so, what could account for the discrepancies?
- How is the global correlation matrix defined? Does it only consider the correlation matrix over the training period?
- In zero-shot experiments on new datasets, how is the domain parameter handled? As the domain parameter cannot be directly learned for unseen domains, is it substituted with that of a similar domain?

[1] Rethinking Channel Dependence for Multivariate Time Series Forecasting: Learning from Leading Indicators (ICLR 2024)

[2] Tiny Time Mixers (TTMs): Fast Pre-trained Models for Enhanced Zero/Few-Shot Forecasting of Multivariate Time Series (arXiv 2024)

[3] Time-Series Forecasting for Out-of-Distribution Generalization Using Invariant Learning (ICML 2024)

[4] ITRANSFORMER: INVERTED TRANSFORMERS ARE EFFECTIVE FOR TIME SERIES FORECASTING (ICLR 2024)

**Questions:**

In conclusion, while the paper presents an extensive array of experimental evidence, the motivation for addressing channel dependency heterogeneity is not entirely convincing in terms of its necessity for building a foundational model. Although channel dependencies may be relevant to specific tasks in the time-series domain (e.g., forecasting), the underlying rationale and its influence on design choices are not fully explained in the manuscript. Additionally, certain key experimental results appear inconsistent, which is problematic given the paper's emphasis on empirical performance gains over theoretical justification. Nevertheless, the proposed architecture is straightforward and appears tailored to address varying channel dependencies, and it has the potential to be impactful if the dependencies are better elucidated and some of the reported results are clarified.

If the authors address these points and resolve potential experimental issues in the main results table, I would be inclined to raise my rating.

---

> ### Author Response · Authors · 2024-11-19
>
> ## **W1.  Principles underlying channel dependency (CD)**
> To better clarify (1) the concept of implicit/explicit heterogeneity and (2) the motivation for addressing CDs varying across TS datasets, we have revised the **1. Introduction** section in the **revised manuscript**. We kindly invite you to reread the revised **1. Introduction** section to facilitate the below discussions.
>
>
> &nbsp;
>
>
> ### **W1-1. Necessity of accounting for varying CD between datasets**
>
> In a conventional setting where a single model is trained on a single dataset (i.e., **non-TSFM**), varying CDs across datasets may not pose a significant issue; we can choose either CI or CD model to perform effectively on that particular dataset.
>
>
> *However*, the challenge arises when training a **TSFM**, which is a ***single*** model designed to generalize across ***multiple*** datasets with varying CDs. Because ***each dataset may benefit from a different approach (CI or CD)***, it is crucial for a TSFM to account for varying CD across datasets, as illustrated in **Figure 1** and noted in **L43--46**.
>
>
> Despite this, previous TSFMs have mainly focused on accommodating **explicit** heterogeneity, such as difference in dataset shape and size, while overlooking the **implicit** heterogeneity stemming from varying CD, as noted in **L47--50**. This oversight motivated us to propose a **PCD framework** that ***adjusts the CDs of various datasets (captured by a single model)*** based on  the **unique characteristics of each dataset**  to handle the  **implicit heterogeneity**. To achieve PCD, we propose a channel mask (CM) constructed  **for each dataset**, which is then element-wise multiplied with the CD captured by a model (attention matrix) for adjustment, with the visualization of CM of ETTh1 dataset shown in the **second panel of Figure 5**.
>
>
>
>
> &nbsp;
>
>
> ### **W1-2. Global CM under distribution shift (at test time)**
>
>
> > **W1-2-a) Intention of CM**
>
>
> By ***"global"***, we refer to features that are **”fixed across time points”** , capturing stable characteristics rather than **”adapting to local changes”**. ***Adjusting the (global) CM based on test-time local changes would thus conflict with the intended purpose***, reducing their ability to encapsulate stable characteristics.
> To address ***local*** variations, we rely on the **”attention matrix”**, which dynamically adapts to (local) input TS. This approach ensures that while the (global) CM provides a stable foundation, the (local) attention matrix directly addresses temporal and local shifts.
> Additionally, we confirmed that (global) CM and the (local) attention matrix effectively **complement each other** on 13 real-world datasets, which are known to have **significant distribution shifts** [A], as shown in **Table 12**. The results indicate that the proposed method effectively captures both stability and adaptability under varying conditions.
>
>
> &nbsp;
>
>
> > **W1-2-b) Stability of (global) CM**
>
>
> To address the reviewer’s concern regarding the **stability of the (global) CM** when local fluctuations occur, we conducted an additional experiment using the Weather dataset with iTransformer: Specifically, we considered two intervals ((0%,80%) and (0%,100%)), and trained the model separately on each interval. The L2 distance between the two trained CMs was approximately **0.0002**, indicating minimal difference, given that two random matrices with the same range (0,1) and shape yield an average distance of approximately **0.02**.
> Regarding the **stability of “performance”** under distribution shifts, we provide a more detailed discussion in **W2-1** below.
>
>
> &nbsp;
>
>
> [A] Han, Lu, Han-Jia Ye, and De-Chuan Zhan. "The capacity and robustness trade-off: Revisiting the channel independent strategy for multivariate time series forecasting."  TKDE (2024)

---

> > ### Author Response · Authors · 2024-11-19
> >
> > ### **W1-3. Is correlation an appropriate metric for capturing CD?**
> >
> >
> >
> >
> > **a) Emphasis on utilizing dataset-specific information (rather than “correlation” itself)**
> >
> >
> > We appreciate the reviewer’s concern regarding the **suitability of correlation** as a metric for measuring channel relationships. However, our primary focus is not on using the "correlation" metric, but on ***leveraging the "dataset-specific information"***, as highlighted in **L16,60,480**.
> >
> >
> > &nbsp;
> >
> > **b) Metrics other than correlation**
> >
> >
> > While our primary focus is on utilizing **dataset-specific information** rather than the correlation metric itself, we acknowledge the reviewer’s concern regarding the choice of metrics for capturing channel relationships. As detailed in **Table 10**,  we have ***already*** conducted an analysis using **four different metrics** - Correlation, Euclidean, Cosine, Dynamic Time Warping (DTW) - demonstrating consistent performance improvements across metrics.
> >
> >
> > &nbsp;
> >
> >
> > The reviewer also raised the possibility of using a metric that captures **causal relationships**. We agree that causal metrics, such as Granger causality, can be effective in certain contexts, particularly forecasting. However, these metrics **may not generalize well to other tasks**, such as classification or anomaly detection, where causal relationships are less critical. Additionally, Granger causality’s computational complexity rendered it impractical for our application, particularly with large-scale datasets used in TSFM training. Instead, we opted for DTW, which effectively captures lagged dependencies between channels and provides a computationally feasible alternative.
> >
> >
> > If the reviewer has additional recommendations for metrics that might enhance our approach, we would be glad to consider incorporating them into our work.
> >
> >
> > &nbsp;
> >
> > ---
> >
> > ## **W2. Design choices**
> >
> >
> >
> > ### **W2-1. “Global” CM & “Local” attention matrix**
> > - **[Global CD]** The primary role of **CM** is to capture the ***varying CD between datasets***, rather than changes *within a dataset* across multiple time steps. Consequently, using the CM to capture **global CD** -- defined as CD shared across time steps (**L178--179**) -- is a natural and effective design choice.
> > - **[Local CD]** At the same time, we recognize that **CD can change over time** within a dataset, making it essential to capture **local CD**. We address this through the **attention matrix**, which is constructed dynamically **based on the input TS**, enabling it to adapt to temporal variations.
> >
> >
> > &nbsp;
> >
> >
> > We believe these two components -- global and local CD -- work synergistically. As shown in **Table 12**, capturing both global and local CD achieves the best results across 13 datasets, which are known to have **significant distribution shifts** [A].
> >
> >
> > Designing a local attention mechanism specifically tailored to multiple domains is an intriguing direction for future research. However, this is beyond the scope of our current work, as it would require developing a new architecture for TSFMs.
> >
> >
> > &nbsp;
> >
> >
> > [A] Han, Lu, Han-Jia Ye, and De-Chuan Zhan. "The capacity and robustness trade-off: Revisiting the channel independent strategy for multivariate time series forecasting."  TKDE (2024)
> >
> >
> > &nbsp;
> >
> >
> > ### **W2-2. Element-wise multiplication of (global) CM and (local) attention matrix**
> >
> >
> > As the reviewer noted, under extreme test-time scenarios where two channels exhibit **low** global correlation but high local correlation, the element-wise multiplication between the CM and the local matrix would indeed result in **low** attention scores.
> >
> >
> > However, we believe this is a ***desirable situation*** of using the CM. In such extreme cases with **no global correlation**, high local correlation is more likely to reflect **noise or an abnormal status** rather than a meaningful relationship. In such instances, it is appropriate that the channels  **do NOT influence each other**.

---

> > > ### Author Response · Authors · 2024-11-19
> > >
> > > ## **W3. Minor issues & clarifications**
> > >
> > >
> > >
> > >
> > > ### **W3-1. Inconsistent results with the original paper**
> > >
> > >
> > > We acknowledge the inconsistency between the results reported in the original paper and our paper. However, we reproduced the results using the **official code repositories** of iTransformer [A] and UniTS [B] for our experiments, running the provided scripts **without any modifications**.
> > >
> > >
> > > Unfortunately, this reproduction yielded **inconsistent results** for both models compared to the original paper, and we found that this inconsistency is shared with others, according to GitHub issues in both the iTransformer and UniTS repositories.
> > > - Specifically, for iTransformer, we found a similar issue reported in a Chinese GitHub discussion, which we translated to confirm its relevance.
> > > - For UniTS under the prompt tuning setting, we encountered an issue where the model failed to converge using the provided script and the problem is also reported by others in a GitHub issue. This issue was resolved by setting the hidden dimension to $D=32$, which yielded degraded performance compared to the original paper. We applied this setting uniformly across both UniTS and its application to our method for fair comparison, as detailed in **Appendix B.2**.
> > >
> > >
> > > &nbsp;
> > >
> > >
> > > Nonetheless, following the reviewer’s feedback,  we have added further reproduction details in **Appendix B** to avoid any confusion.
> > >
> > >
> > > &nbsp;
> > >
> > >
> > > [A] Liu, Yong, et al. "iTransformer: Inverted transformers are effective for time series forecasting." ICLR (2024)
> > >
> > >
> > > [B] Gao, Shanghua, et al. "Units: Building a unified time series model." NeurIPS (2024)
> > >
> > >
> > > &nbsp;
> > >
> > >
> > > ### **W3-2. Definition of (global) correlation matrix in L176**
> > >
> > >
> > > As the reviewer mentioned, the (global) correlation matrix is calculated solely based on the time steps during the **TRAINING period** for forecasting datasets, and the **TRAINING instances** for classification datasets, as follows:
> > > - For a forecasting dataset ($C, L_{\text{train}} + L_{\text{val}} + L_{\text{test}}$), we compute the correlation matrix ($C, C$) using only the training period ($C, L_{\text{train}}$).
> > > - For a classification dataset ($N_{\text{train}} + N_{\text{val}} + N_{\text{test}}, C, L$), we computed the correlation matrix ($C, C$) using only the training instances ($N_{\text{train}}, C, L$), by averaging across these instances.
> > >
> > >
> > > We have included the relevant information in **Appendix B.3**.
> > >
> > >
> > > &nbsp;
> > >
> > >
> > > ### **W3-3. Domain parameters for unseen datasets**
> > >
> > >
> > > This issue has ***already*** been addressed in **L367--371** and **Table 9**, where we provide various candidates for selecting the parameters for these datasets, demonstrating robustness to these choices.

---

> > > > ### Author Response · Authors · 2024-11-19
> > > >
> > > > While we believe we addressed all your concerns in our rebuttal and the revised manuscript, please feel free to ask us if we may have missed anything or you have additional questions; we are happy to address them.

---

> ### Comment · Reviewer_JKMr · 2024-11-26
>
> Thank you for your clarification. While your response addresses certain concerns (e.g., differing reproduction results, though I still believe further clarification is warranted—though I will not contest it given that UniTS, a peer-reviewed paper, reports a similar experimental outcome for iTransformer), it does not fully resolve my earlier concern regarding global and local CD. You have explained that CM is designed for global CD, while local CD is addressed through channel-wise attention. If this is indeed the case, I require further elaboration on how attention mechanisms alone are sufficient to effectively capture local, time-varying CD without the incorporation of additional data. If your assertion holds, iTransformer should demonstrate the most significant performance gains in ablation studies conducted on datasets exhibiting the largest degree of time-varying CD (such as Exchange and Solar-Energy); however, this does not appear to be the case. Furthermore, I find it unclear how the proposed attention mechanism for local CD can be consistently applied across different domains.
>
> Given that some of my major concerns remain unaddressed despite your clarification, I maintain my current score as it is.

---

> ### Author Response · Authors · 2024-11-27
>
> Thank you for engaging in the discussion!
>
> Below we provide further clarification, hopefully it addresses misunderstanding on our work:
>
> &nbsp;
>
> ## Concern regarding **global** and **local** CD
>
> First of all, regarding your question "how the proposed attention mechanism for local CD ~", ***we do not specifically propose a method to capture local CD***; it is naturally captured by the **conventional attention mechanism** applied across channels in Transformers, and this contrasts with global CDs captured by our proposed channel masks (CMs). In other words, our contribution lies in introducing **CMs capturing global CD**, which complements local CDs captured by the attention mechanism.
>
> &nbsp;
>
> To further clarify the type of CDs captured by the attention mechanism and CMs, we provide an example below:
>
> Given a TS with 7 channels, a length of 5000, and a lookback window of 96:
>
> - **CM** is constructed with the **entire TS** of shape **$(7, 5000)$** to capture the (**global**) dependency of channels over the **entire time period**.
> - An **attention matrix** is constructed with the **input TS** segment of shape **$(7, 96)$** to capture the (**local**) dependency of channels within the **lookback window**.
>
>
> Given this example, we merely name (1) the CD captured by the CM as ***global CD***, as the CM is constructed from the (global) **entire TS $(7, 5000)$**, and (2) the CD captured by the attention matrix as ***local CD***, as the attention matrix is constructed from the (local) **input TS $(7, 96)$**, as noted in **L178--180**.
>
> &nbsp;
>
> Based on the above clarification, we answer your questions below:
>
> ### ***"Q1. how attention mechanisms alone are sufficient to effectively capture local, time-varying CD"***
>
> The conventional attention mechanism used in Transformers captures dependencies between tokens, and inspired by prior works such as iTransformer [A], each channel is treated as a token in our setting. Hence, the model captures the ***local CD*** within the input TS segment, where its effectiveness has already been demonstrated in previous studies [A,B,C,D,E].
>
> &nbsp;
>
> ### ***Q2."iTransformer should demonstrate the most significant performance gains ~ on datasets exhibiting the largest degree of time-varying CD (such as Exchange and Solar-Energy); however, this does not appear to be the case."***
>
> First of all, we kindly ask you to share a supporting reference on your claim *"Exchange and Solar-Energy are datasets exhibiting the largest degree of time-varying CD"*, as we believe this will be essential to strengthen our analysis on the effectiveness of our method.
>
> In contrast to your expectation, we argue that if datasets exhibit the largest degree of time-varying (or local) CD, then the performance gain by capturing global CD would be minimal, as iTransformer already captures the essential (local) CDs for such datasets.
>
> In general, it is hard to anticipate a relatively larger gain on a specific dataset than others, as numerous factors contribute to the final performance. For example, the fixed size of the lookback window we used in an experiment could be large enough to capture global CDs in some datasets, such that the gain by our CMs is relatively small regardless of the degree of local CDs.
>
> We emphasize that **Table 12** shows an ablation study ***highlighting the effectiveness of our method***. Specifically, **global CD only** outperforms **local CD only** on both Exchange and Solar-Energy, demonstrating the importance of capturing global CD through our CM. Of course, they have complementary information, such that using both results in the best result across all datasets.
>
> &nbsp;
>
> ### ***Q3."how the proposed attention mechanism for local CD can be consistently applied across different domains.***
>
> Again, the attention mechanism for capturing local CD is essentially the same as those in the conventional Transformers, and its effectiveness has already been demonstrated in previous studies [A,B,C,D,E].
> Furthermore, as the attention mechanism has **no domain-specific parameters**, and we believe there is **no specific issue** on its applicability across different domains [B].
> Introducing domain-specific attention mechanisms would be an interesting idea, but it involves lots of additional learnable parameters; as such, we leave this exploration for future works.
>
> &nbsp;
>
> [A] Liu et al. "iTransformer: Inverted transformers are effective for time series forecasting." ICLR (2024)
>
> [B] Gao et al. "Units: Building a unified time series model." NeurIPS (2024)
>
> [C] Ilbert et al. “SAMformer: Unlocking the Potential of Transformers in Time Series Forecasting with Sharpness-Aware Minimization and Channel-Wise Attention.” NeurIPS (2024)
>
> [D] Wang et al. “CARD: Channel aligned robust blend transformer for time series forecasting.” ICLR (2024)
>
> [E] Yu et al. “Revitalizing Multivariate Time Series Forecasting: Learnable Decomposition with Inter-Series Dependencies and Intra-Series Variations Modeling.” ICML (2024)

---

### Official Review · Reviewer_saeS · 2024-11-04

**Soundness:** 1
**Presentation:** 2
**Contribution:** 1
**Rating:** 3
**Confidence:** 5

**Summary:**

This work introduces the concept of partial channel dependence (PCD) to address implicit heterogeneity in time series (TS) data. By utilizing a channel mask that incorporates a correlation matrix to encode relative dependencies between channels and domain parameters to learn dataset-specific absolute dependencies, the authors refine the correlation matrix for better channel dependency adjustments. The effectiveness of PCD is validated across four TS tasks—forecasting, classification, imputation, and anomaly detection.

**Strengths:**

1. This work focuses on a highly valuable research direction and highlights the importance of Partial Channel Dependence in Time Series analysis.
2. This work proposes a concise approach to adjust the correlation matrix obtained from prior knowledge.

**Weaknesses:**

1. The concept of Partial Channel Dependence (PCD) is already discussed in the CCM[1] where the authors employ a clustering approach to capture latent PCD. The current content lacks a detailed comparison and discussion with this work.
2. It is important to note that not all foundation models are based on attention mechanisms (TTM [2]), and not all methods that utilize attention mechanisms effectively capture the attention between channels like UniTS (TimesFM[3], Timer[4], MOIRAI[5], MOMENT[6]). As a plugin for foundation models, the generality of the CM method is insufficient.
3. The paper does not specify how the correlation matrix mentioned was constructed, and references [7] and [8] primarily focus on capturing correlation relationships with lag properties. In contrast, this work does not explore lag properties but instead utilizes the complete sequences. The construction method of the correlation matrix needs to be explained in detail.
4. Domain parameters highly correlated with the construction method of the correlation matrix, If use other methods, they may not be effective. The effectiveness of domain parameters in the experiments lacks further validation.
5. In the experiments, there is a lack of comparison with other plugins, such as LIFT[8], and each task only validates the plugin's improvement on a few models, making its generality difficult to confirm. More importantly, a large number of foundation models have not been considered in the experiments, such as [2-6]. The existence of these issues raises serious doubts about the effectiveness and generality of the plugin.

[1] From Similarity to Superiority: Channel Clustering for Time Series Forecasting.

[2] Tiny Time Mixers (TTMs): Fast Pre-trained Models for Enhanced Zero/Few-Shot Forecasting of Multivariate Time Series.

[3] A decoder-only foundation model for time-series forecasting.

[4] Timer: Transformers for Time Series Analysis at Scale.

[5] Unified Training of Universal Time Series Forecasting Transformers.

[6] MOMENT: A Family of Open Time-series Foundation Models.

[7] Vcformer: Variable correlation transformer with in-herent lagged correlation for multivariate time series forecasting.

[8] Rethinking channel dependence for multivariate time series  forecasting:Learning from leading indicators.

**Questions:**

See weaknesses.

---

> ### Author Response · Authors · 2024-11-19
>
> ## **W1. PCD vs. CCM**
> We believe the proposed PCD framework is ***fundamentally different*** from the channel clustering method in CCM [A], which is concurrent to our work (Note: CCM will appear in NeurIPS 2024, which has not yet been held!).
>
>
> While CCM and our method share a **superficial similarity** in balancing CI and CD, the ***goals and approaches differ significantly***. CCM enhances performance by incorporating channel cluster information to (CI or CD) models, whereas our method adjusts the varying CD across multiple datasets for TSFMs using dataset-specific information (no clustering involved in our method). Prior works with similar objectives [B, C] also combine CI and CD; however, this does not imply that they represent the same algorithm.
>
>
> &nbsp;
>
>
> [A] Chen, Jialin, et al. "From Similarity to Superiority: Channel Clustering for Time Series Forecasting." NeurIPS (2024)
>
>
> [B] Nie, Tong, et al. "Channel-aware low-rank adaptation in time series forecasting." CIKM (2024)
>
> [C] Chi, Ce, et al. "InjectTST: A Transformer Method of Injecting Global Information into Independent Channels for Long Time Series Forecasting." arXiv preprint (2024).
>
>
> &nbsp;
>
>
> ## **W2, W5-2. Lack of generality: The proposed method is only applicable to models based on attention-mechanism that captures CD.**
>
>
> PCD aims to enhance ***TSFMs capturing CDs*** by leveraging dataset-specific characteristics, rather than proposing a plug-in-play method applicable to any type of TSFMs existing in the world, as mentioned in **L60--61** and **L69--70**.  Specifically, CM is designed to adjust the **CD estimated by the model**, making it **applicable to CD models** but **infeasible for CI models**.
>
>
> &nbsp;
>
>
> Among the **five TSFMs** the reviewer mentioned ([2]--[6]), the reasons we did or could not experiment with them--either due to model architecture limitations ([2,3,4,6]), dataset size constraints ([3,4,5,6]), or code availability ([4]). Most importantly, these works are **concurrent with ours** (NeurIPS 2024: [2], ICML 2024: [3--6]), which made it difficult for us to run experiments due to the large dataset size and the timing of the code release. To address these challenges, we applied our method to two other SOTA models, iTransformer [A] (ICLR 2024) and TimeSiam [F] (ICML 2024), both of which are single-task models with released code.
>
>
> &nbsp;
>
>
> Additionally, while we acknowledge that not all TSFMs rely on attention mechanisms and that Transformers are widely employed to capture temporal dependencies (TD), recent methods--particularly with the **emergence of iTransformer** [A]--have increasingly emphasized **leveraging attention mechanisms to capture CD** [B, C, D, E, F]. Our method aligns with this trend and builds upon the strengths of these approaches.
>
>
> &nbsp;
>
>
> Furthermore, the limitation of being applicable only to the attention mechanism for capturing CD has been ***already*** discussed in **L482--484**, where we identified the development of a novel approach to achieve PCD without relying on Transformer-based methods as a promising direction for future work.
>
>
> &nbsp;
>
>
> [A] Liu, Yong, et al. "iTransformer: Inverted transformers are effective for time series forecasting." ICLR (2024)
>
>
> [B] Gao, Shanghua, et al. "Units: Building a unified time series model." NeurIPS (2024)
>
>
> [C] Ilbert, Romain, et al.  “SAMformer: Unlocking the Potential of Transformers in Time Series Forecasting with Sharpness-Aware Minimization and Channel-Wise Attention.” NeurIPS (2024)
>
>
> [D] Wang, Xue, et al. “CARD: Channel aligned robust blend transformer for time series forecasting.” ICLR (2024)
>
>
> [E] Yu, Guoqi, et al.  “Revitalizing Multivariate Time Series Forecasting: Learnable Decomposition with Inter-Series Dependencies and Intra-Series Variations Modeling.” ICML (2024)
>
>
> [F] Dong, et al. "TimeSiam: A Pre-Training Framework for Siamese Time-Series Modeling." ICML (2024)
>
>
> &nbsp;

---

> > ### Author Response · Authors · 2024-11-19
> >
> > ## **W3-1. Construction of correlation matrix.**
> >
> >
> > As the reviewer mentioned, the (global) correlation matrix is calculated solely based on the time steps during the **TRAINING period** for forecasting datasets, and the **TRAINING instances** for classification datasets, as follows:
> > - For a forecasting dataset ($C, L_{\text{train}} + L_{\text{val}} + L_{\text{test}}$), we compute the correlation matrix ($C, C$) using only the training period ($C, L_{\text{train}}$).
> > - For a classification dataset ($N_{\text{train}} + N_{\text{val}} + N_{\text{test}}, C, L$), we computed the correlation matrix ($C, C$) using only the training instances ($N_{\text{train}}, C, L$), by averaging across these instances.
> >
> >
> > We have included the relevant information in **Appendix B.3**.
> >
> >
> > &nbsp;
> >
> >
> > ## **W3-2. Correlation can not capture (1) lag properties and requires (2) complete sequences.**
> >
> >
> >
> >
> > Our focus is not on using the specific use of the "correlation" metric itself, but rather on ***leveraging the "dataset-specific information"***, as highlighted in **L16,60,480**.
> >
> >
> > Nevertheless, we acknowledge the reviewer’s concerns that correlation
> > (1) cannot capture **lag properties**, and
> > (2) cannot handle **incomplete sequences**.
> >
> >
> > &nbsp;
> >
> > However, we would like to clarify that these issues have ***already*** been addressed as follows:
> >
> >
> > - **[1. Lag properties]** As shown in **Table 10**, we have utilized four different metrics to measure channel relationships, including **Dynamic Time Warping (DTW)**, which is capable of handling TS with both lag properties and incomplete sequences, but at the cost of a computational complexity of $O(L^2)$, where $L$ is the length of the time series. However, we have found that correlation provides sufficiently good performance, as shown in **Table 10**.  Additionally, the lag correlation approach in reference [7], which is noted by the reviewer and mentioned in  **L141--142** as our concurrent work, is closer to an attention module. It calculates lagged correlations between queries and keys in a latent space with varying lag lengths, where each lag length has different learnable contribution weights.
> >
> >
> >
> > - **[2. Incomplete sequences]** Correlation, by itself, cannot directly handle incomplete sequences. To address this, we simulated scenarios where 10%, 25%, 50%, and 75% of data points are **randomly missing**, with the missing values filled using linear interpolation based on adjacent values, as detailed in **L471--475**. As shown in **Figure 10**, the results demonstrate that our method **is robust** under these missing data scenarios, both in terms of **a) forecasting performance** and the **b) CD ratio of the CM**.
> >
> >
> > &nbsp;
> >
> >
> > If the reviewer has additional suggestions for alternative metrics, we would be happy to explore them and incorporate them into our work as appropriate.
> >
> >
> > &nbsp;

---

> > > ### Author Response · Authors · 2024-11-19
> > >
> > > ## **W4. Lack of analysis regarding the domain parameters**
> > >
> > >
> > > Contrary to the reviewer's concern, we believe we have already conducted ***extensive experiments and analyses*** on the domain parameters, as also acknowledged by **reviewers bW9C, JKMr, 21yw**.
> > >
> > >
> > > Specifically, we have conducted experiments with **various backbones and datasets** both (a) quantitatively and (b) qualitatively, as follows:
> > > - (Backbone 1) UniTS
> > >    - (a) **Table 7**: Ablation study regarding the use of correlation matrix and “domain parameters”.
> > >    - (a) **Figure 7**: “Domain parameters” enable the CD ratio to better reflect the preference for channel dependencies.
> > >    - (b) **Figure 8**: “Domain parameters” capture the hidden relationships between channels, which are unobservable in the data space.
> > >    - (b) **Table 8 (+ Figure 6)**: TS visualization vs. CD ratio (w/ and w/o “domain parameters”).
> > > - (Backbone 2) iTransformer
> > >    - (a) **Table 13 (+ Table 15)**: Various options for “domain parameters”.
> > > - (Backbone 3) TimeSiam
> > >    - (a) **Appendix E.1**: Ablation study regarding the use of correlation matrix and “domain parameters”.
> > >
> > >
> > > &nbsp;
> > >
> > >
> > > If the reviewer feels that certain analyses are missing and can provide specific suggestions, we would be happy to include them in our paper.
> > >
> > >
> > > &nbsp;
> > >
> > >
> > > ---
> > >
> > > ## **W5-1. Lack of comparison with other plugins**
> > >
> > >
> > > We acknowledge that we did not include comparisons with other plug-in methods. However, to the best of our knowledge, no other **plug-in methods specifically designed to capture varying CD across multiple datasets** and adapted for TSFMs exist. While methods orthogonal to our approach may improve performance when combined, they are out of scope for our work.
> > >
> > >
> > > While we attempted to compare our method with the method [A] the reviewer mentioned under the setting of a **single-task model** (with a **single dataset**), we found that LIFT [A] does not provide code for **recent CD models** (e.g., iTransformer [B] (ICLR 2024)), and includes only MTGNN [C] (KDD 2020) and Crossformer [D] (ICLR 2023) for its CD models.
> > >
> > >
> > > However, we agree that LIFT, as a “plug-in method allowing lagged variates to utilize advance information from leading indicators”, is related to our work. Therefore, we have included a discussion of LIFT in **L87--88** of **Section 2: Related Works**.
> > >
> > >
> > > &nbsp;
> > >
> > >
> > > [A] Zhao, Lifan, and Yanyan Shen. "Rethinking Channel Dependence for Multivariate Time Series Forecasting: Learning from Leading Indicators." ICLR (2024)
> > >
> > >
> > > [B] Liu, Yong, et al. "iTransformer: Inverted transformers are effective for time series forecasting." ICLR (2024)
> > >
> > >
> > > [C] Wu, Zonghan, et al. "Connecting the dots: Multivariate time series forecasting with graph neural networks." KDD (2020)
> > >
> > >
> > > [D] Zhang, Yunhao, and Junchi Yan. "Crossformer: Transformer utilizing cross-dimension dependency for multivariate time series forecasting." ICLR (2023)

---

> > > > ### Author Response · Authors · 2024-11-19
> > > >
> > > > While we believe we addressed all your concerns in our rebuttal and the revised manuscript, please feel free to ask us if we may have missed anything or you have additional questions; we are happy to address them.

---

> ### Comment · Reviewer_saeS · 2024-11-25
>
> Thanks for the response. I'm still concerned about the applicability of CM. As the author emphasizes in the title of the paper and lines 22-23, CM serves the ***time series foundation models***. However, this paper only uses a foundation model for experiments, which I think is not enough. Specifically, the ICML 2024 papers were made available online in February, accepted in May, and the latest ones were open-sourced in May as well. ***The UniTS architecture used in this paper was accepted by NeurIPS 2024 (the acceptance time of this conference is later than ICML 2024). Therefore, the excuse of working during the same period may not be reasonable***. Furthermore, the issues with the architecture and dataset further demonstrate the limitations of the proposed method. Your approach is designed for foundation models, but it does not easily apply to most existing foundation models, which leads to significant limitations in applicability. Therefore, I will maintain my score.

---

> > ### Author Response · Authors · 2024-12-01
> >
> > We realized that you also left your comments here as well as below our general comment. Below we provide responses to your concerns, which are mostly duplicated with those above.
> >
> > &nbsp;
> >
> > > ### 1. Applicability of CM
> >
> > Our title explicitly addresses **"channel dependence (CD)"**, and as such, we believe its scope is ***naturally limited to CD models***. Additionally, **L69–70** and **L55–56** clarify that our method applies only to models that capture CD using attention mechanisms, not to all TSFMs. Therefore, we think the title accurately reflects the scope of our method while maintaining conciseness.
> >
> > &nbsp;
> >
> > > ### 2. Experiments with TSFMs of ICML 2024
> >
> > It is infeasible to experiment with the models mentioned by saeS due to **"fundamental limitations"** of these models. As stated above, our method is designed for ***models that "capture CDs using attention mechanisms"***, and the works referenced by saeS do not meet this criterion.
> >
> > Additionally, we believe your criticism regarding the timeline is both unreasonable and invalid. Following up on UniTS [NeurIPS 2024] quickly ***does not imply we could experiment with all prior works***. Given the two-month gap between the ICML 2024 and ICLR 2025 submission, addressing all requested papers was infeasible. Our progress on UniTS was only possible because their code is released early and affordable enough for quick adaptation.
> >
> > In summary, UniTS is the state-of-the-art TSFM around the time of submission, and our work effectively improves it, outperforming two TSFMs (UniTS and GPT4TS). All other TSFMs (including those mentioned by Reviewer saeS) were not applicable/reproducible at that time, and/or they should be considered as concurrent works to ours.

---

> ### Comment · Reviewer_saeS · 2024-12-03
>
> Thank you for your response. After careful consideration, I maintain my decision based on the following points:
>
> 1.Utilizing attention mechanisms to capture channel correlations is not a universally accepted approach; alternatives such as GNN[1] also exist. While PCD offers potential value, its applicability appears limited to a narrow range of scenarios.
>
> 2.Verifying the method only with UniTS does not sufficiently demonstrate the generalizability of PCD.
>
> 3.A comprehensive comparison with mainstream time series foundation models is essential. If the model with PCD cannot achieve competitive performance against state-of-the-art models, its contribution to the field may be perceived as limited.
>
> Given these considerations, I suggest allocating more time to strengthen the evidence for your work. This conference may not be the most appropriate venue for its publication at this stage. Additionally, it seems the authors are more enthusiastic about engaging in discussions than optimizing the manuscript itself, which is worth reflecting upon.
>
> [1] Wu, Z., Pan, S., Long, G., Jiang, J., Chang, X., & Zhang, C. (2020, August). Connecting the dots: Multivariate time series forecasting with graph neural networks. In Proceedings of the 26th ACM SIGKDD international conference on knowledge discovery & data mining (pp. 753-763).

---

> > ### Author Response · Authors · 2024-12-03
> >
> > Thank you for clarifying your points.
> >
> > While we are enthusiastic about *both* engaging in discussions and optimizing the manuscript itself, please note that the deadline for updating the manuscript already passed around a week ago. However, we assure that all meaningful discussions during this rebuttal period will be reflected in the final revision.
> >
> > &nbsp;
> >
> >
> > > ### *1. Utilizing attention mechanisms to capture channel correlations is not a universally accepted approach*
> >
> >
> > We believe that "utilizing attention mechanisms to capture channel correlations" is indeed a universally accepted approach, following the **iTransformer [A] (ICLR 2024, citation 432)**, which shifts the focus of utilizing attention mechanism **from capturing temporal dependencies (TD) to capturing CD** by treating each "channel" as a token rather than each "patch".
> >
> >
> > Regarding your statement *"alternatives such as GNN[1] also exist"*, we are aware that GNNs have successfully been applied to spatio-temporal TS forecasting. We agree that the usage of GNNs to process channel-wise interaction is an interesting idea, but we believe this is beyond the scope of our work.
> >
> > &nbsp;
> >
> >
> > > ### *2. Verifying the method only with UniTS does not sufficiently demonstrate the generalizability of PCD.*
> >
> >
> > Our experimental verification includes **three CD models**: UniTS [B], iTransformer [A], TimeSiam [C]. Please note that UniTS was the only available CD TSFM at the time of submission (and based on our recent survey, still no other CD TSFMs are available).
> >
> > &nbsp;
> >
> > > ### *3. A comprehensive comparison with mainstream time series foundation models is essential. If the model with PCD cannot achieve competitive performance against state-of-the-art models, its contribution to the field may be perceived as limited.*
> >
> > We have already compared two TSFMs: GPT4TS [D] and UniTS [B]. On top of that, (although the comparison might not be fair, because the authors of TSFMs mostly did not release their code for replication,) as we mentioned in the General Comment (*Why not compare with other (CI or CD) TSFMs?*), PCD seems to be competitive or even outperform other types of TSFMs. Please note that all prior/concurrent TSFM works have a similar amount of comparison (0--2 prior TSFMs), due to the difficulty of reproducibility.
> >
> > &nbsp;
> >
> > Please let us know if you have further concerns.
> >
> > &nbsp;
> >
> > [A] Liu, Yong, et al. "iTransformer: Inverted transformers are effective for time series forecasting." ICLR (2024)
> >
> > [B] Gao, Shanghua, et al. "Units: A Unified Multi-task Time Series Model." NeurIPS (2024)
> >
> > [C] Dong, et al. "TimeSiam: A Pre-Training Framework for Siamese Time-Series Modeling." ICML (2024)
> >
> > [D] Zhou, et al. “One fits all: Power general time series analysis by pretrained lm.” NeurIPS (2023)

---

### Official Review · Reviewer_bW9C · 2024-11-04

**Soundness:** 3
**Presentation:** 3
**Contribution:** 2
**Rating:** 6
**Confidence:** 4

**Summary:**

This paper presents Partial Channel Dependence (PCD), a method designed to capture the varying dependencies between channels across different datasets. PCD achieves this by applying channel-wise attention multiplied by the corresponding dataset mask. Experimental results demonstrate that PCD yields an average performance improvement across various time series tasks.

**Strengths:**

* The method is straightforward, and the motivation is clearly articulated.
* Extensive experiments across various scenarios validate that PCD enhances the performance of Transformer-based multivariate time series models.

**Weaknesses:**

* The results for iTransformer and PatchTST presented in Table 3 differ significantly from those reported in the original papers. Additionally, the ETTh2, ETTm1, and ETTm2 datasets are well-known multivariate time series datasets. Could you please provide a comprehensive comparison of the same baselines on these datasets across all prediction lengths?
* While PCD does enhance the performance of Transformer-based models for multivariate time series forecasting, I recommend including recent MLP-based and CNN-based models in the baselines, such as RLinear and ModernTCN. Additionally, GNN-based models like CrossGNN are also adept at capturing multivariate relationships. Including these would strengthen the performance comparisons.
* The length of the input time series is an important factor influencing experimental results. Therefore, I would like to see the impact of varying sequence lengths on the results w CM and w/o CM.
* There are several literature on channel dependency modeling, such as [1-2]. Detailed discussion is needed.

[1] Zhao et al., Rethinking Channel Dependence for Multivariate Time Series Forecasting: Learning from Leading Indicators, ICLR 2024

[2] Qi et al., Enhancing Multivariate Time Series Forecasting with Mutual Information-driven Cross-Variable and Temporal Modeling

**Questions:**

1.	Could the authors apply a similar masking approach to CI models like PatchTST or PITS, and compare their performance across different datasets? If possible, please provide performance comparisons for the original settings w CM and w/o CM.

---

> ### Author Response · Authors · 2024-11-19
>
> ## **W1-1. Inconsistent results with the original paper**
>
>
> We acknowledge the inconsistency between the results reported in the original paper and our paper. However, we reproduced the results using the **official code repositories** of iTransformer [A] and UniTS [B] for our experiments, running the provided scripts **without any modifications**.
>
>
> Unfortunately, this reproduction yielded **inconsistent results** for both models compared to the original paper, and we found that this inconsistency is shared with others, according to GitHub issues in both the iTransformer and UniTS repositories.
> - Specifically, for iTransformer, we found a similar issue reported in a Chinese GitHub discussion, which we translated to confirm its relevance.
> - For UniTS under the prompt tuning setting, we encountered an issue where the model failed to converge using the provided script and the problem is also reported by others in a GitHub issue. This issue was resolved by setting the hidden dimension to $D=32$, which yielded degraded performance compared to the original paper. We applied this setting uniformly across both UniTS and its application to our method for fair comparison, as detailed in **Appendix B.2**.
>
>
>
>
> Nonetheless, following the reviewer’s feedback,  we have added further reproduction details in **Appendix B** to avoid any confusion.
>
>
> &nbsp;
>
>
> [A] Liu, Yong, et al. "iTransformer: Inverted transformers are effective for time series forecasting." ICLR (2024)
>
>
> [B] Gao, Shanghua, et al. "Units: Building a unified time series model." NeurIPS (2024)
>
>
> &nbsp;
>
>
>
>
>
>
> ## **W1-2. Full results of ETT datasets (across all prediction lengths)**
>
>
> We agree with the reviewer’s observation that the ETT datasets are widely used and important for TS forecasting tasks. However, due to **space constraints**, we have provided the comprehensive results for these datasets, including comparisons across all prediction lengths, in **Appendix C.1**, with a reference included in **L251**.
>
>
> &nbsp;
>
> ---
>
> ## **W2.  Comparison with other baseline methods.**
>
>
> Our primary focus is on ***improving TSFMs' ability to better capture CD***, rather than achieving *better performance compared to non-TSFMs*, including recent methods (e.g., RLinear, ModernTCN, CrossGNN) mentioned by the reviewer.
> Nevertheless, we compared some non-TSFMs (task-specific models) in Table 3, following the **baseline established by the backbone model** we used (UniTS [A]), as the proposed CM is a plug-in method.
> Nonetheless, recognizing the significance of the methods mentioned by the reviewer, **we have included them** in  **Section 2. Related Works** to highlight their relevance to this area of research.
>
>
> &nbsp;
>
>
> [A] Gao, Shanghua, et al. "Units: Building a unified time series model." NeurIPS (2024)
>
>
> &nbsp;
>
> ---
>
> ## **W3.  Varying size of lookback window ($L$)**
>
>
> Thank you for your insightful comment.
>
>
> We set $L$ to be the **same** as that used in the **backbone method we applied**, ensuring a fair and consistent comparison.
> Nonetheless, we agree with the reviewer on the importance of the **size of the lookback window ($L$)** and we have conducted an **additional ablation study** presented in **Appendix F**, using three datasets (ECL, Traffic, PEMS03). In these experiments, we varied $L \in \\{48, 96, 192, 336, 720\\}$, with a forecast horizon of $H = 12$ for PEMS03 and $H = 96$ for the other datasets, in accordance with previous work [A]. The results demonstrate that the impact of CM ***remains consistent across different values of $L$***, further validating its robustness.
>
>
> &nbsp;
>
>
> [A] Liu, Yong, et al. "iTransformer: Inverted transformers are effective for time series forecasting." ICLR (2024)

---

> > ### Author Response · Authors · 2024-11-19
> >
> > ## **W4.  Additional related Works.**
> >
> >
> > Thank you again for your insightful comment.
> >
> >
> > We agree that LIFT [A], a plug-in method enabling lagged variates to utilize advanced information from leading indicators, and CDAM [B], which captures CD by minimizing redundant information while enhancing relevant mutual information, are related to our work.
> >
> >
> > However, to the best of our knowledge, no other **plug-in methods specifically designed to capture varying CD across multiple datasets** and adapted for TSFMs exist. While methods orthogonal to our approach may improve performance when combined, they are out of scope for our work.
> >
> >
> > While we attempted to compare our method with the methods [A,B] the reviewer mentioned under the setting of a **single-task model** (with a **single dataset**), we found this infeasible for the following reasons:
> > - (1) **LIFT**: LIFT does not provide code for **recent CD models** (e.g., iTransformer [C] (ICLR 2024)), and includes only MTGNN [D] (KDD 2020) and Crossformer [E] (ICLR 2023) for its CD models.
> > - (2) **CDAM**: CDAM does not provide code.
> >
> >
> > Nonetheless, as both methods (LIFT [A] and CDAM [B]) are related to our work as plug-in methods for handling CD in TS forecasting tasks, we have included them in **L87-89** of **Section 2: Related Works**.
> >
> >
> > &nbsp;
> >
> >
> > [A] Zhao, Lifan, and Yanyan Shen. "Rethinking Channel Dependence for Multivariate Time Series Forecasting: Learning from Leading Indicators." ICLR (2024)
> >
> >
> > [B] Qi, Shiyi, et al. "Enhancing Multivariate Time Series Forecasting with Mutual Information-driven Cross-Variable and Temporal Modeling." arXiv (2024)
> >
> >
> > [C] Liu, Yong, et al. "iTransformer: Inverted transformers are effective for time series forecasting." ICLR (2024)
> >
> >
> > [D] Wu, Zonghan, et al. "Connecting the dots: Multivariate time series forecasting with graph neural networks." KDD (2020)
> >
> >
> > [E] Zhang, Yunhao, and Junchi Yan. "Crossformer: Transformer utilizing cross-dimension dependency for multivariate time series forecasting." ICLR (2023)
> >
> >
> > &nbsp;
> >
> > ---
> >
> > ## **Q1.  Application to CI models.**
> >
> >
> > PCD aims to enhance ***TSFMs capturing CDs*** by leveraging dataset-specific characteristics, rather than proposing a plug-in-play method applicable to any type of TSFMs existing in the world, as mentioned in **L60--61** and **L69--70**. Specifically, CM is designed to adjust the **CD estimated by the model**, making it **applicable to CD models** but **infeasible for CI models**.
> >
> >
> > &nbsp;
> >
> >
> > We focus on ***TSFMs capturing CDs*** because recent methods--particularly with the **emergence of iTransformer** [A]--have increasingly emphasized **leveraging attention mechanisms to capture CD** [B, C, D, E]. Our method aligns with this trend and builds upon the strengths of these approaches. Furthermore, it is also important to highlight the inherent **trade-offs** between CD and CI models: CD models offer greater capacity but lower robustness, while CI models exhibit higher robustness at the expense of capacity [F]. This makes CD models more suited for settings where the model is trained on ***large or multiple datasets, aligning with the use of CM***.
> >
> > &nbsp;
> >
> >
> > Nonetheless, as discussed in **L482--483**, the development of a novel approach to achieve PCD without relying on Transformer-based methods is left for future work. Our current focus remains on **utilizing the dataset-specific information** to enhance the capture of CD within CD models.
> >
> >
> > &nbsp;
> >
> >
> > [A] Liu, Yong, et al. "iTransformer: Inverted transformers are effective for time series forecasting." ICLR (2024)
> >
> >
> > [B] Gao, Shanghua, et al. "Units: Building a unified time series model." NeurIPS (2024)
> >
> >
> > [C] Ilbert, Romain, et al.  “SAMformer: Unlocking the Potential of Transformers in Time Series Forecasting with Sharpness-Aware Minimization and Channel-Wise Attention.” NeurIPS (2024)
> >
> >
> > [D] Wang, Xue, et al. “CARD: Channel aligned robust blend transformer for time series forecasting.” ICLR (2024)
> >
> >
> > [E] Yu, Guoqi, et al.  “Revitalizing Multivariate Time Series Forecasting: Learnable Decomposition with Inter-Series Dependencies and Intra-Series Variations Modeling.” ICML (2024)
> >
> >
> > [F] Han, Lu, Han-Jia Ye, and De-Chuan Zhan. "The capacity and robustness trade-off: Revisiting the channel independent strategy for multivariate time series forecasting."  TKDE (2024)

---

> > > ### Author Response · Authors · 2024-11-19
> > >
> > > While we believe we addressed all your concerns in our rebuttal and the revised manuscript, please feel free to ask us if we may have missed anything or you have additional questions; we are happy to address them.

---

### Official Review · Reviewer_BSUW · 2024-11-04

**Soundness:** 2
**Presentation:** 3
**Contribution:** 3
**Rating:** 6
**Confidence:** 3

**Summary:**

The paper introduces a novel concept called Partial Channel Dependence (PCD) to address implicit heterogeneity in time series data, specifically focusing on varying dependencies between channels. The authors propose a channel mask mechanism that combines correlation matrices (for relative dependencies) with learned domain parameters (for absolute dependencies). The approach is evaluated across multiple time series tasks (forecasting, classification, imputation, and anomaly detection) in both few-shot and zero-shot settings, demonstrating its versatility across different foundation models and single-task models.

**Strengths:**

- Introduces a novel methodology for handling channel relationships specifically designed for pretraining foundation models in time series
- Proposes a systematic framework for incorporating dataset-specific channel dependencies into the pretraining process
- Demonstrates superior performance in challenging scenarios (few-shot and zero-shot learning)

**Weaknesses:**

- **Dataset Identification Requirement**: The method requires knowing exactly which samples belong to which dataset during pretraining. This is a strong assumption that may not hold in real-world applications where data sources might be mixed or unclear.
- **Lack of Fixed vs. Variable Dependency Analysis**: The paper assumes variable channel dependencies are necessary but doesn't justify why a simpler fixed dependency structure wouldn't work equally well. Without this comparison, the added complexity of variable dependencies might be unnecessary.

- **Missing Critical Baseline Comparison**

    - No comparison against the same architecture with full channel dependence
    - No comparison against the same architecture with full channel independence

**Questions:**

- Can you provide comparisons of your method against the same architecture with full channel dependence and full channel independence?
- What theoretical guarantees or analysis can you provide to show when partial dependence would outperform the extreme cases?
- How does your method handle scenarios where dataset boundaries are ambiguous or when data comes from multiple unknown sources?
- Can you quantify the computational overhead and memory requirements of your approach compared to full dependence and independence cases?

---

> ### Author Response · Authors · 2024-11-19
>
> ## **W1, Q3.  Dataset Identification Requirement**
> The assumption noted by the reviewer is a ***fundamental premise for TSFMs in general*** [A, B, C, D]. While TSFMs are capable of training on multiple heterogeneous datasets simultaneously, they **typically require knowledge of each sample’s data source during training**. Assuming the dataset source is unknown during **training**--as opposed to **inference**--would be uncommon in practice. Instead, a more realistic scenario involves conducting inference on datasets that were unseen during training, which we have already addressed in **Table 6** and **Table 9**.
>
>
> That said, developing a TSFM that does not rely on dataset source information during training could be an intriguing direction for future research, even though it falls beyond the scope of our current work.
>
>
>
> If the reviewer is aware of any settings or models where this assumption is unnecessary, we would be grateful for the reviewer’s insight.
>
>
> &nbsp;
>
>
> [A] Zhou, et al. “One fits all: Power general time series analysis by pretrained lm.” NeurIPS (2023)
>
>
> [B] Goswami, et al. “Moment: A family of open time-series foundation models.” ICML (2024)
>
>
> [C] Liu, et al. “Timer: Generative pre-trained transformers are large time series models.” ICML (2024)
>
>
> [D] Woo, et al. “Unified training of universal time series forecasting transformers.” ICML (2024)
>
>
> &nbsp;
>
> ---
>
> ## **W2.  Lack of Fixed vs. Variable Dependency Analysis**
>
>
> ### **W2-1. Why fixed dependency structure for TSFM wouldn't work equally well?**
>
>
> In a conventional setting where a single model is trained on a single dataset (i.e., **non-TSFM**), varying CDs across datasets may not pose a significant issue; we can choose either CI or CD model to perform effectively on that particular dataset.
>
>
> *However*, the challenge arises when training a **TSFM**, which is a ***single*** model designed to generalize across ***multiple*** datasets with varying CDs. Because ***each dataset may benefit from a different approach (CI or CD)***, it is crucial for a TSFM to account for varying CD across datasets, as illustrated in **Figure 1** and noted in **L43--46**.
>
>
> Despite this, previous TSFMs have mainly focused on accommodating **explicit** heterogeneity, such as difference in dataset shape and size, while overlooking the **implicit** heterogeneity stemming from varying CD, as noted in **L47--50**. This oversight motivated us to propose a **PCD framework** that ***adjusts the CDs of various datasets (captured by a single model)*** based on  the **unique characteristics of each dataset**  to handle the  **implicit heterogeneity**. To achieve PCD, we propose a channel mask (CM) constructed  **for each dataset**, which is then element-wise multiplied with the CD captured by a model (attention matrix) for adjustment, with the visualization of CM of ETTh1 dataset shown in the **second panel of Figure 5**.
>
>
>
>
>
>
>
>
> &nbsp;
>
>
> ### **W2-2. Comparison of fixed vs. variable dependency analysis**
>
>
> The “fixed vs. variable dependency analysis” raised by the reviewer aligns with our **“baseline vs. baseline with CM”** comparisons, and is extensively addressed throughout the manuscript .
>
>
> The “baseline” assumes **fixed CD across datasets** by employing a **single** model, while the “baseline with CM” adjust the CDs captured by a single model by leveraging **dataset-specific information** to handle the **varying CD across datasets** .

---

> ### Author Response · Authors · 2024-11-19
>
> ## **W3, Q1.  Missing critical baseline comparison (PCD vs. Full CD/CI)**
>
> > **PCD  ($\mathbf{A}=\mathbf{M}$) vs. Full CD ($\mathbf{A}=\mathbf{1}$)**
>
> It is important to clarify that the **"full CD"** model mentioned by the reviewer corresponds to the **"baseline method" (without CM)**, as it employs CD captured by the model **without adjustment**, which also corresponds to the **fixed dependency structure** in **W2-2**. This (baseline (= Full CD)  vs. baseline with CM (= PCD)) comparison has been extensively conducted in our experiments, consistently demonstrating the superiority of our PCD over full CD.
>
> &nbsp;
>
>
> > **PCD  ($\mathbf{A}=\mathbf{M}$) vs. Full CI ($\mathbf{A}=\mathbf{I}$)**
>
> As discussed in **L60---61**,  CM is designed to adjust the **CD estimated by the model**, making it **applicable to CD models** but **infeasible for CI models**. However, to facilitate a comparison, we can enforce the CD model to behave as CI by replacing $\mathbf{A}$ in **Equation 1** with the identity matrix, allowing us to compare CD and CI under the same architecture. We have conducted this comparison in **Figure 7**, which shows the performance gain achieved by applying (full) CD over (full) CI, demonstrating that **full CD > full CI**, and consequently **PCD > full CI**.
>
> &nbsp;
>
>
> Initially, we omitted the full CI case from **Table 7** as our focus was on adjusting the CD model (starting with the CD model as the baseline). However, we have now included the results in the **first row of Table 7**, with the first and second rows corresponding to full CI and full CD, respectively.
>
> &nbsp;
>
> ---
>
> ## **Q2.  PCD in the extreme (full CI/CD) cases**
>
> In the extreme cases mentioned by the reviewer, the **correlation matrix** aligns naturally with either CI or CD for the following reasons:
> - **Full CI**: If the two channels are **completely independent**, the correlation is “0”, resulting in the correlation matrix becoming the **identity matrix**.
> - **Full CD**: If the two channels are **completely correlated** (either positively or negatively), the correlation is either **+1 or -1**, resulting in an absolute correlation of one and producing a **matrix of ones** (or negative ones).
>
>
> The CM extends the flexibility of the correlation matrix by applying **monotonic transformation** via affine adjustments and mapping values to the range (0,1) using a sigmoid function. As a result, the CM aligns with either CI or CD in these extreme cases, while accommodating intermediate levels of dependence.
>
> &nbsp;
>
> To further address concerns regarding PCD’s performance under extreme cases, we performed an **additional toy experiment** in **Appendix G**. Specifically, we designed a scenario where the channels in TS exhibit **no correlation**. We generated a synthetic TS dataset with two channels using sine waves oscillating at frequencies of 0.5 and 2.0 over a length of 18,000 (similar to ETTh1). The results show that the **CD ratio of CM is approximately 0.018** and the **forecasting MSE is around 0.0014**, confirming strong channel independence and demonstrating the effectiveness of our method even under extreme CI conditions.
>
>
> &nbsp;
>
> ---
>
> ## **Q4.  Efficiency analysis (vs. CD vs. CI)**
>
> As stated in W3(Q1), CM adjusts CD captured from the CD model and applies only to it, so we compare exclusively with the CD model. The results of this comparison are ***already*** presented in **Table 14** and **L465--470**. These results show that incorporating CMs **does not significantly impact computational time**, even for datasets with a large number of channels. Additionally, as noted in **L463--464**, incorporating CM requires only two additional parameters per dataset, resulting in minimal memory overhead and maintaining computational efficiency.

---

> > ### Author Response · Authors · 2024-11-19
> >
> > While we believe we addressed all your concerns in our rebuttal and the revised manuscript, please feel free to ask us if we may have missed anything or you have additional questions; we are happy to address them.

---

### Official Review · Reviewer_21yw · 2024-11-04

**Soundness:** 1
**Presentation:** 2
**Contribution:** 2
**Rating:** 3
**Confidence:** 3

**Summary:**

This paper addresses the limitation that time series models often consider only explicit heterogeneity among datasets, such as varying sequence lengths and numbers of channels, while overlooking implicit heterogeneity like channel dependencies within the data. The authors propose a module called the Channel Mask (CM) to enable models to reflect channel dependence (CD) by adjusting the degree of CD based on dataset characteristics, achieving Partial Channel Dependence (PCD). By integrating CM into existing time series models, they demonstrate improved performance across various time series tasks.

**Strengths:**

1. The paper critiques the limitations of using correlation coefficients for measuring inter-channel relationships and proposes a new way to measure channel dependence through the CD ratio. The CM introduces very few parameters (α and β from domain parameters) yet effectively learns the implicit inter-channel relationships, leading to performance improvements in time series models.

2. The authors validate their approach through various experiments, including few-shot and zero-shot settings, and provide thorough ablation studies and analyses to demonstrate the effectiveness and suitability of the CM structure and its relationship with CD.

**Weaknesses:**

1. The paper lacks theoretical grounding or in-depth analysis to explain why the proposed method leads to performance improvements. A deeper understanding of the underlying mechanisms would clarify and strengthen the contribution.
1.1. For example, how can (or how should) we define the concept of “implicit heterogeneity”? Why do we need this concept? While there are studies on channel dependence in time series machine learning, is the concept in this work different from existing studies? Furthermore, how is the rigorous definition of implicit heterogeneity connected to the proposed CM method?  What is the rationale behind the use of correlation information between different real-world time series datasets in different domains to achieve better performance?
1.2. For another example, while the use of correlation matrices primarily captures linear relationships between channels, the application of a sigmoid function in the CM introduces nonlinearity to the model. The authors mention that static correlations (global CD) are reflected through the CM, while dynamic, local correlations are captured by the attention mechanism. This design may help address some aspects of nonlinearity and temporal variation in channel dependencies. However, it remains a question whether this approach is fully sufficient to reflect the complex and changing characteristics inherent in time series data. A clear explanation of the rationale behind how the CM models these dynamic changes, or further investigation into its effectiveness in this regard, would strengthen our understanding. These considerations might be related to the rigorous definition of implicit heterogeneity in time series.
1.3. Meanwhile, the authors state that “However, most previous works have focused on the model architecture to either capture or disregard CD, often overlooking the potential differences in CD across datasets.” However, they do not show specific theoretical analysis results on why and how existing studies on the model architecture are limited in capturing the differences in CD across datasets.

2. The technical contribution is also limited.
2.1. Following the above comment, the claim that existing CD models overlook differences in CD between datasets may not be fully substantiated. Since models like Crossformer, iTransformer, and TimeSiam learn attention patterns specific to each dataset, it's unclear whether they truly neglect dataset-specific CD differences.
2.2. While the authors introduce PCD as an intermediary concept between channel independence (CI) and channel dependence (CD), the structure of CM, which uses a channel correlation matrix, cannot be applied to CI models. The study applies CM to CD models (iTransformer, UniTS) and shows performance improvements but does not verify whether applying CM to CI models can enhance performance. Thus, the proposed PCD cannot be extended to CI settings, limiting its utility in models that assume channel independence.
2.3. The CM seems to be applicable only to Transformer models that apply attention along the channel axis and cannot be applied to models that apply attention along the time axis. For instance, existing studies like ST-MEM [1] and UniTST [2], which learn CD by applying attention along the time axis after channel flattening, cannot utilize CM. This limitation reduces the general applicability of the proposed method to other architectures.
- [1] Na, Y., Park, M., Tae, Y., & Joo, S. (2024). Guiding Masked Representation Learning to Capture Spatio-Temporal Relationship of Electrocardiogram. arXiv preprint arXiv:2402.09450.
- [2] Liu, J., Liu, C., Woo, G., Wang, Y., Hooi, B., Xiong, C., & Sahoo, D. (2024). UniTST: Effectively Modeling Inter-Series and Intra-Series Dependencies for Multivariate Time Series Forecasting. arXiv preprint arXiv:2406.04975.

2.4. The CD ratio proposed in the study is calculated after combining CM with the time series model and training, and its value may vary depending on the model used. The paper does not provide sufficient evidence to demonstrate that the CD ratio is consistent across different models for the same dataset, raising concerns about its adequacy as a metric for measuring channel dependence.
2.5. Meanwhile, although the authors started their argument from the emergence of time series foundation models (TSFMs), they do not provide sufficient validation of this work for different TSFMs.

**Questions:**

Please see the weaknesses. Some other (related) questions are as follows.
1. In the experiments involving TimeSiam, which encoder was used? The TimeSiam paper proposes both PatchTST (CI properties) and iTransformer (CD properties) as encoders. Since they have different characteristics, specifying the encoder used is essential. Additionally, since TimeSiam is utilized for classification tasks in your paper, it should be categorized appropriately, and the performance of CM + TimeSiam in classification tasks should also be evaluated.
2. The CI framework does not attend between channels but attends with timestamps or patches in each channel. Then, how is the formulation of A in equation (1) justified as the identity matrix in equation (1) indicates the channel relationship? For example, how does PatchTST match to this formulation?
3. Is the CD ratio consistent across different models for the same dataset? To establish the CD ratio as a reliable metric for measuring dataset CD, it should be verified whether CD ratios computed using different models (e.g., iTransformer vs. UniTS) yield consistent results.
4. Since the CM has been validated on datasets with significant CD, it would be beneficial to test its performance on synthetic datasets with uncorrelated channels to verify that CM yields a low CD ratio and does not introduce unnecessary dependencies.
5. Time series data often exhibit non-stationarity, which can affect correlation measures. How does your method handle non-stationary data, and does the CM adjust for changes in channel dependencies over time?
6. Have you considered other measures that can capture nonlinear or more complex dependencies between channels, such as mutual information? This could potentially enhance the CM's ability to model complex inter-channel relationships.

---

> ### Author Response · Authors · 2024-11-19
>
> ## **W1. Lack of theoretical grounding or in-depth analysis**
>
>
> While we believe our paper has ***already*** addressed most of the concerns raised by the reviewer, below we provide explanations and references for each (W1-1, 1-2, W-3).
> &nbsp;
>
>
> Nonetheless, to better clarify (1) the concept of implicit/explicit heterogeneity and (2) the motivation for addressing CDs varying across TS datasets, we have revised the **1. Introduction** section in the **revised manuscript**. We kindly invite you to reread the revised **1. Introduction** section to facilitate the below discussions.
>
>
> &nbsp;
>
>
> ### **W1-1. Definition of implicit heterogeneity**
>
>
> > **W1-1-a) Concept of implicit heterogeneity and its necessity**
>
> - **[Concept]** As stated in **L37--43**, ***”explicit” heterogeneity*** refers to observable differences among datasets, such as varying sequence lengths and the number of channels. In contrast, ***”implicit” heterogeneity*** pertains to **differences in unobservable factors** across datasets, such as variations in CD, where some datasets exhibit strong inter-channel dependencies, while others require weak or minimal dependencies.
> - **[Necessity]** As highlighted in **L43--46**, addressing implicit heterogeneity is **essential for TSFMs**; unlike a dataset-specific model tailored to a single dataset, a TSFM is a **single** model trained on **multiple** datasets with varying CD, each potentially requiring a distinct approach (CI or CD), as illustrated in **Figure 1**. We also show that addressing this heterogeneity brings a significant performance gain on various tasks, as shown in **Table 7** ( second row (CD) vs. last row (PCD) ).
> - **[Motivation]** This underscores the importance of considering TSFMs not only in terms of explicit heterogeneity (focusing on the model architecture), but also in terms of **implicit heterogeneity (focusing on the dataset itself)**.
>
>
> &nbsp;
>
>
> > **W1-1-b) Concept of CD & Comparison with previous works**
>
> The concept of CD we refer to **aligns with** that in prior works, but the **key distinction lies in focus**. While previous works emphasize capturing CD within a ***single*** dataset through ***model architecture***, we address varying CD across ***multiple*** datasets by concentrating on ***dataset characteristics***.
>
>
>
>
> We believe the following comparison with previous works (focusing on CD) highlights the distinctiveness of our work and the differing focus on CD:
>
>
> - **(1) Non-TSFMs**: A single model is trained on a single dataset, and works on the trained data distribution. Their research focus is on designing model architectures capturing CD within a single dataset.
> - **(2) Prior TSFMs**: A single model is trained across multiple datasets, and works on any data distribution. Their research focus is on designing model architectures accommodating multiple datasets with varying input shapes (**explicit heterogeneity**)
> - **(3) Ours**: A single model is trained across multiple datasets, and works on any data distribution. Our research focus is on designing model architectures better accommodating multiple datasets with varying CD (**implicit heterogeneity**) based on **dataset** characteristics.
>
>
> &nbsp;
>
>
> > **W1-1-c) Implicit heterogeneity and its connection with CM**
>
>
> To address **”implicit heterogeneity”** across TS datasets, we introduce the concept of partial channel dependence (PCD), which adjusts the CD captured by the model using **dataset-specific information** to align with the unique characteristics of each dataset.
> PCD is implemented through a **”channel mask (CM)”**--a matrix constructed from **dataset-specific information (e.g., correlation)**--that is multiplied with the model's CD (e.g., the attention matrix) for adjustment.

---

> ### Author Response · Authors · 2024-11-19
>
> ### **W1-2. CM may be insufficient to capture temporal variation in CD**
> Regarding the phrase ***"complex and changing characteristics inherent in time series data"*** mentioned by the reviewer, it could be interpreted in two ways:
> - (1) (Narrow meaning) Temporal variation (distribution shift)
> - (2) (broad meaning) Inherent dynamic nature of the TS dataset
>
>
> &nbsp;
>
>
> Below are the responses to each interpretation:
> &nbsp;
> > **(Narrow meaning) Temporal variation**
>
>
> It is important to emphasize that the primary role of CM is to capture the varying CD ***between*** datasets, rather than ***within*** a single dataset over multiple time steps. Therefore, ***adjusting the (global) CM based on dynamic changes would conflict with its intended purpose***, potentially diminishing its ability to capture the stable, dataset-specific characteristics. To address dynamic changes within datasets the reviewer noted, we rely on the attention matrix, which is constructed using the local TS. This approach ensures that while the (global) CM provides a stable foundation, the (local) attention matrix directly addresses temporal and local shifts.
>
>
> Furthermore, as shown in **Table 12**, our experiments on 13 real-world datasets, known to exhibit **significant distribution shifts** [A], demonstrate that the (global) CM and the (local) attention matrix effectively complement each other. The results indicate that the proposed method effectively captures both stability and adaptability under temporal variations.
>
>
> &nbsp;
>
>
> > **(Broad meaning) Inherent dynamic nature of the TS dataset**
>
> In response to the question about the sufficiency of our approach in capturing the dynamic and complex characteristics of TS, our work provides a **practical solution** for addressing CD in the TS domain. While we recognize the value of a comprehensive theoretical analysis, *exploring the full complexity of TS data goes beyond the scope of this study and is challenging to address within a conference paper*. We leave further investigation in this direction to future works.
>
>
> &nbsp;
>
>
> [A] Han, Lu, Han-Jia Ye, and De-Chuan Zhan. "The capacity and robustness trade-off: Revisiting the channel independent strategy for multivariate time series forecasting."  TKDE (2024)
>
> &nbsp;
>
> ---
>
> ## **W2. Limited technical contribution**
>
> ### **W2-1. Dataset-specific model for capturing dataset-specific CD**
> It is important to note that our work focuses on ***time series foundation models (TSFM)***, which involve training a **single** model on **multiple** datasets.
> By contrast, the models mentioned by the reviewer are ***dataset-specific models*** (i.e., single dataset-single model), which are beyond the scope of our current work . In such cases, varying CD across datasets is less of a concern, as each dataset is trained with its own individual model. However, this approach requires a **separate model for each dataset**, resulting in inefficiency.

---

> > ### Author Response · Authors · 2024-11-19
> >
> > ### **W2-2. CM cannot be applied to CI models.**
> > PCD aims to enhance ***TSFMs capturing CDs*** by leveraging dataset-specific characteristics, rather than proposing a plug-in-play method applicable to any type of TSFMs existing in the world, as mentioned in **L60--61** and **L69--70**. Specifically, CM is designed to adjust the **CD estimated by the model**, making it **applicable to CD models** but **infeasible for CI models**.
> >
> >
> > &nbsp;
> >
> >
> > We focus on ***TSFMs capturing CDs*** because recent methods--particularly with the **emergence of iTransformer** [A]--have increasingly emphasized **leveraging attention mechanisms to capture CD** [B, C, D, E]. Our method aligns with this trend and builds upon the strengths of these approaches. Furthermore, it is also important to highlight the inherent **trade-offs** between CD and CI models: CD models offer greater capacity but lower robustness, while CI models exhibit higher robustness at the expense of capacity [F]. This makes CD models more suited for settings where the model is trained on ***large or multiple datasets, aligning with the use of CM***.
> >
> > &nbsp;
> >
> >
> > Nonetheless, as discussed in **L482--483**, the development of a novel approach to achieve PCD without relying on Transformer-based methods is left for future work. Our current focus remains on **utilizing the dataset-specific information** to enhance the capture of CD within CD models.
> >
> >
> > &nbsp;
> >
> >
> > [A] Liu, Yong, et al. "iTransformer: Inverted transformers are effective for time series forecasting." ICLR (2024)
> >
> >
> > [B] Gao, Shanghua, et al. "Units: Building a unified time series model." NeurIPS (2024)
> >
> >
> > [C] Ilbert, Romain, et al.  “SAMformer: Unlocking the Potential of Transformers in Time Series Forecasting with Sharpness-Aware Minimization and Channel-Wise Attention.” NeurIPS (2024)
> >
> >
> > [D] Wang, Xue, et al. “CARD: Channel aligned robust blend transformer for time series forecasting.” ICLR (2024)
> >
> >
> > [E] Yu, Guoqi, et al.  “Revitalizing Multivariate Time Series Forecasting: Learnable Decomposition with Inter-Series Dependencies and Intra-Series Variations Modeling.” ICML (2024)
> >
> >
> > [F] Han, Lu, et al. "The capacity and robustness trade-off: Revisiting the channel independent strategy for multivariate time series forecasting."  TKDE (2024)
> >
> > &nbsp;
> >
> > ### **W2-3. The proposed method is only applicable to models based on attention-mechanism that captures CD.**
> >
> >
> > PCD aims to enhance ***TSFMs capturing CDs*** by leveraging dataset-specific characteristics, rather than proposing a plug-in-play method applicable to any type of TSFMs existing in the world, as mentioned in **L60--61** and **L69--70**.  Specifically, CM is designed to adjust the **CD estimated by the model**, making it **applicable to CD models** but **infeasible for CI models**.
> >
> >
> >
> > &nbsp;
> >
> >
> > Additionally, while we acknowledge that not all TSFMs rely on attention mechanisms and that Transformers are widely employed to capture temporal dependencies (TD), recent methods--particularly with the **emergence of iTransformer** [A]--have increasingly emphasized **leveraging attention mechanisms to capture CD** [B, C, D, E]. Our method aligns with this trend and builds upon the strengths of these approaches.
> >
> >
> > &nbsp;
> >
> >
> > Furthermore, we believe our method ***can be applied*** to two methods the reviewer mentioned (ST-MEM [F] and UniTST [G] ) that apply attention along the time axis after **channel flattening**. As these methods apply attention mechanism to capture both TD and **CD** simultaneously, we can extend the proposed CM (with shape $(C \times C)$) to shape $(C \cdot P \times C \cdot P)$, where $P$ is the number of patches in each channel, by duplicating the values within the same channels.
> >
> >
> > &nbsp;
> >
> >
> > Lastly, the limitation of being applicable only to the attention mechanism for capturing CD has been ***already*** discussed in **L482--484**, where we identified the development of a novel approach to achieve PCD without relying on Transformer-based methods as a promising direction for future work.
> >
> >
> > &nbsp;
> >
> >
> > [A] Liu, Yong, et al. "iTransformer: Inverted transformers are effective for time series forecasting." ICLR (2024)
> >
> >
> > [B] Gao, Shanghua, et al. "Units: Building a unified time series model." NeurIPS (2024)
> >
> >
> > [C] Ilbert, Romain, et al.  “SAMformer: Unlocking the Potential of Transformers in Time Series Forecasting with Sharpness-Aware Minimization and Channel-Wise Attention.” NeurIPS (2024)
> >
> >
> > [D] Wang, Xue, et al. “CARD: Channel aligned robust blend transformer for time series forecasting.” ICLR (2024)
> >
> >
> > [E] Yu, Guoqi, et al.  “Revitalizing Multivariate Time Series Forecasting: Learnable Decomposition with Inter-Series Dependencies and Intra-Series Variations Modeling.” ICML (2024)
> >
> >
> > [F] Na, Y. et al, “Guiding Masked Representation Learning to Capture Spatio-Temporal Relationship of Electrocardiogram”. ICLR (2024)
> >
> >
> > [G] Liu, J. et al, “UniTST: Effectively Modeling Inter-Series and Intra-Series Dependencies for Multivariate Time Series Forecasting” arxiv (2024)

---

> > > ### Author Response · Authors · 2024-11-19
> > >
> > > ### **W2-4, Q3.  Inconsistency of CD ratio across various models**
> > >
> > > We acknowledge the reviewer’s concern that the CD ratio is a **relative metric** rather than an absolute one. However, it is important to clarify that the CD ratio is designed to ***compare preferences for CD across multiple datasets*** rather than *across multiple models for a single dataset*. Its primary purpose is to provide comparisons ***between datasets*** trained on the same model. Therefore, we believe this issue does not significantly impact the effectiveness of the CD ratio as a metric for measuring channel dependence.
> > >
> > > &nbsp;
> > >
> > > ---
> > >
> > > ### **W2-5. Application to other TSFMs**
> > > PCD aims to enhance ***TSFMs capturing CDs*** by leveraging dataset-specific characteristics, rather than proposing a plug-in-play method applicable to any type of TSFMs existing in the world, as mentioned in **L60--61** and **L69--70**.  Specifically, CM is designed to adjust the **CD estimated by the model**, making it **applicable to CD models** but **infeasible for CI models**.
> > >
> > >
> > > &nbsp;
> > >
> > >
> > > Additionally, while we acknowledge that not all TSFMs rely on attention mechanisms and that Transformers are widely employed to capture temporal dependencies (TD), recent methods--particularly with the **emergence of iTransformer** [A]--have increasingly emphasized **leveraging attention mechanisms to capture CD** [B, C, D, E, F]. Our method aligns with this trend and builds upon the strengths of these approaches.
> > >
> > >
> > > &nbsp;
> > >
> > >
> > > Furthermore, the limitation of being applicable only to the attention mechanism for capturing CD has been ***already*** discussed in **L482--484**, where we identified the development of a novel approach to achieve PCD without relying on Transformer-based methods as a promising direction for future work.
> > >
> > >
> > > &nbsp;
> > >
> > >
> > > [A] Liu, Yong, et al. "iTransformer: Inverted transformers are effective for time series forecasting." ICLR (2024)
> > >
> > >
> > > [B] Gao, Shanghua, et al. "Units: Building a unified time series model." NeurIPS (2024)
> > >
> > >
> > > [C] Ilbert, Romain, et al.  “SAMformer: Unlocking the Potential of Transformers in Time Series Forecasting with Sharpness-Aware Minimization and Channel-Wise Attention.” NeurIPS (2024)
> > >
> > >
> > > [D] Wang, Xue, et al. “CARD: Channel aligned robust blend transformer for time series forecasting.” ICLR (2024)
> > >
> > >
> > > [E] Yu, Guoqi, et al.  “Revitalizing Multivariate Time Series Forecasting: Learnable Decomposition with Inter-Series Dependencies and Intra-Series Variations Modeling.” ICML (2024)
> > >
> > >
> > > [F] Dong, et al. "TimeSiam: A Pre-Training Framework for Siamese Time-Series Modeling." ICML (2024)
> > >
> > >
> > >
> > >
> > > &nbsp;
> > >
> > > ---
> > >
> > > ## **Q1.  Backbone for TimeSiam and additional experiments**
> > >
> > > Thank you for highlighting the need for more detail in our experimental settings.
> > >
> > > As mentioned in **W2-2**, the proposed CM is applicable to **CD models**; therefore, we applied our method to the **iTransformer [A]** for TimeSiam. Although the choice of iTransformer for Timesiam is straightforward in our context as it is the only CD model candidate, we agree that clarifying this is beneficial to readers. We have added further details in **Appendix E** to improve transparency.
> > >
> > > &nbsp;
> > >
> > >
> > > We clarify that **Table 1** correctly categorizes the task performed with TimeSiam, which is forecasting (***not classification***).  We could not experiment TimeSiam and TimeSiam+Ours in classification tasks because the official GitHub repository for Timesiam does not provide code for classification, and the datasets are also not easily accessible, as they require permission from the official website and additional data processing steps, which are not clearly guided.
> > >
> > > If the reviewer is particularly interested in classification tasks and can suggest alternative backbones, we would be glad to explore and incorporate such experiments into our paper.
> > >
> > >
> > > &nbsp;
> > >
> > >
> > > [A] Liu, Yong, et al. "iTransformer: Inverted transformers are effective for time series forecasting." ICLR (2024)
> > >
> > >
> > > &nbsp;
> > >
> > > ---
> > >
> > > ## **Q2. Formulation of $\mathbf{A}$ in Equation (1) for CI frameworks**
> > >
> > > It is crucial to clarify that **Equation 1** does not represent a *GENERAL attention* mechanism, but specifically refers to ***CHANNEL-WISE attention***, as noted in **L157--158** (~ *adjust the* ***CD estimated by the model*** *by performing element-wise multiplication~*).
> > >
> > > For **CI models** such as PatchTST, which focus on temporal attention rather than channel-wise attention, this formulation is consistent because the **identity matrix for $\mathbf{A}$** represents the absence of channel interactions, aligning with the CI framework. Thus the equation remains valid for these models.

---

> > > > ### Author Response · Authors · 2024-11-19
> > > >
> > > > ## **Q4.  CM under extreme CI cases**
> > > >
> > > > We believe that our experiments already **encompass a wide range of real-world datasets with varying levels of CD, including both high and low CD scenarios**. These datasets come from various domains, including 5 domains with a total of 15 datasets for forecasting and 7 domains with a total of 18 datasets for classification.
> > > >
> > > > &nbsp;
> > > >
> > > > To further address concerns regarding PCD’s performance under extreme cases, we performed an **additional toy experiment** in **Appendix G**. Specifically, we designed a scenario where the channels in TS exhibit **no correlation**. We generated a synthetic TS dataset with two channels using sine waves oscillating at frequencies of 0.5 and 2.0 over a length of 18,000 (similar to ETTh1). The results show that the **CD ratio of CM is approximately 0.018** and the **forecasting MSE is around 0.0014**, confirming strong channel independence and demonstrating the effectiveness of our method even under extreme CI conditions.
> > > >
> > > >
> > > >
> > > >
> > > > &nbsp;
> > > >
> > > > ---
> > > >
> > > >
> > > > ## **Q5.  Effectiveness of CM under non-stationarity**
> > > >
> > > >
> > > > As the reviewer mentioned, TS data can exhibit **non-stationarity** in the real-world scenarios. However, we have observed the effectiveness of CM under such conditions:
> > > > - **(1. Distribution shift)** The datasets used in our experiments (13 real-world datasets across various domains) show **significant distribution shifts** [A]. Results in **Tables 2,3,E.1** demonstrate the effectiveness of CM across these datasets, tested with three different backbones.
> > > > - **(2. Missingness in TS)** To address scenarios with missing data, which can introduce non-stationarity, we simulated datasets with 10%, 25%, 50%, and 75% of data points randomly missing. Missing values were linearly interpolated using adjacent points, as detailed in **L471--475**. The results in **Figure 10** demonstrate the robustness of our method under these missing data scenarios, both in terms of **a) forecasting performance** and  **b) the CD ratio of the CM**, further validating its applicability and effectiveness under non-stationarity.
> > > >
> > > >
> > > > &nbsp;
> > > >
> > > >
> > > > Regarding the reviewer’s concern about the ability of CM to capture changes in CD over time, this issue is addressed in detail in **W1-2**, where we discuss how (global) CM focuses on capturing stable, dataset-specific characteristics, while temporal changes in CD are managed by the (local) attention matrix.
> > > >
> > > >
> > > > &nbsp;
> > > >
> > > >
> > > > [A] Han, Lu, Han-Jia Ye, and De-Chuan Zhan. "The capacity and robustness trade-off: Revisiting the channel independent strategy for multivariate time series forecasting."  TKDE (2024)
> > > >
> > > >
> > > > &nbsp;
> > > >
> > > > ---
> > > >
> > > > ## **Q6.  Other metrics besides correlation**
> > > >
> > > >
> > > > **a) Emphasis on utilizing dataset-specific information (rather than “correlation” itself)**
> > > >
> > > >
> > > > We appreciate the reviewer’s concern regarding the **suitability of correlation** as a metric for measuring channel relationships. However, our primary focus is not on using the "correlation" metric, but on ***leveraging the "dataset-specific information"***, as highlighted in **L16,60,480**.
> > > >
> > > >
> > > > &nbsp;
> > > >
> > > >
> > > > **b) Metrics other than correlation**
> > > >
> > > >
> > > > While our primary focus is on utilizing **dataset-specific information** rather than the correlation metric itself, we acknowledge the reviewer’s concern regarding the choice of metrics for capturing channel relationships. As detailed in **Table 10**,  we have ***already*** conducted an analysis using **four different metrics** - Correlation, Euclidean, Cosine, Dynamic Time Warping (DTW) - demonstrating consistent performance improvements across metrics.

---

> > > > > ### Author Response · Authors · 2024-11-19
> > > > >
> > > > > While we believe we addressed all your concerns in our rebuttal and the revised manuscript, please feel free to ask us if we may have missed anything or you have additional questions; we are happy to address them.

---

> ### Comment · Reviewer_21yw · 2024-12-02
>
> I apologize for my delayed response, which was due to an accident early in the last Thanksgiving break. I appreciate the authors' efforts and detailed responses to my comments as well as those from the other reviewers. I now understand this work better than before. I acknowledge the motivation of this work, but I believe the authors' argument that their proposed method can be used for various TSFMs (despite the fact that most TSFMs have emerged recently) or at least existing CD-based models is not supported by either concrete theoretical analysis or a rich set of experimental results. I think that the authors' conceptual elaboration on the theoretical grounding and the limited empirical results focused on experiments with UniTS and iTransformer are still insufficient to support their argument. In conclusion, I believe the authors' claim of broad applicability for the proposed method is not well supported. Personally, I think the authors do not need to claim that their method can be applied to multiple TSFMs but should instead focus on rigorously demonstrating the methodological contribution of PCD to capture varying CD across different datasets and models.

---

> > ### Author Response · Authors · 2024-12-03
> >
> > Thank you for further engaging in the discussion!
> >
> > &nbsp;
> >
> > > ### **1. Application to TSFMs capturing CD**
> >
> >
> > Regarding the reviewer’s statement *”authors' argument that their proposed method can be used for various TSFMs”*, we believe this is a misunderstanding of our argument.
> > As stated in (1) discussions with Reviewer saeS, (2) the general comment (GC), and (3) the revised manuscript, PCD is designed to enhance **TSFMs that capture CD through attention mechanisms** by leveraging dataset-specific characteristics, **rather than acting as a universal solution for all TSFMs**.
> >
> >
> > Please refer to the following for further details:
> > - **Discussion with Reviewer saeS**
> > - **GC3. Generality of our method**
> > - **L55--56**: "which adjusts the CD ***estimated by the Transformer-based model***"
> > - **L60--61**: is multiplied to the (channel-wise) attention matrix (i.e., ***CD estimated by the model***)"
> > - **L69--70**: "The proposed CM is a plug-and-play method applicable to any model that ***captures CD using an attention mechanism***"
> >
> > &nbsp;
> >
> > Furthermore, the contribution of our work, regardless of some reviewers' expectations regarding its scope, has been addressed in **Follow-up discussion on 3. Generality of our method (21yw, bW9C, saeS)**, where we believe our work remains significant in other aspects.
> >
> >
> > &nbsp;
> >
> > ---
> > > ### **2. Broad applicability of the proposed method**
> >
> > Regarding the reviewer’s statement *”limited empirical results focused on experiments with UniTS and iTransformer are still insufficient to support their argument”*, we believe this has **been addressed**.
> >
> > &nbsp;
> >
> > As shown in **Table 1**, we have conducted extensive experiments with **three different CD models** (UniTS, iTransformer, and TimeSiam; as previously mentioned, our method applies only to CD models) with four different tasks, which we believe is sufficient.
> >
> > The use of only a **single TSFM (NeurIPS 2024)** among the three models was raised by Reviewer saeS and has  been addressed in **Section 4: Why not apply our method to other CD models, e.g., Moirai?**, leading us to incorporate two alternative SOTA models (ICLR 2024, ICML 2024).
> > This limitation stems from the unavailability of trainable code for certain models and the impracticality of testing large, complex concurrent works, which have also been discussed.
> >
> > &nbsp;
> >
> > ---
> > > ### **3. Demonstrating the contribution of PCD**
> >
> > Regarding the reviewer’s statement *”should instead focus on rigorously demonstrating the methodological contribution of PCD to capture varying CD across different datasets and models.”*, we believe this has been thoroughly addressed through **extensive empirical evaluations**.
> >
> > Below, we summarize the experiments conducted to **demonstrate the effectiveness of the PCD** concept and the proposed channel masks (CMs), validated both (a) quantitatively and (b) qualitatively:
> >
> > 1. (Backbone 1) UniTS
> > - (a) Table 7: Ablation study regarding the use of correlation matrix and domain parameters.
> > - (a) Figure 7: Domain parameters enable the CD ratio to better reflect the preference for channel dependencies.
> > - (b) Figure 8: Domain parameters capture the hidden relationships between channels, which are unobservable in the data space.
> > - (b) Table 8 (+ Figure 6): TS visualization vs. CD ratio (w/ and w/o domain parameters).
> > 2. (Backbone 2) iTransformer
> > - (a) **Table 13 (+ Table 15)**: Various options for domain parameters.
> > - (a) **Figure 10**: Effectiveness of CM under missing scenarios
> > 3. (Backbone 3) TimeSiam
> > - (a) **Appendix E.1**: Ablation study regarding the use of correlation matrix and domain parameters.
> >
> > &nbsp;
> >
> > If you have specific suggestions for additional analyses to further demonstrate the contributions of PCD and CM, we would be happy to incorporate them in the revised version.
> >
> > &nbsp;
> >
> >
> > We hope the above three responses have addressed your concerns.

---

### Author Response · Authors · 2024-11-23

### **7. Inconsistent results with the original paper (bW9C, JKMr)**

We reproduced experiments using the official iTransformer [A] and UniTS [B] code repositories without modifications, but found that iTransformer exhibited similar issues reported on GitHub, and UniTS required setting the hidden dimension to $D=32$ to resolve convergence issues, which resulted in degraded performance, with further details in **Appendix B**.

&nbsp;

### **8. Minor Issues**
- **Varying Size of Lookback Window $L$ (bW9C)**: We set $L$ to match the backbone method for a fair comparison. Following the reviewer’s feedback, we conducted an ablation study (**Appendix F**) with different values of $L$ across multiple datasets. The results demonstrate that the impact of CM remains consistent regardless of $L$, validating its robustness.

- **Additional Related works (bW9C)**: While LIFT [A] and CDAM [B] are related to our work in handling CD, no plug-in methods specifically capture varying CD across multiple datasets for TSFMs. Nonetheless, we attempted to compare our method with them in a single-task model, but LIFT lacks support for recent CD models, and CDAM does not provide code. We have included both methods in the **2.Related Works** section (**L87--89**).

&nbsp;

For your convenience, we have uploaded a PDF that includes both the revised manuscript and appendices, with key changes highlighted in **green**. We believe the discussions and empirical results will strengthen our contribution. If there are any points we may have missed or if you have further questions or suggestions, please do not hesitate to share them with us. We would be happy to address them.

&nbsp;

Thank you very much.

---

### Author Response · Authors · 2024-11-23

# General Comment
&nbsp;

Dear AC and reviewers,

First of all, we deeply appreciate your time and effort in reviewing our paper. Our work introduces the concept of **partial channel dependence (PCD)**, which adjusts channel dependencies (CD) captured by the model based on dataset-specific information. The approach uses a **channel mask (CM)** composed of two components: a correlation matrix for relative dependencies and domain parameters for absolute dependencies, refining the correlation matrix for each dataset.

As highlighted by the reviewers, our work introduces a **novel approach** addressing implicit heterogeneity across TS datasets (21yw, BSUW). It is **straightforward and well-motivated** (bW9C, JKMr), supported by **extensive experiments** across various tasks (all). The method demonstrates improved performance in challenging scenarios, including few-shot and zero-shot settings (BSUW, JKMr), while effectively capturing dataset-specific dependencies (21yw, BSUW).

&nbsp;

In our responses, we have addressed the concerns raised by all reviewers and strengthened our claims with additional analyses. Below, we present key highlights to support your post-rebuttal discussion (all sections/figures/tables labeled according to the revised version):

&nbsp;

### **1. Heterogeneity across TS datasets (21yw, JKMr)**

Reviewers (21yw, JKMr) asked to further clarify the **definition of heterogeneity** across TS datasets. As stated in **L37--43**, explicit heterogeneity refers to observable differences like sequence length and number of channels, while implicit heterogeneity involves unobservable factors, such as varying channel dependencies (CD) across datasets. To enhance clarity, we have revised the **1.Introduction** section.

&nbsp;

### **2. Necessity of capturing varying CD across datasets (21yw, JKMr)**
Reviewers (21yw, JKMr) questioned the necessity of capturing varying CDs across datasets. As noted in **L43--46**, addressing implicit heterogeneity is essential for TSFMs, which are trained on **multiple** datasets with varying CDs which **require different approaches** (e.g., CI or CD). We show that considering this heterogeneity improves performance, as shown in **Table 7** (CD vs. PCD).

&nbsp;

### **3. Generality of our method (21yw, bW9C, saeS)**
Reviewers (21yw, bW9C, saeS) raised concerns about the generality of our method. However, PCD is proposed to enhance TSFMs that **capture CDs with attention mechanism** by leveraging dataset-specific characteristics, rather than being a universal solution for all TSFMs, as noted in **L60--61** and **L69--70**. We focus on TSFMs capturing CDs due to the growing emphasis on attention mechanisms for capturing CDs, and our approach builds on this trend and its strengths. Additionally, CD models, with their higher capacity, are better suited for large or multiple datasets, aligning with the use of CM.

&nbsp;

### **4. Appropriateness of correlation metric (21yw, saeS, JKMr)**
Reviewers (21yw, saeS, JKMr) questioned the appropriateness of using correlation as a metric. While our primary focus is on utilizing **dataset-specific information** rather than the correlation metric itself, we acknowledge the reviewers' concerns regarding the choice of metrics for capturing channel relationships. As detailed in **Table 10**, we have already conducted an analysis using four different metrics -- Correlation, Euclidean, Cosine, and Dynamic Time Warping (DTW) -- showing consistent performance improvements across all metrics.

&nbsp;

### **5. Design choice of CM and attention matrix (21yw, JKMr)**
Reviewers (21yw, JKMr) raised concerns about our design choice of the global CM and attention matrix. The (global) CM is designed to capture stable characteristics across time, focusing on global CD rather than adapting to local changes. Adjusting the CM for local variations would undermine its stability. To capture local variations, we use the attention matrix, which dynamically adjusts to the input time series. This approach ensures both stability and adaptability, as demonstrated in experiments on 13 real-world datasets with significant distribution shifts (**Table 12**). Additional experiments with the Weather dataset showed minimal changes in the global CM during local fluctuations (L2 distance of 0.0002), confirming its stability.

&nbsp;

### **6. Effectiveness of CM under non-stationarity (21yw)**
As noted by a reviewer (21yw), TS data can exhibit non-stationarity. However, we observed CM's effectiveness under such conditions through experiments on 13 real-world datasets with distribution shifts (**Tables 2, 3, E.1**) and datasets with varying levels of missing data (10%, 25%, 50%, 75%), as detailed in **L471--475**. Results in **Figure 10** show that CM remains robust in both forecasting performance and the CD ratio even under these non-stationary scenarios.

---

> ### Comment · Reviewer_saeS · 2024-11-25
>
> It is worth emphasizing that, first of all, the author may have misunderstood my summary.  ***I did not think that the method proposed in this paper is novel, but only used the words "a concise approach"***. Secondly, just as most reviewers have concerns about the generalization of the proposed method, the author emphasizes in the title and lines 22-23 that this paper is suitable for ***time series foundation models***, but the authors ***refuse to employ other foundation models (except UniTS )*** conducted more experiments to demonstrate the generalization of the proposed method. Therefore, I will keep my original rating.

---

> > ### Author Response · Authors · 2024-11-25
> >
> > Our paper advances **TSFM that captures CD with attention mechanisms** and introduces the concept of **PCD** to emphasize the importance of **both model architecture and “datasets”** when building TSFMs, with this approach validated by achieving SOTA across four tasks using three prominent models.
> >
> > While this is an emerging field (most works published in 2024) and a few algorithms focus on capturing CD with attention, we believe it will provide a strong foundation for future research as (1) TSFMs increasingly use **large-scale datasets**, amplifying the importance of CD, and (2) recent TS forecasting models (not limited to TSFM) also highlight CD's role, as CI is suboptimal when CD is not effectively captured by the model.
> >
> > &nbsp;
> >
> > We hope saeS understands the points of misunderstanding in our work via our responses and reconsiders their evaluation. Furthermore, we are actively exploring TSFMs which capture CD using an attention mechanism and will try to incorporate relevant findings until the camera-ready submission.

---

> ### Author Response · Authors · 2024-11-25
>
> Thank you for further continuing to engage in the discussion.
>
> While we appreciate the reviewer's efforts, we believe that some criticisms may stem from misunderstandings of our work and research on FMs. Below we further clarify them and provide detailed responses below.
>
> &nbsp;
>
> > **I did not think that the method proposed in this paper is novel, but only used the words "a concise approach".**
>
> We apologize for our miscategorization of reviews when summarizing them. We updated our general comment accordingly.
>
> &nbsp;
>
> > **the author emphasizes in the title and lines 22-23 that this paper is suitable for time series foundation models**
>
> Thank you for specifically raising this concern. Our **title** already addresses ***"channel dependence (CD)"***: namely, Partial CD (with CM) for TSFMs. Additionally, the statements in **L69--70** (***"The proposed CM is a plug-and-play method applicable to any model that captures CD using an attention mechanism"***) and **L55--56** (***"which adjusts the CD estimated by the Transformer-based model"***) both demonstrate that the scope is not "all" TSFMs but specifically models that capture CD using attention mechanisms. **L22-23** the reviewer mentioned refers to the applicability to UniTS, iTransformer, and TimeSiam that we experimented with. Hence, we believe this title already limits the application of our PCD to TSFMs modeling CDs, so we thought that it reflects its applicability while concise enough. To account for your concern, *"Partial Channel Dependence with Channel Masks for **Channel-Dependent** Time Series Foundation Models"* would be an alternative, while it has redundancy because of repeating "CD". We think this requires further discussion with other reviewers, and if they agree, we are willing to revise the title accordingly.
>
> &nbsp;
>
> > **but the authors refuse to employ other foundation models (except UniTS ) conducted more experiments to demonstrate the generalization of the proposed method.**
>
> We emphasize that we **did NOT refuse** to experiment other FMs; rather, it is **infeasible** due to fundamental limitations on them. As previously stated, our method is designed for "models that capture CDs using attention mechanisms." The models mentioned by saeS do not fall into this category, and hence, they cannot be used. Instead, we experimented with iTransformer [ICLR 2024] and TimeSiam [ICML 2024] that capture CDs using attention mechanisms to show the applicability of our proposed method.
>
> &nbsp;
>
> ---
>
> We found that the reasons why saeS gives rejection is summarized as
> - 1) the title is somewhat broad, and
> - 2) we did not experiment with FMs published this year.
>
> &nbsp;
>
> However, regarding 1), we are willing to revise upon consensus with reviewers, so this should not be the reason for rejection.
>
> &nbsp;
>
> Regarding 2), we argue that this is not reasonable for recent research in deep learning, especially for FMs; for example, the time between the ICLR 2025 submission and ICML 2024 presentation was only around **2 months**, and saeS is **asking us to address many papers including 4 ICML 2024 papers [2--5] with experiments**, which is not feasible. The reason why we could make quick progress on top of UniTS [NeurIPS 2024] is that they released their code before acceptance, and the model/dataset size of UniTS was relatively small while applicable to multiple tasks; however, all works mentioned by saeS do not meet this condition, i.e., they did not make their work quickly reproducible. We believe research in FMs is quickly progressing in this way, and asking a research on FMs to experiment ALL concurrent works would be a bottleneck of the progress in this field. There might be multiple ways of building TSFMs, and we believe our proposed PCD is a meaningful step forward progressing TSFMs capturing CDs.
>
> In other words, for our understanding, your criticism is *"your work is on top of a NeurIPS 2024 paper, and I am going to reject your paper because you did not take account of several ICML 2024 papers,"* which we believe is not appropriate for research on deep learning and/or FMs. Generally speaking, reviews with this kind of criticism will simply result in rejecting most research on FMs.
>
>
> Furthermore, all concurrent works saeS mentioned are **out of scope** for our work due to the inherent difference in the model architecture.

---

> ### Author Response · Authors · 2024-11-25
>
> ## Follow-up discussion on **3. Generality of our method** (21yw, bW9C, saeS)
>
> We emphasize again that our method is designed to work exclusively with models that **capture CD using attention mechanisms**, as stated in **L55--56** and **L69--70**. We admit that the emphasis on the scope of applicability in the initial submission might be insufficient. To address this, we have revised the manuscript to clarify the applicability of the proposed **PCD** during the rebuttal; we emphasized the changes in green.
>
> &nbsp;
>
> Although it might not meet some reviewers' expectations regarding the scope, we believe our contribution is still significant in other aspects:
>
> - ***CD models' relevance to TSFMs***: We highlight the importance of CD, as CI can be achieved with CD by disregarding dependencies between irrelevant channels when well captured [A].  We further argue that the CD architecture is especially crucial for TSFMs which are trained on large-scale datasets, due to the **capacity-robustness trade-off**, where CD models benefit more from larger datasets than CI models [B].
>
> - ***Emphasis on dataset when building TSFMs***: Our work highlights the importance of considering both model architecture and **datasets** when constructing TSFMs. As large-scale datasets and TSFMs become more prevalent, capturing **varying CD across datasets** becomes increasingly critical, as each dataset may require a different approach (CI or CD). Our proposed PCD addresses this need by explicitly accounting for **dataset-specific characteristics**.
>
> &nbsp;
>
> We recognize that the current title might suggest broader applicability than intended, though we initially thought that "Partial CD (with CM) for TSFMs" already limits the application of our PCD to "TSFMs modeling CDs." We welcome suggestions for a more precise title. For example, *"Partial Channel Dependence with Channel Masks for **Channel-Dependent** Time Series Foundation Models"* would be an alternative, while it has redundancy because of repeating "CD". Please feel free to share your opinion with us, we are willing to revise the title accordingly.
>
> &nbsp;
>
> [A] Nie, Yuqi, et al. "A time series is worth 64 words: Long-term forecasting with transformers." ICLR (2023)
>
> [B] Han, Lu, Han-Jia Ye, and De-Chuan Zhan. "The capacity and robustness trade-off: Revisiting the channel independent strategy for multivariate time series forecasting." TKDE (2024)

---

> > ### Comment · Reviewer_saeS · 2024-11-30
> >
> > My concerns are summarized below:
> >
> > - **Limitation of Generalization**: PCD can only be applied to models that use attention mechanisms to capture correlations, which significantly limits its scope of application. Most time series foundation models adopt CI strategies, making PCD inapplicable. Moreover, for the few models like MOIRAI[1] that consider correlation through positional encoding, PCD is still unsuitable.
> >
> > - **Effectiveness of the Method**: Furthermore, to demonstrate the effectiveness of the method, I believe it is necessary to compare it with state-of-the-art time series foundation models, such as MOIRAI[1], Timer[2], and TTM[3].( Of course, you do not need to conduct experiments at this stage. You can refer to the results in the original papers or the reproduction results of these models in other works.)
> >
> > I appreciate the effort in your rebuttal, but for the reasons stated above and combined with other reviewers' feedback, I will maintain my score, and I think this work is not yet ready for publication.
> >
> > [1] Woo, G., Liu, C., Kumar, A., Xiong, C., Savarese, S., & Sahoo, D. (2024). Unified training of universal time series forecasting transformers. *arXiv preprint arXiv:2402.02592*.
> >
> > [2] Liu, Y., Zhang, H., Li, C., Huang, X., Wang, J., & Long, M. (2024). Timer: Transformers for time series analysis at scale. arXiv preprint arXiv:2402.02368.
> >
> > [3] Ekambaram, V., Jati, A., Nguyen, N. H., Dayama, P., Reddy, C., Gifford, W. M., & Kalagnanam, J. (2024). TTMs: Fast Multi-level Tiny Time Mixers for Improved Zero-shot and Few-shot Forecasting of Multivariate Time Series. arXiv preprint arXiv:2401.03955.

---

> ### Author Response · Authors · 2024-12-01
>
> Thank you for providing a summary of your concerns. While we appreciate Reviewer saeS is aware of recent progress in TSFMs and trying to provide helpful comments, we believe your concerns come from **misunderstanding of our work and/or progress in this area of research**. Below we response to each of your concern:
>
> &nbsp;
>
> > ### 1. *"PCD can only be applied to models that use attention mechanisms to capture correlations, which significantly limits its scope of application."*
>
> We believe *"limited scope of application"* does not deteriorate the contribution of our work, as the scope of general TSFMs is **too broad**. In particular, "foundation model" is a broad term that does not specify its methodology. For example, we believe no one would criticize the popular Wasserstein GAN paper because it was not applied to other generative models, such as VAE or diffusion models.
>
> Indeed, our method effectively improves the state-of-the-art TSFM (and 2 other models) in this line and we compared it with 2 TSFMs (UniTS [1] and GPT4TS [14]), validated through extensive experiments.
>
> &nbsp;
>
> > ### 2. *"Most time series foundation models adopt CI strategies"*
>
> We ***respectfully disagree with this statement***.  Based on our survey, **5 out of 9 TSFMs adopt CD strategies**, while 4 of them adopt CI strategies. Below we summarize recent TSFM works, which were mostly uploaded this year (2024), including the works cited by the reviewer:
>
> CD Models
> - [1] UniTS (NeurIPS 2024, final code update: 2024.02)
> - [2] Moirai (ICML 2024, final code update: 2024.11)
> - [3] GTT (CIKM 2024, final code update: 2024.02)
> - [4] UniST (KDD 2024, final code update: 2024.07)
> - [5] Tiny Time Mixers (TTMs) (arXiv 2024.01, no GitHub / available only on Hugging Face): Pretrained with CI, fine-tuned with CD
>
> CI Models
> - [6] Timer (ICML 2024): CI
> - [7] TimesFM (ICML 2024): CI
> - [8] MOMENT (ICML 2024): CI
> - [9] ROSE (arxiv 2024.05): CI
>
> Reviewer saeS's misunderstanding seems to come from the fact that TSFMs published in ICML 2024 mostly adopt CI strategies.
>
> &nbsp;
>
> > ### 3. "for the few models like Moirai [1] that consider correlation through positional encoding, PCD is still unsuitable."
>
> ***We also respectfully disagree with this assertion***. Moirai [2] applies attention along the time axis after **channel flattening**. As these methods utilize the attention mechanism to **capture both TD and CD simultaneously**, the proposed CM can still be applied by extending its shape from $C \times C$ to $C \cdot P \times C \cdot P$, where $P$ is the number of patches in each channel, by duplicating the values within the same channels. Furthermore, as the reviewer mentioned, Moirai considers correlation through "positional encoding"; however, this is intended to *distinguish between channels in a flattened single sequence*, which is orthogonal to our approach.
>
> &nbsp;
>
> > ### 4. *Why not apply our method to other CD models, e.g., Moirai?*
>
> Although five CD models are listed above, the following explains why only UniTS [1] was used:
>
> - [2] **Moirai (ICML 2024)**:
> Their GitHub repository **does not provide enough information** for replicating pretraining, such that implementing "Moirai + CM" requires more effort than adding CM on top of it.
> Additionally, the reproduced fine-tuning results **perform significantly worse** than the zero-shot performance reported in the original paper (e.g., MSE of 0.424 (reproduced) vs. 0.375 (original paper) on ETTh1 with $H=96$),
> which is not only our issue but ***also shared by others in the GitHub issue***.
> Furthermore, experiments are **only feasible with the ETTh1 dataset**.
> Additionally, the **dataset size (231B)** and the base **model size (91M)** make experiments not feasible to moderate experimental setup (they specified A100 for hardware).  In fact, the official Moirai GitHub focuses mainly on enabling users to utilize the model as a package, e.g., performing zero-shot inference on their own datasets via Jupyter Notebooks, rather than replicating their model from scratch.
>
> - [3] **GTT (CIKM 2024)**: Only provides inference and fine-tuning code, ***with NO pretraining code available***.
> - [4] **UniST (KDD 2024)**: Designed specifically for **urban spatio-temporal prediction**, using “4D-grid” spatial input $(N,T,H,W)$, where $H$ and $W$ represent spatial dimensions.
> - [5] **Tiny Time Mixers (TTMs) (arXiv 2024)**: It employs a special non-Transformer architecture without **attention mechanism**.

---

> ### Author Response · Authors · 2024-12-01
>
> > ### 5. *to demonstrate the effectiveness of the method, I believe it is necessary to compare it with state-of-the-art time series foundation models*
>
>
> ### Why not compare with other (CI or CD) TSFMs?
>
> The reviewer further mentioned that we could still compare our method with the **concurrent TSFMs**. However, most TSFMs have emerged recently, and the field is still in its early stages, with no standardization and significant variability in pretraining datasets and experimental setups. This makes ***direct comparisons among TSFMs difficult, and only recently has research begun to address the necessity of benchmarking in this area [10, 11, 12], with both papers submitted as part of the ICLR 2025 submissions.***.
>
> Notably, even Moirai [2], cited by Reviewer saeS, does not compare its results with other TSFMs. Similarly, TimesFM compares only with GPT4TS [10], an earlier LLM-based TSFM.
>
> We emphasize that we already incorporated the comparison with two state-of-the-art TSFMs (UniTS and GPT4TS). Since other concurrent TSFM works also compare a similar number (0--2) of TSFMs, we believe this is sufficient. Nonetheless, we are actively exploring other TSFM works, and will try to incorporate relevant findings until the camera-ready submission.
>
> &nbsp;
>
> Indeed, several works including studies on benchmarking TSFMs [11--15] report **substantial variations in performance** for the same TSFM ***due to differences in experimental setups***.
>
> For instance, on the Weather dataset, the zero-shot performance of **Moirai** measured by averaged MSE across four prediction lengths $\\{96,192,336,720\\}$ **varies significantly**:
> - Moirai [2]: 0.238 (Table 6)
> - FlexTSF [14]: 0.246 (Table 3)
> - Time-MoE [13, ICLR 2025 submission]: 0.287 (Table 3)
> - FoundTS [12]: 0.312 (Table 11)
> - Gift-Eval [11]: used different metrics (MAPE, CRPS, rank)
> - WaveToken [15, ICLR 2025 submission] : used different metrics (WQL, MASE, VRSE)
>
> From **Table 3** in our paper, the average MSE of ours is 0.245. Based on the replication of Moirai by concurrent works, our work is competitive to Moirai with much less parameters (1.57M vs. 91M), though we cannot assure the comparison is fair.
>
> As setting a fair comparison of TSFMs is beyond our scope, we leave it to future works.
>
> &nbsp;
>
> [1] Gao, Shanghua, et al. "Units: A Unified Multi-task Time Series Model." NeurIPS (2024)
>
> [2] Woo et al. “Unified training of universal time series forecasting transformers.” ICML (2024)
>
> [3] Feng et al. “Only the Curve Shape Matters: Training Foundation Models for Zero-Shot Multivariate Time Series Forecasting through Next Curve Shape Prediction.” arxiv (2024)
>
> [4] Yuan, Yuan, et al. "Unist: a prompt-empowered universal model for urban spatio-temporal prediction." KDD (2024)
>
> [5] Ekambaram, V. et al. “TTMs: Fast Multi-level Tiny Time Mixers for Improved Zero-shot and Few-shot Forecasting of Multivariate Time Series”, arxiv (2024)
>
> [6] Liu, Y., Zhang, H., Li, C., Huang, X., Wang, J., & Long, M. (2024). Timer: Transformers for time series analysis at scale. arXiv preprint arXiv (2024)
>
> [7] Das, Abhimanyu, et al. "A decoder-only foundation model for time-series forecasting." (ICML 2024)
>
> [8] Goswami, et al. “Moment: A family of open time-series foundation models.” ICML (2024)
>
> [9] Wang, Yihang, et al. "ROSE: Register Assisted General Time Series Forecasting with Decomposed Frequency Learning." arxiv (2024)
>
> [10] Zhou, et al. “One fits all: Power general time series analysis by pretrained lm.” NeurIPS (2023)
>
> [11] Aksu et al., “GIFT-Eval: A Benchmark For General Time Series Forecasting Model Evaluation” NeurIPS Workshop (2024)
>
> [12] Z Li et al., “FoundTS: Comprehensive and Unified Benchmarking of Foundation Models for Time Series Forecasting” arxiv (2024)
>
> [13] https://openreview.net/forum?id=e1wDDFmlVu
>
>
> [14] Xiao, Jingge, et al. "FlexTSF: A Universal Forecasting Model for Time Series with Variable Regularities." arxiv (2024)
>
> [15] https://openreview.net/forum?id=D9liZ0D8z8

---

### Comment · Area_Chair_6QSb · 2024-11-26
**Encouragement to Actively Participate in the Discussion Phase**

Dear Reviewers,

Thank you for your valuable contributions to the review process so far. As we enter the discussion phase, I encourage you to actively engage with the authors and your fellow reviewers. This is a critical opportunity to clarify any open questions, address potential misunderstandings, and ensure that all perspectives are thoroughly considered.

Your thoughtful input during this stage is greatly appreciated and is essential for maintaining the rigor and fairness of the review process.

Thank you for your efforts and dedication.

---

### Meta-Review · Area_Chair_6QSb · 2024-12-19

**Metareview:**

(a) Summary of Scientific Claims and Findings
The submission introduces Partial Channel Dependence, a concept designed to handle varying channel dependencies in multivariate time series datasets by using a proposed Channel Mask. This mask integrates two components: (1) a correlation matrix to encode relative dependencies and (2) domain-specific parameters for dataset-specific absolute dependencies. The method is implemented into Transformer-based time series foundation models and is evaluated across tasks such as forecasting, classification, imputation, and anomaly detection, with special emphasis on few-shot and zero-shot scenarios.

(b) Strengths of the Paper
Novelty: The introduction of PCD to address implicit heterogeneity across datasets is novel and demonstrates improved adaptability compared to existing methods focused solely on explicit heterogeneity.
Empirical Evaluation: Extensive experiments are conducted across diverse tasks and settings. The inclusion of ablation studies helps clarify the role of the CM components in improving performance.
Simplicity and Scalability: The CM design introduces minimal additional parameters, making it computationally efficient and scalable.

(c) Weaknesses of the Paper
Limited Generalization: The method is restricted to Transformer-based models that use attention mechanisms for channel dependencies. It is inapplicable to many existing TSFMs that adopt channel-independent (CI) strategies or alternate CD formulations.
Theoretical Gaps: The paper lacks a rigorous theoretical analysis to explain the observed performance gains or validate the CD ratio as a universal metric for channel dependence across models.
Inadequate Comparison: Comparisons with concurrent state-of-the-art models are either missing or insufficient, particularly for newly proposed TSFMs such as MOIRAI, Timer, and TTMs.
Validation Across Architectures: The experiments are limited to a subset of TSFMs and specific datasets. Broader architectural compatibility and robustness under non-standard scenarios, such as extreme non-stationarity, are insufficiently validated.
Reviewer Critiques: Persistent misunderstandings in the rebuttal stage, such as inadequate generalization or misalignment with prior works, indicate potential issues in communication and framing.

(d) Reasons for Rejecting the Paper
Scope Limitation: The method’s applicability is confined to attention-based CD models, leaving out a significant portion of TSFM research.
Lack of Robust Benchmarking: While the authors conducted comprehensive experiments, the absence of meaningful comparisons with concurrent models diminishes the credibility of the claimed state-of-the-art performance.
Incomplete Theoretical Contributions: The lack of a rigorous framework for implicit heterogeneity and the CD ratio metric weakens the foundational significance of the proposed approach.
Feedback from Reviewers: Significant concerns from reviewers about limited generalizability, insufficient theoretical insights, and inadequate comparisons were not fully addressed in the rebuttal phase.

**Additional Comments On Reviewer Discussion:**

Concern: Multiple reviewers questioned the limited applicability of the proposed PCD method. It is designed for Transformer-based models with channel dependence (CD) mechanisms and is not generalizable to models using channel independence (CI) or alternate CD formulations.
Author Response: The authors acknowledged this limitation but argued that the method’s scope was intentionally focused on CD models due to their relevance in handling multivariate datasets. They proposed future work to explore extensions to non-attention-based models.
Decision Impact: While the authors provided reasonable explanations, the lack of broader applicability reduced the work’s significance, especially in a rapidly evolving TSFM domain.

Concern: Reviewers requested comparisons with recent state-of-the-art models like MOIRAI, Timer, and TTMs. The absence of these comparisons was seen as a critical weakness, particularly since these models were published before the submission deadline.
Author Response: The authors argued that these comparisons were infeasible due to a lack of publicly available code, different experimental setups, and resource constraints. They highlighted their comparisons with UniTS and GPT4TS, which are prominent baselines.
Decision Impact: The justification was partially accepted, but the absence of key benchmarks weakened the claims of state-of-the-art performance.

Concern: Reviewers pointed out a lack of theoretical rigor in defining and validating implicit heterogeneity, the CD ratio metric, and the rationale behind the proposed method’s design choices.
Author Response: The authors provided additional clarifications and revisions to the introduction, emphasizing implicit heterogeneity and explaining the CM’s components in more detail. However, they admitted the absence of in-depth theoretical grounding and deferred this to future work.
Decision Impact: The response improved clarity but did not address the core issue of insufficient theoretical contributions, leaving the work conceptually underdeveloped.
The paper was deemed not ready for publication at ICLR due to these shortcomings.

---

### Decision · Program_Chairs · 2025-01-22

Reject